



# Response of carbon and water fluxes to meteorological and phenological variability in two eastern North American forests of similar age but contrasting species composition – a multiyear comparison

**Eric R. Beamesderfer**[1,a], **M. Altaf Arain**[1], **Myroslava Khomik**[1], **Jason J. Brodeur**[1], and **Brandon M. Burns**[1]

[1]School of Geography and Earth Sciences and McMaster Centre for Climate Change, McMaster University, Hamilton, Ontario, L8S 4L8, Canada
[a]now at: School of Informatics, Computing & Cyber Systems, Northern Arizona University, Flagstaff, Arizona, 86011, United States

**Correspondence:** M. Altaf Arain (arainm@mcmaster.ca)

**Abstract.** The annual carbon and water dynamics of two eastern North American temperate forests were compared over a 6-year period from 2012 to 2017. The geographic location, forest age, soil, and climate were similar between the two stands; however, stand composition varied in terms of tree leaf-retention and shape strategy: one stand was a deciduous broadleaf forest, while the other was an evergreen needleleaf forest. The 6-year mean annual net ecosystem productivity (NEP) of the coniferous forest was slightly higher and more variable ($218 \pm 109\,\mathrm{g\,C\,m^{-2}\,yr^{-1}}$) compared to that of the deciduous forest NEP ($200 \pm 83\,\mathrm{g\,C\,m^{-2}\,yr^{-1}}$). Similarly, the 6-year mean annual evapotranspiration (ET) of the coniferous forest was higher ($442 \pm 33\,\mathrm{mm\,yr^{-1}}$) than that of the deciduous forest ($388 \pm 34\,\mathrm{mm\,yr^{-1}}$), but with similar interannual variability. Summer meteorology greatly impacted the carbon and water fluxes in both stands; however, the degree of response varied among the two stands. In general, warm temperatures caused higher ecosystem respiration (RE), resulting in reduced annual NEP values – an impact that was more pronounced at the deciduous broadleaf forest compared to the evergreen needleleaf forest. However, during warm and dry years, the evergreen forest had largely reduced annual NEP values compared to the deciduous forest. Variability in annual ET at both forests was related most to the variability in annual air temperature ($T_a$), with the largest annual ET observed in the warmest years in the deciduous forest. Additionally, ET was sensitive to prolonged dry periods that reduced ET at both stands, although the reduction at the coniferous forest was relatively larger than that of the deciduous forest. If prolonged periods (weeks to months) of increased $T_a$ and reduced precipitation are to be expected under future climates during summer months in the study region, our findings suggest that the deciduous broadleaf forest will likely remain an annual carbon sink, while the carbon sink–source status of the coniferous forest remains uncertain.

## 1 Introduction

Temperate forests play a significant role in the global carbon and water cycles through their photosynthetic $CO_2$ uptake and through their evapotranspiration (ET) (Huntington, 2006; Houghton, 2007). In eastern North America, temperate forests are a significant sink of carbon and are an important element of future climate mitigation strategies; however, these forests have been going through transformations due to both natural and anthropogenic impacts for quite some time (Bonan, 2008; Cubasch et al., 2013; Weed et al., 2013). At the start of the 20th century, many of these forests were cleared for agricultural purposes, effectively releasing carbon into the atmosphere (Bonan, 2008; Richart and Hewitt, 2008). With the rise of industrial development and movement of agricultural practices to other regions, many of these agricultural lands were abandoned and subsequently

reforested through natural regrowth and afforestation practices (Canadell and Raupach, 2008). Currently, much of the forested area within the mixed-wood plains ecozone in the Great Lakes region of Canada and the USA is comprised of reforested or plantation stands which are in different stages of growth (Wiken et al., 2011).

Climate change and the associated changes in extreme weather events and the hydrologic cycle such as warmer spring temperatures, intense heat and drought events in the summer, early snowmelt, reduced snowfall, or increased freeze and thaw cycles in winter may impact the ability of these local forests to sequester carbon, and thus impact regional forest–atmosphere interactions (Bonan, 2008; Allen et al., 2010; Teskey et al., 2015). However, climate change will impact deciduous and coniferous forest ecosystems differently due to their physiological differences. Even in deciduous and coniferous forests of similar age, geographic location, climatic conditions, and soil properties, differences in the timing and rate of photosynthesis, ecosystem respiration, and evapotranspiration may lead to asymmetries in the overall forest productivity, water use, and hence longevity and survival. Consequently, regions once dominated by coniferous forests may yield to more deciduous species (Givnish, 2002; Bonan, 2008). Such a shift could disturb regional carbon and water budgets, as deciduous forests typically have shorter growing seasons and higher photosynthetic rates and water use efficiencies when compared to coniferous forests (Givnish, 2002; Ciais et al., 2005). While many studies have examined the annual carbon and water fluxes within specific land use and forest types, to date, only a handful of studies have compared these fluxes among similar-age deciduous and coniferous forests growing in close proximity, in similar climatic and edaphic conditions (Gaumont-Guay et al., 2009; Baldocchi et al., 2010; Novick et al., 2015; Wagle et al., 2016). Even fewer studies have reported multi-annual time series.

This study examined the carbon and water fluxes in two eastern North American forest ecosystems of different tree species but similar age, climate, and edaphic conditions during a 6-year period from 2012 to 2017. One stand was an 80-year-old (as of 2019) evergreen needleleaf forest, while the other was a roughly 90-year-old broadleaf deciduous forest. The specific objectives of the study were to (1) examine seasonal and interannual dynamics of carbon and water exchanges in the two forests, (2) determine the impact of meteorological controls on overall forest productivities, and (3) analyze and contrast the varying responses of the two different forests to extreme meteorological events such as heat and drought.

## 2 Methods

### 2.1 Study sites

The two forests are located within 20 km of each other, situated on the north side of Lake Erie in Norfolk County, Ontario, Canada (Table 1). These forests are a part of the Turkey Point Observatory in association with the global FluxNet program. The landscape in the region is dominated by agricultural lands, while plantation and regenerated forests cover a small fraction (∼ 25 %) of the land cover; accounting for the highest forest cover in southeastern Ontario. The broadleaf deciduous forest (from here on abbreviated and referred to as Turkey Point Deciduous, TPD) was naturally regenerated in the early 1900s from abandoned agricultural land on sandy terrain. The forest is unevenly aged (70–110 years old) with a mean age of roughly 90 years. The stand is dominated by white oak (*Quercus Alba*), with secondary hardwood species including red maple (*Acer Rubrum*), sugar maple (*Acer Saccharum*), black oak (*Quercus Velutina*), red oak (*Quercus Rubra*), white ash (*Fraxinus Americana*), yellow birch (*Betula alleghaniensis*), and American beech (*Fagus Grandifolia*). Conifer species only account for a minor component (∼ 5 %) of the total tree population (Kula, 2014). The understory is made up of young deciduous trees as well as Canada mayflower (*Maianthemum canadense*), putty root (*Aplectrum hyemale*), yellow mandarin (*Disporum lanuginosum*), red trillium (*Trillium erectum*), and horsetail (*Equistum*). The stand has been managed in the past with the last commercial harvesting occurring in 1984 and 1986, during which 440 and 39.97 m$^3$ (wood volume) of wood were removed, respectively. The specific harvesting of white pine (*Pinus Strobus L.*; 106 m$^3$), red pine (*Pinus Resinosa*; 71.42 m$^3$), poplar (*Populus*; 48.22 m$^3$), and dead oak (61.35 m$^3$) also occurred from 1989 to 1994 (Long Point Region Conservation Authority records). Since 1994, no management activity has occurred in this stand.

The evergreen needleleaf conifer forest, referred to as Turkey Point 39 (TP39 from here on), was planted in 1939 on cleared oak-savanna lands. The dominant tree species in this approximately 80-year-old stand are eastern white pine and balsam fir (*Abies balsamifera L. Mill*), making up 82 % and 11 % of the total tree population, respectively. The remaining 7 % of trees are typical native eastern North American forest species, which includes white oak, black oak, red maple, wild black cherry (*Prunus serotina Ehrh.*), and white birch (*Betula papyrifera*). The understory consists of young white pines, oak, balsam fir, and black cherry trees, as well as other ground vegetation, including bracken fern (*Pteridium aquilinum*), blackberry (*Rubus spp.*), poison ivy (*Rhus radicans*), moss (*Polytrichum spp.*), and Canada mayflower. The conifer forest has also been managed on two occasions. A thinning was performed in 1983 in which 4044 m$^3$ of wood was removed from 38.6 ha land area CE1 (Ontario Ministry of Natural Resources and Forestry records). In the early winter

**Table 1.** Site characteristics of the deciduous (TPD) and coniferous (TP39) forest stands. The TP39 values in brackets indicate pre-thinning (2003–2011) values, prior to the period of focus.

| | Turkey Point 1939 (TP39) 42.71° N, 80.357° W | Turkey Point Deciduous (TPD) 42.635° N, 80.558° W |
|---|---|---|
| **Stand** | | |
| Previous land use | Afforested on oak savanna, cleared for afforestation | Naturally regenerated on abandoned agricultural land |
| Age (in 2017) | 78 years | 70–110 years |
| Elevation (m) | 184 | 265 |
| DBH (cm) | 39.0 (37.2) | 23.1 |
| Density (trees ha$^{-1}$) | 321 (413) | 504 |
| Tree height (m) | 23.4 (22.9) | 25.7 |
| LAI (m$^2$ m$^{-2}$) | 5.3 (8.5) | 8.0 |
| Dominant species | *Pinus strobus L.* | *Quercus Alba* |
| Secondary and understory | *Abies Balsamea, Q. Velutina, A. Rubrum, Prunus Serotina* | *Acer Saccharum, A. Rubrum, Fagus Grandifolia, Q. Velutina, Q. Rubra, Fraxinus Americana* |
| Ground | *M. Canadense, Rubus Spp., Rhus Radicans, Ferns, Mosses* | *Maianthemum Canadense, Aplectrum Hyemale, Equisetum* |
| **Soil** | | |
| Drainage | Well-drained | Rapid to well-drained |
| Classification | Brunisolic grey brown luvisol | Brunisolic grey brown luvisol |
| Texture | Very fine sandy-loam | Predominantly sandy |
| Bulk density (kg m$^{-3}$) | 1.35 g m$^{-3}$ | 1.15 g m$^{-3}$ |

of 2012, the stand was again thinned by harvesting one-third of the trees (2308 m$^3$), leading to a reduction in stand density (Table 1). A subsequent study conducted by our group found that while the 2012 thinning of the coniferous stand significantly reduced the annual net ecosystem productivity (NEP) when compared to the 9-year pre-thinning (2003–2011) mean annual NEP values, the post-thinning NEP was still within the range of interannual variability (Trant, 2014). Additionally, Skubel et al. (2017) reported that stand-level ET was not impacted by the 2012 thinning, as increased soil evaporation and understory transpiration resulted due to a more open forest canopy. Ultimately, the objectives of this study did not focus on examining the impacts of this disturbance.

While edaphic and climatic conditions are similar between both sites, they differ in vegetation cover and canopy structure and physiology. The soils in each stand are predominantly sandy (greater than 90 % sand), classified by the Canadian Soil Classification Scheme and FAO World Reference Base as Brunisolic Grey-Brown Luvisol and Albic Luvisol/Haplic Luvisol, respectively (Present and Acton, 1984;

Lavkulich and Arocena, 2011). These sandy soils are part of the Southern Norfolk Sand Plain, an area shaped by past ice age glacial melt processes (Richart and Hewitt, 2008). Soils at both sites are well-drained with a low-to-moderate water holding capacity (McLaren et al., 2008). Further soil and site details can be found in Arain and Restrepo-Coupe (2005), Peichl et al. (2010), and Beamesderfer et al. (2020). The climate of the region is humid continental with warm, humid summers and cool winters. The 30-year (from 1981 to 2010) mean annual air temperature and total precipitation measured at the Environment Canada Delhi CDA weather station (25 km north of the sites) is $8.0 \pm 1.6$ °C and 997 mm, respectively. Total precipitation is normally evenly distributed throughout the year, with 13 % of that falling as snow (Environment and Climate Change Canada, 2019).

## 2.2 Eddy covariance and meteorological measurements

Half-hourly fluxes of water vapor and $CO_2$ ($F_c$) have been measured continuously at TP39 and TPD using closed-path eddy covariance (EC) systems since 2003 and 2012, respec-

tively. This study examines the first 6 years (2012 to 2017) of data at the deciduous forest and the corresponding period for the conifer forest. Measurements at both sites are still ongoing. The closed-path EC systems at each site consist of a 3D sonic anemometer (CSAT3, Campbell Scientific Inc.) and an infrared gas analyzer (IRGA), an LI-7000 (LI-COR Inc.) at TP39, and an LI-7200 (LI-COR Inc.) at TPD. The specific details of the EC systems are outlined in the Appendix (Table A1). At both sites, IRGAs are calibrated monthly using high-purity $N_2$ gas for the zero offset and $CO_2$ gas (360 µmol mol$^{-1}$ $CO_2$; following WMO standards) for the $CO_2$ check.

The $CO_2$ storage ($S_{CO_2}$) in the air column below the EC system is calculated by vertically integrating the half-hourly difference in $CO_2$ concentrations. This calculation is completed for both the canopy and mid-canopy gas analyzers (Table A1). Half-hourly net ecosystem exchange (NEE, µmol m$^{-2}$ s$^{-1}$) is calculated as the sum of the vertical $CO_2$ flux ($F_c$) and the rate of $CO_2$ storage ($S_{CO_2}$) change in the air column below the IRGA (NEE $= F_c + S_{CO_2}$). Horizontal and vertical advection values are assumed to average to zero over long periods and were not considered. Half-hourly net ecosystem productivity (NEP) is calculated as the opposite of NEE (NEP $= -$NEE), where positive NEP ($-$NEE) indicates net carbon uptake by the forest (sink), and negative NEP ($+$NEE) is carbon loss from the forest to the atmosphere (source).

Meteorological measurements have been conducted alongside EC measurements during the entire measurement period at both sites. Air temperature ($T_a$), relative humidity (RH), wind speed and direction, downward and upward photosynthetically active radiation (PAR), and the four-components of radiation (Rn) are measured at the specified EC sampling heights for both sites (Table A1). Soil temperature ($T_s$) and soil water content ($\theta$) are measured at 2, 5, 10, 20, 50, and 100 cm depths in two soil pit locations at both sites. At TPD, precipitation ($P$) is measured in a small forest opening, 350 m southwest of the tower. However, this analysis used $P$ data from an accumulation rain gauge (T-200B, GEONOR) installed 1 km south of TP39. All meteorological, soil, and $P$ data were recorded using data loggers with automated data downloads occurring every half-hour on desktop computers located at the base of the scaffold walk-up towers located at each site.

Following an AmeriFlux visit to TP39 for an instrument and data comparison (in 2019; data were processed after this study took place CE2), the downward PAR sensor at that site was found to be identical to the AmeriFlux measurements. Consequently, downward PAR at TPD was thus underestimating (likely due to sensor differences, i.e., PAR-Lite vs. PQSI, and their coefficients) actual PAR values. A correction factor of 1.22 (slope between the two sites) was applied to daily mean PAR data at TPD for each year.

## 2.3 Meteorological and eddy covariance data processing

All meteorological and flux data were processed on lab-developed software following the FluxNet Canada Research Network (FCRN) guidelines as described by Brodeur (2014). Meteorological variables were sampled at 5 s intervals and averaged at a half-hourly scale. A two-step cleaning process was used to remove outliers in half-hourly meteorological data: coarse upper and lower thresholds were applied to half-hourly values to remove obvious outliers, and additional erroneous half-hourly data were removed from time series when instruments were known to be malfunctioning or visual inspection by multiple reviewers resulted in certain agreement that an outlier was present. Missing meteorological data of all lengths were gap-filled using extant data for the same half-hours from either (in order of preference) a second sensor at the site or an equivalent sensor from a nearby (1–3 km away) station in the network (sites described in Peichl et al., 2010). This approach was supported by a very high correlation between variables ($R^2 > 0.96$). Linear regressions between variables from different sources were used to correct for any offset and gain discrepancies.

The same two-step cleaning process was also used to remove outliers from the flux data. For eddy-covariance-derived fluxes, the spike detection method described in Papale et al. (2006) was subsequently applied. After these quality control measures were applied, the mean flux data coverage was 91 % (from 83 % to 94 %) at TPD and 88 % (from 79 % to 94 %) at TP39 over the 6 years of data collection. Each time series was then subjected to a footprint filtering process, in which a footprint model (Kljun et al., 2004) was applied to exclude fluxes when greater than 10 % of the flux footprint extended outside of the defined forest boundary (Brodeur, 2014). This process removed approximately 32 % of half-hourly fluxes from TPD and 16 % from TP39. Finally, nighttime (PAR $< 100$ µ mol m$^{-2}$ s$^{-1}$) fluxes were subjected to friction velocity ($u^*$) filtering to remove half-hours where low turbulence may lead to underestimations by EC systems. The moving-point test determination method (Reichstein et al., 2005; Papale et al., 2006; Barr et al., 2013) was used to estimate annual $u^*$ threshold ($u^*$Th) values at each site, and the nighttime half-hourly flux data were removed when the measured friction velocity ($u^*$) was below the calculated threshold ($u^*$Th). The mean site-specific $u^*$Th values were 0.40 m s$^{-1}$ at TPD and 0.49 m s$^{-1}$ at TP39. The resulting final mean flux data recovery following both threshold filtering methods was 49 % (from 46 % to 53 %) at TPD and 53 % (from 48 % to 57 %) at TP39 for the 6 years of measurements. Confidence intervals (95 %) incorporating the effect of random instrument error as well as systematic and random errors associated with the gap-filling process used for annual NEE estimates were calculated using a functional relationship with an annual gap percentage, developed for these sites by Brodeur (2014). The NEE model uncertainty ranged

from $\pm 33$–$37\,\mathrm{g\,C\,m^2\,yr^{-1}}$ at TPD and $\pm 31$–$36\,\mathrm{g\,C\,m^2\,yr^{-1}}$ at TP39. Furthermore, uncertainties in annual ET values totaling the sum of both measurement uncertainties and data gap-filling (as described by Arain et al. 2003) were estimated to be $\pm 35$–$43\,\mathrm{mm\,yr^{-1}}$ at TPD and $\pm 41$–$50\,\mathrm{mm\,yr^{-1}}$ at TP39.

NEE gap-filling and its partitioning into components of ecosystem respiration (RE) and gross ecosystem productivity (GEP) were achieved using the methods described in Peichl et al. (2010), which are summarized below. RE was assumed to be equivalent to NEE during nighttime periods (PAR $< 100\,\mathrm{\mu mol\,m^{-2}\,s^{-1}}$) that passed both footprint and friction velocity filters. These values were used to model a continuous RE time series based on a non-linear regression with $T_{\mathrm{s5\,cm}}$ and $\theta_{0\text{--}30\,\mathrm{cm}}$ (depth-weighted average from measurements made at 5, 10, 20, and 50 cm depths) using the functional form:

$$\mathrm{RE} = R_{10} \times Q_{10}^{\frac{(T_{\mathrm{s5\,cm}}-10)}{10}} \times \frac{1}{\left[1 + \exp\left(a_1 - a_2\theta_{0\text{--}30\,\mathrm{cm}}\right)\right]}, \quad (1)$$

where parameters $R_{10}$ and $Q_{10}$ define controls of $T_{\mathrm{s5\,cm}}$ on RE. The $\theta_{0\text{--}30\,\mathrm{cm}}$ related controls are defined as follows:

$$f(\theta_{0\text{--}30\,\mathrm{cm}}) = \frac{1}{\left[1 + \exp\left(a_1 - a_2\theta_{0\text{--}30\,\mathrm{cm}}\right)\right]}, \quad (2)$$

where $a_1$ and $a_2$ are fitted parameters that describe a sigmoidal curve that ranges from 0 to 1 (Richardson et al., 2007). In this approach, the $T_{\mathrm{s5\,cm}}$ component of the function defines a theoretical maximum half-hourly respiration rate based on soil temperature (i.e., driving variable), while the $\theta_{0\text{--}30\,\mathrm{cm}}$ component modulates the resultant predicted value as a function of the volumetric water content (i.e., scaling variable). Parameter values for Eq. (1) were derived for each site and year; values were estimated simultaneously using the Nelder–Mead simplex optimization approach via the MATLAB fminsearch function (The MathWorks, Inc). The estimated parameters were then used to model RE for all half-hour periods using the measured values of $T_{\mathrm{s5\,cm}}$ and $\theta_{0\text{--}30\,\mathrm{cm}}$.

Half-hourly GEP was derived as the difference between modeled daytime RE and footprint-filtered NEE. Gaps in the GEP time series were filled using predicted values derived from the following relationship:

$$\mathrm{GEP} = \frac{\alpha \mathrm{PAR} A_{\max}}{\alpha \mathrm{PAn} + A_{\max}} \times f\left(T_{\mathrm{s5\,cm}}\right) \times f(\mathrm{VPD})$$
$$\times f\left(\theta_{0\text{--}30\,\mathrm{cm}}\right), \quad (3)$$

where the first term is a rectangular hyperbolic functional relationship between PAR and GEP, defined by the values of the photosynthetic flux per quanta ($\alpha$, quantum yield) and the light-saturated rate of $CO_2$ fixation ($A_{\max}$). The remaining terms use the functional form introduced in Eq. (2) to described the responses of GEP to $T_{\mathrm{s5\,cm}}$, vapor pressure deficit

(VPD), and $\theta_{0\text{--}30\,\mathrm{cm}}$, respectively. Parameters were optimized using the same approach described above. Finally, gaps in the NEP time series were filled using the differences between the filled GEP and modeled RE time series.

Following the aforementioned threshold and point cleaning, gaps in the latent heat flux (LE), and therefore the mass equivalent evapotranspiration, were filled using an artificial neural network which utilized net radiation (Rn), wind speed, $T_{\mathrm{s5\,cm}}$, VPD, and $\theta_{0\text{--}30\,\mathrm{cm}}$ (Brodeur, 2014). Following the approach outlined by Amiro et al. (2006), any remaining gaps in LE data were filled using a moving-window linear regression method. Past studies examining the relationships between ET and meteorological variables for the forests of the Turkey Point Observatory have found $T_{\mathrm{a}}$ to largely drive ET, with smaller secondary effects driven by VPD and $\theta_{0\text{--}30\,\mathrm{cm}}$ during low-water or high-heat periods (McLaren et al., 2008; MacKay et al., 2012; Skubel et al., 2015; Burns, 2017). All data processing and analyses were completed using MATLAB software (The MathWorks, Inc.).

## 2.4 Estimating effects of meteorological variables on carbon component fluxes

The partitioning models described above were further used to explore interannual differences in controlling meteorological variables and their impacts on annual RE and GEP values at each site. In this analysis, the RE and GEP models (see Eqs. 1 and 3 above, respectively) were parameterized for the phenologically derived summer months (end of greenup to the start of browndown, defined in the next section) for all years (2012 to 2017). To overcome issues of equifinality that arose when fitting the parameters of Eq. (1) and (3) to each year of data, parameterization was performed as a two-step process, in which parameters describing "scaling" variable relationships (i.e., $\theta_{0\text{--}30\,\mathrm{cm}}$ for RE; $T_{\mathrm{a}}$, $\theta_{0\text{--}30\,\mathrm{cm}}$, VPD for GEP) were fixed to all years of data, while relationships with "driving" variables (i.e., $T_{\mathrm{s5\,cm}}$ for RE; PAR for GEP) were parameterized to each year of data with other parameters fixed. Furthermore, the mean annual value for each scaling variable function was used to compare the quality of meteorological conditions between years. Given that these variables scale between 0 and 1, higher annual values (i.e., closer to 1) indicated that the variable was relatively more favorable for RE or GEP production in that given year. To present this in true relative terms, the annual values for a given functional relationship were normalized by the highest annual value. Thus, reported annual values represent a proportion of the most favorable year. A similar metric was derived for the driving variables by modeling RE and GEP using the driving relationships only (i.e., no scaling variables). Modeled annual values were normalized by the highest one, thus creating a relative annual score like that for scaling variables. Finally, all metrics derived for scaling and driving variables in a given year were multiplied together to provide a measure

of the cumulative effect of all meteorological variables to a given component flux in a given year.

## 2.5 Definitions of key climatic and plant physiological variables

In this study, we define the term drought similarly to Wolf et al. (2013), where drought periods are related to deficits in precipitation, which impose either plant physiological stress due to decreased soil moisture ($\theta$) or impose stress due to stomatal closures in response to high VPD.

Two resource efficiencies were calculated at both forests to compare the links between productivity and resource supply in order to reveal differences in their responses to changing climatic conditions. The amount of carbon fixed through photosynthesis per unit of absorbed solar radiation, described as the photosynthetic light use efficiency (LUE) was calculated as follows:

$$\text{LUE} = \frac{\text{GEP}}{\text{APAR}}, \tag{4}$$

where GEP is equivalent to the carbon fixed through photosynthesis, and APAR is the portion of PAR that is absorbed (Jenkins et al., 2007; Liu et al., 2019). The forest canopy radiation budget used in the calculation of APAR is described as follows:

$$\text{APAR} = \text{PARdn} - \text{PARup} - \text{PARground}, \tag{5}$$

where PARdn is the incident PAR measured by PAR sensors mounted at the top of each tower facing skyward, and PARup is measured as reflected PAR by instruments mounted at the same height as the PARdn sensor, but facing downward towards the forest canopy. PARground is the PAR transmitted through the canopy to a ground sensor located at 2 m height. Furthermore, the forest-level water-use efficiency (WUE), describing the carbon fixed through photosynthesis per water lost, was calculated as the ratio of GEP to ET (Keenan et al., 2013).

Using the methods of Gonsamo et al. (2013), we calculated phenologically derived seasons for each year for each site. From half-hourly non-gap-filled data (calculated as the difference between modeled RE and measured non-gap-filled NEE), the maximum daily photosynthetic uptake (GEP$_\text{max}$) was calculated and fit using a double logistic function described by Gonsamo et al. (2013). From the initial fit, a Grubb's test was conducted to statistically ($p < 0.01$) remove outliers in GEP$_\text{max}$ data using the approach outlined by Gu et al. (2009). With outliers removed, the function was fit once more. This approach identified photosynthetic tranition dates, hereafter described as phenological dates, using first, second, and third derivatives of the logistic curves. The local minima of the second derivatives estimated the end of greenup (EOG), the length of canopy closure (LOCC), and the start of browndown (SOB), while the local maxima of

the third derivatives estimated the start of the growing season (SOS) and the end of the growing season (EOS). The start of the growing season (SOS) marked the end of winter dormancy and the beginning of the spring season, leaf emergence/greenup. The phenologically defined spring season is defined as the period from SOS to EOG. The phenologically defined summer or peak carbon uptake period is defined as the entire LOCC period from the final day of greenup (EOG) to the initiation of leaf senescence (SOB), bound by spring and autumn shoulder seasons. Finally, the resulting phenologically defined autumn season is from SOB to EOS, with EOS marking leaf abscission and the end of photosynthetic activity in autumn. The length of the overall growing season (LOS) was calculated as the number of days between SOS and EOS.

Lastly, the impact of climate on phenology was examined by the use of growing degree days (GDDs) and cooling degree days (CDDs), in order to understand the thermal response of each forest. GDD accumulation was defined as occurring when the mean daily $T_\text{a}$ was greater than $0\,^\circ\text{C}$, while CDD were calculated using the daily mean $T_\text{a}$ below a base $T_\text{a}$ of $20\,^\circ\text{C}$ (Richardson et al., 2006). Cumulative GDD and CDD were briefly considered in this analysis.

## 3 Results

### 3.1 Meteorological variability

Air temperature measurements conducted above the canopies at both sites showed that the daily mean values of $T_\text{a}$ at TP39 (Fig. 1a) and TPD (Fig. 1b) behaved similarly (Fig. 1c) over the study period. All years experienced annual mean $T_\text{a}$ greater than the 30-year mean ($8.0\,^\circ\text{C}$). Record high $T_\text{a}$ conditions (exceeding 30-year mean daily maximum values) were measured throughout the majority of the year in 2012 and during the summer of 2016. Cooler conditions dominated 2013 and 2014, while these years had a higher magnitude of extreme cold days in winter (exceeding 30-year mean daily minimum values), acting to decrease mean annual $T_\text{a}$. In 2015, 2016, and 2017, autumn warming was observed, with record $T_\text{a}$ outside of the typical summer (June–August) period. Overall, daily mean $T_\text{a}$ at both sites was almost identical (Fig. 1c), highlighting the similar climate both forests were growing in during the study period.

Meteorological conditions at both sites were examined over the study period. At TP39, APAR exhibited a similar parabolic curve each year due to the seasonal amplitude in PARdn and the continuous presence of an apparently dense coniferous canopy promoting a nearly constant fraction (fPAR) of PARdn being absorbed (Fig. 2a). At TPD, APAR exhibited lower values in the winter seasons when the forest remained leafless. The timing of the peak APAR at TPD was similar to TP39, though it varied each year based on the annual timing of leaf-out and spring canopy development.

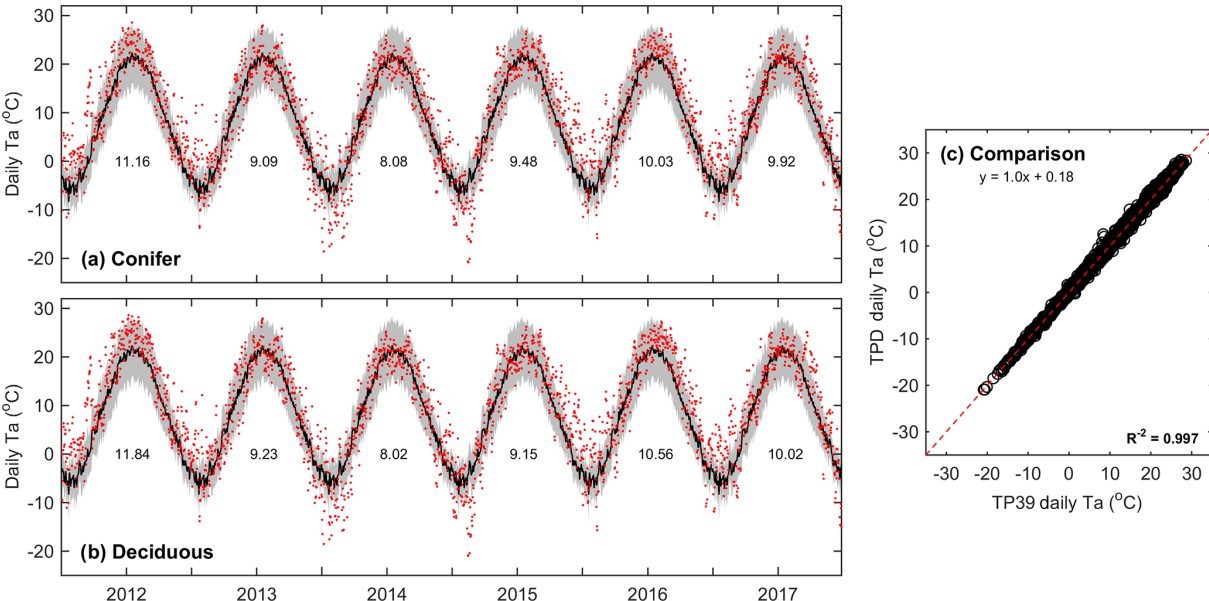

**Figure 1.** Daily above-canopy air temperature ($T_a$, red dots) measured from 2012 to 2017 at the **(a)** conifer forest (TP39) and **(b)** deciduous forest (TPD), with the grey shading and black line corresponding to the 30-year Environment Canada (Delhi station) minimum and maximum range of daily $T_a$ and mean daily $T_a$, respectively. Values shown represent the annual mean $T_a$ for each year of measurements. Also included is the **(c)** comparison of daily $T_a$ at TP39 and TPD.

Daily reductions in PARdn and APAR often resulted from cloudy conditions and precipitation ($P$) events (Fig. 2a).

Fewer $P$ events were measured during the first half of 2012 and most of 2015, 2016, and the late summer of 2017, as the latter three years had annual $P$ less than the 30-year mean (997 mm). Autumn $P$ in 2012 helped the forests to recover from the record heat and water deficits, while 2013 and 2014 experienced consistent rain throughout much of the year. Heightened daily VPD (Fig. 2b) was experienced throughout 2012 by both sites, with seasonal maximum values measured during warm and dry conditions. In all years, except for 2012 and the autumn of 2016, daily VPD at TP39 was higher than at TPD (Fig. 2c). Annually, mean VPD was on average about 0.04 kPa higher at TP39 than TPD, with 2012 being the obvious exception (Fig. 2c).

$T_s$ at 5 cm soil depth followed $T_a$ closely (Fig. 1) with dampening effects evident at deeper (100 cm) soil layers (Fig. 2d). The differences in $T_{s5\,cm}$ were explained by the species compositions of the two forests (Fig. 2e). At TPD, when the deciduous forest was leafless in winter and spring, $T_{s5\,cm}$ was higher than at TP39 as the soil received more direct radiation. However, during the summer and autumn of each year, $T_{s5\,cm}$ at TP39 exceeded that of at TPD due to differences in canopy cover. Lastly, $\theta$ from 0–30 cm ($\theta_{0-30\,cm}$) followed similar patterns at both sites, with prolonged summer $\theta$ declines in 2012, 2016, and 2017 (Fig. 2f). The magnitudes were again different, but each forest experienced similar declining $\theta$ and the subsequent recharging $\theta$ analogous to local $P$ events. In the summer, $\theta$ was typically lower at TPD than

TP39, while during all other times of the year $\theta$ at TP39 was higher (Fig. 2g).

## 3.2 Phenological variability

The meteorological conditions had a significant impact on the timing and duration of key phenological events, although ultimately the response was governed by different leaf strategies of the various dominant tree species in each forest. The phenological transition dates and seasons calculated from EC flux data are shown in Table 2 and Fig. 3. The SOS varied considerably between the two forests, with the SOS at the evergreen forest, TP39, beginning on average $38 \pm 14$ d earlier than at the deciduous forest, TPD. TP39 experienced a larger variation in SOS dates, spanning a period of 26 d between the earliest (10 March 2012; day of year, DOY, 70) and latest (6 April 2014; DOY 96) years, while TPD varied by 11 d between years.

Growing degree days are a proxy used to assess the amount of heat the ecosystem has absorbed, as a result of increasing air temperatures. The response of the forest to increasing GDD was shown to be a trigger for the SOS. The total GDD from the start of the year (1 January, DOY 1) to 6-year mean day of season growth (25 March; DOY 84) was found to be highly correlated to SOS at TP39 ($R^2 = 0.81$), but not at TPD (Fig. 4a, b). GDD for DOY 117–127 (27 April to 7 May, which represents the range of the 6-year mean SOS date $\pm 1$ standard deviation) was found to significantly influence the SOS at TPD ($R^2 = 0.95$), with a weaker influence at TP39 (DOY 73–95; $R^2 = 0.76$). This difference

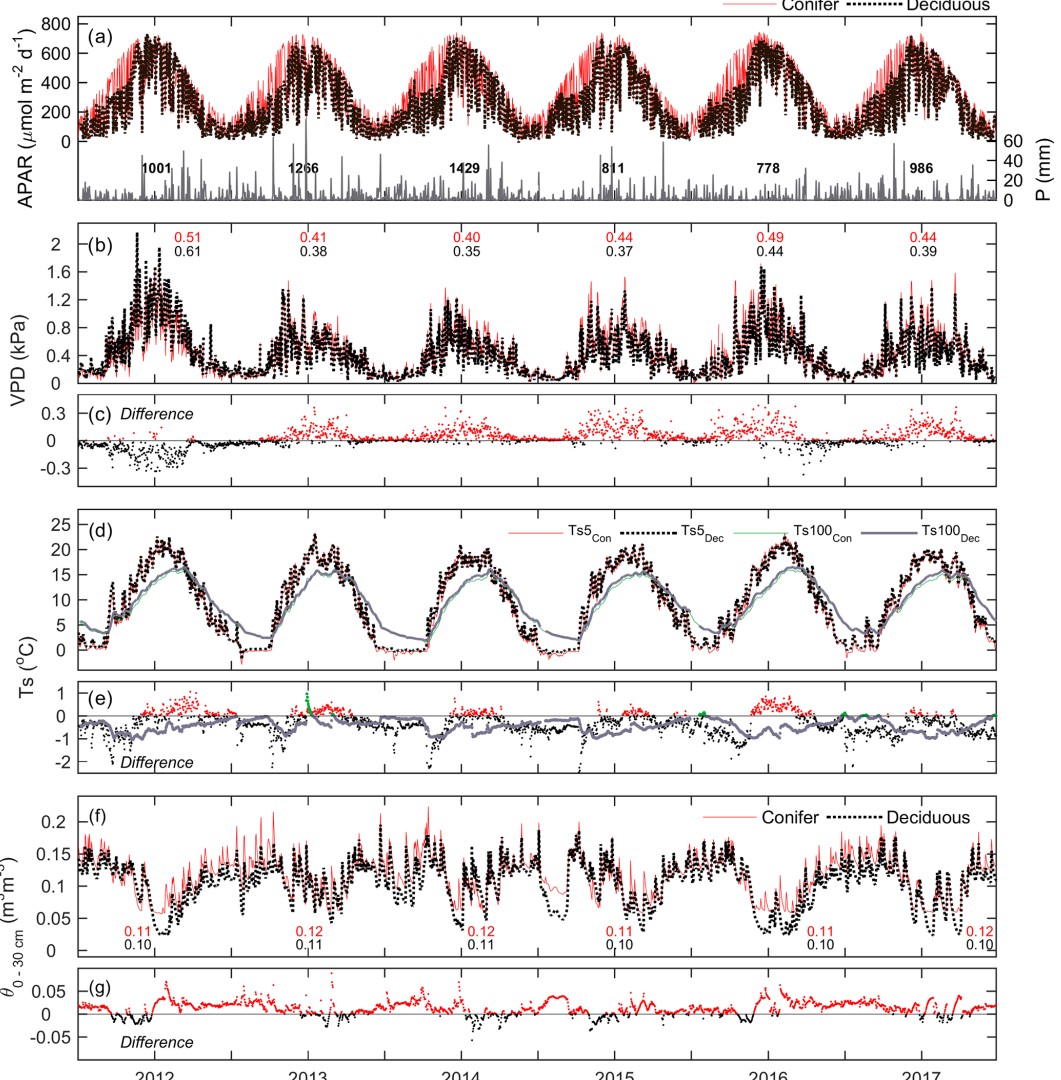

**Figure 2.** Time series of daily mean meteorological variables measured at the conifer (TP39, red line) and deciduous (TPD, black dashed line) forests from 2012 to 2017, including (**a**, left) absorbed photosynthetically active radiation (APAR), (**a**, right) total precipitation ($P$), (**b**) vapor pressure deficit (VPD), (**c**) the difference in VPD between the two forests (conifer – deciduous), (**d**) soil temperatures ($T_s$) at 5 and 100 cm depths, (**e**) the difference in $T_s$ between the two forests at both depths, (**f**) soil volumetric water content from 0–30 cm depths ($\theta_{0-30\,cm}$), and (**g**) the difference in $\theta$ between the two forests.

likely reflects the different leaf strategies, in that evergreen trees are ready to start photosynthesizing as soon as conditions are favorable, while the deciduous trees still need to grow their leaves once conditions are favorable, before comparable rates of photosynthesis are reached. Spring, defined as the period from SOS to EOG, was more than double the length ($69 \pm 14$ d) at TP39 when compared to TPD ($31 \pm 5$ d). However, even with largely different SOS and spring lengths, the peak summer period, defined as the period between EOG in spring and SOB in autumn, was essentially identical between the forests (Fig. 3). This period, spanning June, July, and August, was found to be a key contributor to the net annual productivity of each forest (discussed below).

With similar peak summer lengths, the forests began senescence at similar times, though the length of autumn, the period from the SOB to the EOS, varied considerably between the forests, due to differences in the timing of the EOS (Fig. 3). Drought conditions in the summer of 2012 led both sites to have the shortest autumns and earliest EOS (Figs. 2f and 3). Conversely, late season warming in the autumns of 2016 and 2017 helped to prolong the growing season at both sites, but the impacts of late season warming in 2015 were not as evident in shaping the timing of EOS (Figs. 1 and 3; Table 2).

At both sites, the cumulative CDDs from DOY 230 to 290 (mid-August to mid-October; loosely based on the range of

**Table 2.** The top section of the table contains the annual calculated phenological dates (reported as day of year) for both the conifer (TP39, bold) and deciduous (TPD, italicized) forests from the year 2012 to the year 2017. Phenological dates were calculated following Gonsamo et al. (2013) from eddy-covariance-measured GEP$_{max}$ data. The 6-year mean values and standard deviations are included in the final column. The resulting phenological periods (seasons) and their duration in days are also shown, in the lower section of the table.

| Phenology transition dates | 2012 | 2013 | 2014 | 2015 | 2016 | 2017 | Mean |
|---|---|---|---|---|---|---|---|
| Start of season | **70** | **96** | **96** | **91** | **74** | **79** | **84 ± 12** |
| (SOS, bud-break) | *120* | *116* | *127* | *118* | *126* | *125* | *122 ± 5* |
| Middle of greenup | **119** | **137** | **132** | **122** | **127** | **130** | **128 ± 7** |
| (MOG, fastest greenup) | *136* | *141* | *148* | *136* | *144* | *147* | *142 ± 5* |
| End of Greenup | **147** | **160** | **153** | **140** | **158** | **159** | **153 ± 8** |
| (EOG, end of leaf-out) | *145* | *155* | *160* | *146* | *154* | *159* | *153 ± 6* |
| Peak of season | **214** | **205** | **202** | **193** | **212** | **201** | **204 ± 8** |
| (midpoint between EOG and SOB) | *198* | *199* | *205* | *193* | *203* | *207* | *201 ± 5* |
| Start of browndown | **271** | **258** | **258** | **257** | **270** | **248** | **260 ± 9** |
| (SOB, start of senescence) | *257* | *249* | *255* | *249* | *262* | *261* | *255 ± 6* |
| Mid of browndown | **287** | **292** | **287** | **289** | **305** | **287** | **291 ± 7** |
| (MOB, fastest senescence) | *275* | *273* | *274* | *271* | *286* | *282* | *277 ± 6* |
| End of Season (EOS) | **314** | **351** | **338** | **345** | **366** | **354** | **345 ± 17** |
| | *306* | *314* | *307* | *309* | *328* | *318* | *314 ± 8* |
| Phenologically defined seasons | 2012 | 2013 | 2014 | 2015 | 2016 | 2017 | Mean |
| Spring | **78** | **64** | **58** | **48** | **84** | **80** | **69 ± 14** |
| (EOG – SOS) | *25* | *39* | *34* | *28* | *28* | *34* | *31 ± 5* |
| Summer (SOB – EOG) | **124** | **97** | **105** | **117** | **112** | **89** | **107 ± 13** |
| (LOCC, Length of Canopy Closure) | *112* | *94* | *95* | *103* | *107* | *102* | *102 ± 7* |
| Autumn (EOS – SOB) | **43** | **94** | **80** | **89** | **96** | **106** | **85 ± 22** |
| | *49* | *65* | *52* | *61* | *67* | *57* | *58 ± 7* |
| Length of growing season (LOS) | **245** | **255** | **242** | **254** | **292** | **275** | **260 ± 19** |
| | *186* | *198* | *180* | *191* | *202* | *193* | *192 ± 8* |

dates in Oishi et al., 2018) were highly correlated to the EOS at TP39 ($R^2 = 0.84$) and TPD ($R^2 = 0.95$) (Fig. 4e and f). Temperature responses in both the spring (i.e., GDD) and autumn (i.e., CDD) were much higher for TPD than TP39 (Fig. 4). These results suggest that warmer winter and early spring (i.e., January to April) conditions will lead to an advancement of the SOS in the conifer forest, but the same cannot be said for the deciduous forest, whose SOS dates were heavily dependent on late-April–early-May growing conditions. To a certain degree, both forests responded similarly in autumn; however, physiological constraints of the different tree leaf strategies defined the overall differences in growing season lengths.

## 3.3 Carbon and water fluxes

The water (evapotranspiration) and carbon (photosynthesis and respiration) fluxes were analyzed in both forests from 2012 to 2017, with the seasonal patterns of these fluxes illustrated in Fig. 3 and cumulative fluxes in Table 3. Seasonal and total fluxes provide insight into each stand's ability to sequester carbon and release water over interannually comparable timescales. Annual photosynthesis (GEP) at the conifer forest (TP39) was the highest in 2017 (1709 g C m$^{-2}$ yr$^{-1}$) and 2015 (1701 g C m$^{-2}$ yr$^{-1}$), while the lowest annual GEP was measured in 2012 (1452 g C m$^{-2}$ yr$^{-1}$) and 2013 (1501 g C m$^{-2}$ yr$^{-1}$). GEP reductions during these years were due to opposing influences, with 2012 experiencing heat and drought conditions for most of the year and 2013 experiencing cooler $T_a$ and the highest annual $P$ (1266 mm), reducing PAR and therefore GEP (Fig. 3a). At the deciduous forest (TPD), similar GEP reductions were captured in 2012 (1198 g C m$^{-2}$ yr$^{-1}$) but not in 2013 (1369 g C m$^{-2}$ yr$^{-1}$), due to high photosynthetic gains outside of the 2013 peak growing season (i.e., in the early spring and autumn periods). The highest annual GEP at TPD was found in 2016 (1420 g C m$^{-2}$ yr$^{-1}$) and 2017 (1447 g C m$^{-2}$ yr$^{-1}$) due to warm summer conditions (Fig. 3b). Although 2014 had one of the shortest sum-

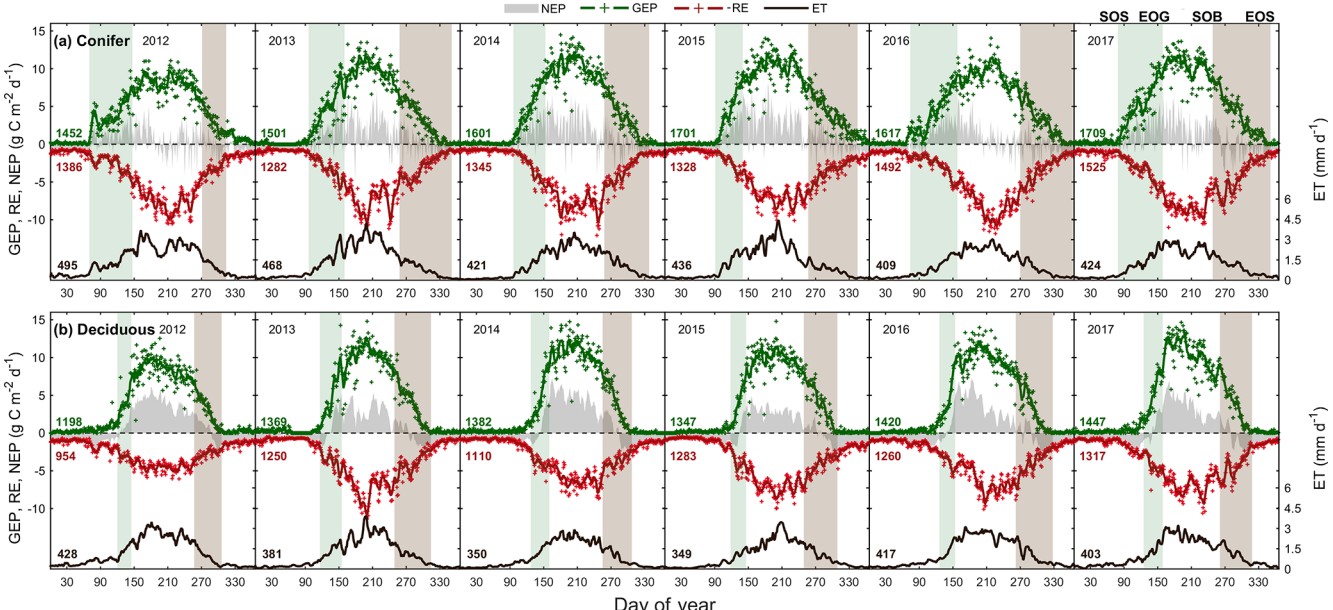

**Figure 3.** Time series from 2012 to 2017 of the daily total gross ecosystem productivity (GEP, green +), ecosystem respiration (RE, red +), net ecosystem productivity (NEP, grey shading), and evapotranspiration (ET, black, right) for the **(a)** conifer forest (TP39), and the **(b)** deciduous forest (TPD). Solid lines of GEP, RE, NEP, and ET are derived from 5 d moving averages of the measured data, while the colored values for each year correspond to annual GEP (green), RE (red), and ET (black) for each site. The annual EC-derived phenological spring (green shading) and autumn (brown shading) are included for each site, and can be found in Table 2.

mers and the shortest overall growing season length of all years, high daily GEP rates were sustained through the summer, resulting in the year having above-average annual GEP ($1382 \, \mathrm{g \, C \, m^2 \, yr^{-1}}$). In all 6 years, spring was the only season in which daily rates of GEP were similar between the forests, as the advancement of SOS at TP39 did not statistically benefit carbon uptake due to seasonal meteorological conditions (i.e., low PAR, $T_a$, etc.) acting to limit photosynthesis. Using the analysis of variance (ANOVA) technique, $t$ tests were completed to evaluate statistical differences between the two groups (i.e., deciduous broadleaf vs. evergreen needleleaf) of data. Summer and autumn daily GEP values were higher at TPD when compared to TP39 across the 6 years ($p < 0.01$). In all years, TP39 annual GEP was greater than TPD due to the longer growing seasons.

TP39 RE was highly variable in all years (Fig. 3a), with the greatest annual total RE values found in 2016 ($1492 \, \mathrm{g \, C \, m^2 \, yr^{-1}}$) and 2017 ($1525 \, \mathrm{g \, C \, m^2 \, yr^{-1}}$). The annual RE during these years was about 100 to $200 \, \mathrm{g \, C \, m^2 \, yr^{-1}}$ greater than during the other years. Cooler spring $T_a$ and reductions in RE during the summer of 2013 led to the lowest annual RE ($1282 \, \mathrm{g \, C \, m^2 \, yr^{-1}}$) of the 6 years. While 2012 encountered reduced ET and GEP during the summer, RE was largely unaffected, leading to the third highest annual RE ($1386 \, \mathrm{g \, C \, m^2 \, yr^{-1}}$). Conversely, the 2012 RE within the deciduous forest was greatly reduced, leading to an apparent outlier (exceeding mean and standard deviation) in annual RE at that site ($954 \, \mathrm{g \, C \, m^2 \, yr^{-1}}$). Similar to TP39, but to a

lesser degree, the annual RE at TPD during 2017 was the greatest of the 6 years ($1317 \, \mathrm{g \, C \, m^2 \, yr^{-1}}$). Annually, the RE at both forests behaved similarly, with 2012 being the exception (Fig. 3b). The highest daily rates of RE at both sites were measured during the summer of 2013, coinciding with similar maximums in ET. In both cases, maximum rates of RE and ET occurred following $P$ events, as the soil was sufficiently wet, helping to promote ET and enhance RE through respiration pulses (suggested in Misson et al., 2006). Daily summer RE was higher at TP39 in all years, with 2013 and 2015 being the exceptions.

The resulting balance between GEP and RE, net ecosystem productivity (NEP), was found to be largely irregular between sites during individual years due to site-specific differences in the timing, magnitude, and duration of daily fluctuations in GEP and RE. The trajectory of growing season NEP was strikingly different between sites (Fig. 3a and b). TPD (deciduous) captured consistently positive daily NEP (sink), while TP39 (conifer) was highly variable, with negative daily NEP (source) often occurring throughout the growing season. The annual NEP in the conifer forest was the lowest in 2012 ($76 \, \mathrm{g \, C \, m^2 \, yr^{-1}}$) and 2016 ($139 \, \mathrm{g \, C \, m^2 \, yr^{-1}}$), coinciding with heat and drought stress in both years (Fig. 5a). At TP39, July 2012 was the only month during the 6 years of measurements in which the peak summer growing season monthly NEP for either site was negative (Fig. 5a inset). The most productive years (largest annual sink) at the conifer site were 2015 ($395 \, \mathrm{g \, C \, m^2 \, yr^{-1}}$) and 2014

**Table 3.** Seasonal and annual sums of eddy-covariance-measured carbon (GEP, RE, and NEP, g C m$^{-2}$ yr$^{-1}$) and water fluxes (ET, mm yr$^{-1}$) from 2012 to 2017 for both the conifer (TP39, bolded C) and deciduous (TPD, italicized D) forests. The phenologically defined seasonal dates were calculated using the timing of transitions in phenological dates, outlined in Table 2. The 6-year mean and standard deviations are also included for each row.

| | Season | | 2012 | 2013 | 2014 | 2015 | 2016 | 2017 | Mean |
|---|---|---|---|---|---|---|---|---|---|
| GEP sum | 1 January to SOS | | – | – | – | – | – | – | – |
| | Spring (SOS to EOG) | C | **308** | **306** | **279** | **213** | **359** | **418** | **314 ± 70** |
| | | D | *104* | *197* | *165* | *117* | *129* | *174* | *148 ± 36* |
| | Summer (EOG to SOB) | C | **990** | **942** | **1070** | **1160** | **1014** | **930** | **1018 ± 86** |
| | | D | *942* | *949* | *1023* | *1006* | *1084* | *1070* | *1012 ± 59* |
| | Autumn (SOB to EOS) | C | **132** | **264** | **265** | **340** | **249** | **377** | **271 ± 85** |
| | | D | *147* | *239* | *200* | *240* | *219* | *213* | *210 ± 34* |
| | EOS to 31 December | | – | – | – | – | – | – | – |
| | Annual | C | **1452** | **1501** | **1601** | **1701** | **1617** | **1709** | **1597 ± 104** |
| | | D | *1198* | *1369* | *1382* | *1347* | *1420* | *1447* | *1360 ± 87* |
| RE sum | 1 January to SOS | C | **66** | **83** | **78** | **79** | **82** | **81** | **78 ± 6** |
| | | D | *167* | *107* | *129* | *109* | *163* | *170* | *141 ± 30* |
| | Spring (SOS to EOG) | C | **205** | **205** | **169** | **122** | **233** | **276** | **202 ± 53** |
| | | D | *78* | *151* | *133* | *109* | *109* | *144* | *121 ± 27* |
| | Summer (EOG to SOB) | C | **908** | **718** | **809** | **790** | **888** | **735** | **808 ± 78** |
| | | D | *500* | *672* | *581* | *714* | *684* | *700* | *642 ± 84* |
| | Autumn (SOB to EOS) | C | **142** | **272** | **269** | **310** | **302** | **434** | **288 ± 94** |
| | | D | *138* | *269* | *196* | *259* | *266* | *252* | *230 ± 52* |
| | EOS to 31 December | C | **77** | **14** | **33** | **39** | **–** | **13** | **35 ± 26** |
| | | D | *82* | *64* | *84* | *110* | *55* | *65* | *77 ± 20* |
| | Annual | C | **1386** | **1282** | **1345** | **1328** | **1492** | **1525** | **1393 ± 96** |
| | | D | *954* | *1250* | *1110* | *1283* | *1260* | *1317* | *1196 ± 138* |
| NEP sum | 1 January to SOS | C | **–58** | **–74** | **–68** | **–73** | **–66** | **–66** | **–67 ± 6** |
| | | D | *–117* | *–79* | *–88* | *–82* | *–129* | *–130* | *–104 ± 24* |
| | Spring (SOS to EOG) | C | **103** | **101** | **110** | **92** | **128** | **144** | **113 ± 19** |
| | | D | *25* | *45* | *30* | *4* | *18* | *29* | *25 ± 14* |
| | Summer (EOG to SOB) | C | **80** | **223** | **262** | **374** | **127** | **196** | **210 ± 104** |
| | | D | *442* | *276* | *441* | *288* | *398* | *371* | *369 ± 73* |
| | Autumn (SOB to EOS) | C | **–12** | **–5** | **–6** | **33** | **–48** | **–51** | **–15 ± 31** |
| | | D | *16* | *–26* | *4* | *–18* | *–46* | *–37* | *–18 ± 24* |
| | EOS to 31 December | C | **–35** | **–12** | **–30** | **–24** | **–** | **–12** | **–23 ± 10** |
| | | D | *–68* | *–56* | *–79* | *–103* | *–51* | *–58* | *–69 ± 19* |
| | Annual | C | **76** | **228** | **263** | **395** | **139** | **208** | **218 ± 109** |
| | | D | *292* | *156* | *305* | *90* | *185* | *169* | *200 ± 83* |
| ET sum | 1 January to SOS | C | **22** | **23** | **11** | **19** | **13** | **15** | **17 ± 5** |
| | | D | *56* | *28* | *33* | *24* | *39* | *44* | *37 ± 12* |
| | Spring (SOS to EOG) | C | **105** | **97** | **67** | **65** | **85** | **106** | **87 ± 18** |
| | | D | *36* | *55* | *45* | *31* | *39* | *48* | *42 ± 9* |
| | Summer (EOG to SOB) | C | **315** | **277** | **260** | **286** | **249** | **210** | **266 ± 36** |
| | | D | *283* | *231* | *213* | *219* | *266* | *237* | *242 ± 27* |
| | Autumn (SOB to EOS) | C | **43** | **73** | **82** | **67** | **64** | **97** | **71 ± 18** |
| | | D | *45* | *63* | *50* | *66* | *69* | *64* | *60 ± 9* |
| | EOS to 31 December | C | **15** | **2** | **6** | **4** | **–** | **3** | **6 ± 5** |
| | | D | *14* | *9* | *12* | *14* | *11* | *15* | *12 ± 2* |
| | Annual | C | **495** | **468** | **421** | **436** | **408** | **424** | **442 ± 33** |
| | | D | *428* | *381* | *350* | *349* | *417* | *403* | *388 ± 34* |

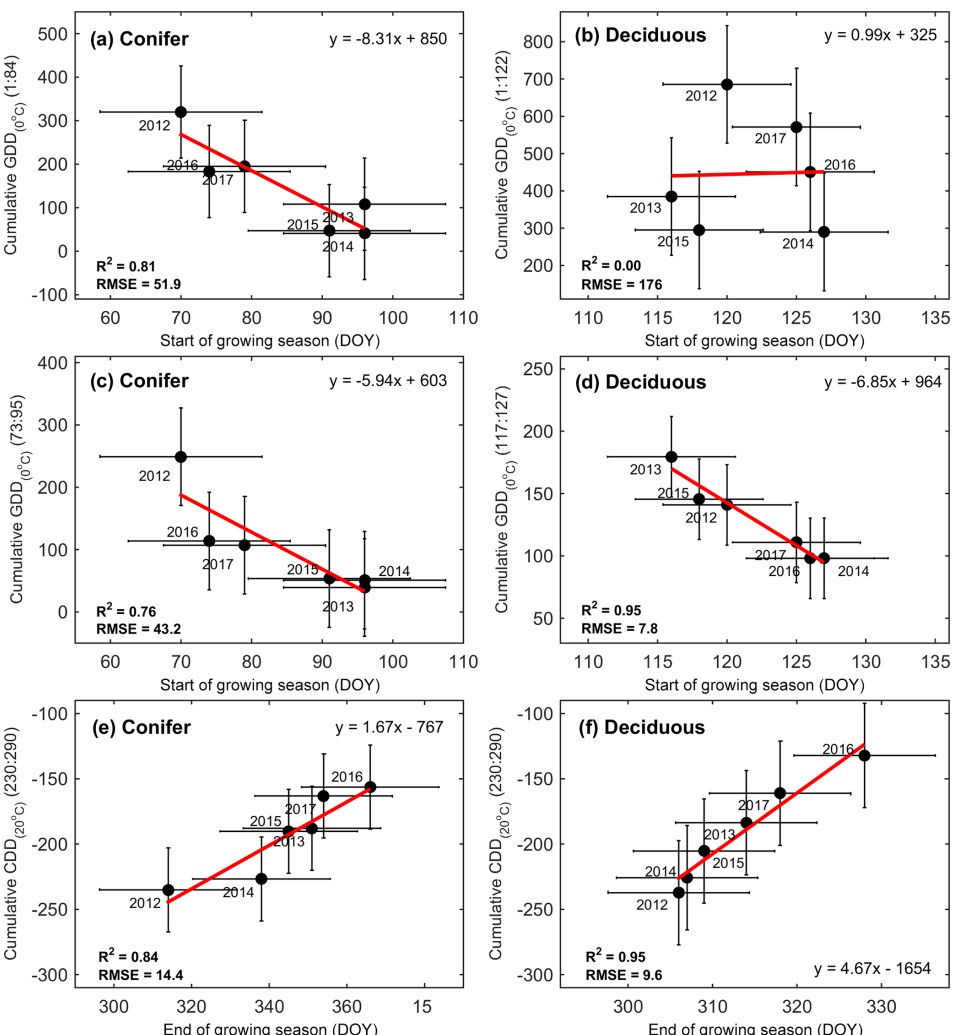

**Figure 4.** Correlations between growing degree days (GDDs), cooling degree days (CDD), the phenological start of the growing season (SOS), and the end of the growing season (EOS) from 2012 to 2017 at both the conifer and deciduous forests. Shown are **(a)** cumulative GDDs from 1 January to the mean SOS at TP39 and **(b)** TPD, **(c)** cumulative GDDs from the mean SOS ± standard deviation at TP39 and **(d)** TPD, and **(e)** cumulative CDDs from DOY 230 through 290 at TP39 and **(f)** TPD. Error bars represent the standard deviation of the data, with $R^2$, RMSE, and linear fit equations included for each correlation.

($263 \, \mathrm{g \, C \, m^{-2} \, yr^{-1}}$). While 2014 ($305 \, \mathrm{g \, C \, m^{-2} \, yr^{-1}}$) was simultaneously the most productive year at the deciduous forest, 2015 ($90 \, \mathrm{g \, C \, m^{-2} \, yr^{-1}}$) was the lowest annual sink, highlighting the differences between sites (Fig. 5b). Similarly, the least productive year at TP39 (2012) was the second most productive year at TPD ($292 \, \mathrm{g \, C \, m^{-2} \, yr^{-1}}$). The cumulative site differences in NEP were analyzed to focus on seasonal differences (Fig. 5c). With earlier SOS at TP39, the conifer site quickly became a sink in spring, while the growing season had not yet begun at TPD. Following SOS, daily NEP at TPD exceeded that at TP39 in all years except 2015 ($p < 0.01$). In the autumn, there was no statistical difference between sites, although as GEP ceased at TPD with leaf abscission, the cumulative difference in NEP be-

tween sites benefited the extended photosynthesis measured at TP39 (Fig. 5c).

Within the evergreen conifer forest (TP39), annual ET was highest in 2012 ($495 \, \mathrm{mm \, yr^{-1}}$) and 2013 ($468 \, \mathrm{mm \, yr^{-1}}$). High $T_a$ throughout much of the year and high summer VPD caused 2012 to have the highest annual ET, while continuous spring and summer $P$ (Fig. 2a) allowed 2013 to sustain higher daily rates of ET (Fig. 3a). Cooler $T_a$ during all of 2014 ($421 \, \mathrm{mm \, yr^{-1}}$) and cooler $T_a$ in the phenological spring of 2016 ($409 \, \mathrm{mm \, yr^{-1}}$), combined with the lowest annual $P$ (in 2016), caused these years to have the lowest ET for the conifer forest (Table 3). Within the deciduous forest (TPD), 2012 ($428 \, \mathrm{mm \, yr^{-1}}$), 2016 ($417 \, \mathrm{mm \, yr^{-1}}$), and 2017 ($403 \, \mathrm{mm \, yr^{-1}}$) had the greatest annual ET, coinciding with the years with the highest annual $T_a$ (Fig. 1b). In 2014, the

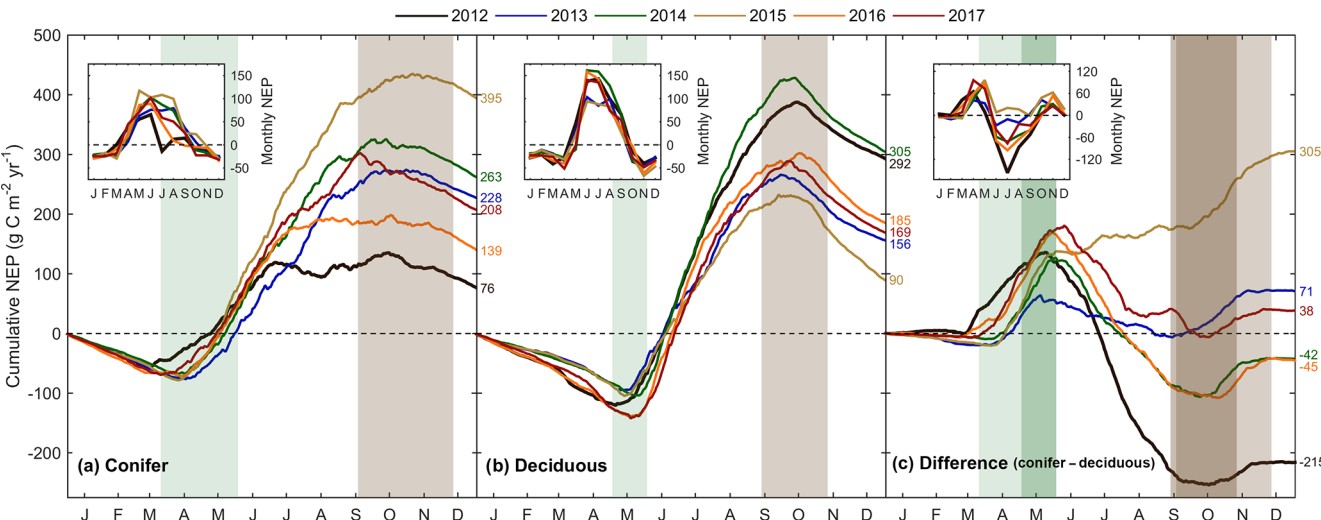

**Figure 5.** Cumulative daily sums of net ecosystem productivity (NEP) at **(a)** the conifer forest (TP39), **(b)** the deciduous forest (TPD), and **(c)** the cumulative difference (conifer – deciduous), with appropriate monthly NEP sums in each panel inset, from 2012 to 2017. Green shading in each panel corresponds to the site-specific 6-year mean phenological spring duration, while brown shading corresponds to the 6-year mean phenological autumn duration (Table 2). Dark shading in panel **(c)** represents the deciduous forest seasons overlaid on the conifer seasons. Cumulative annual values are shown for each site and year, with colors found in the key.

coolest year during the 6 years of measurements, annual ET ($350\,\mathrm{mm\,yr}^{-1}$) was greatly reduced at TPD. While the ET of each forest ultimately responded differently to the local meteorological forcings, on a few occasions, similar daily ET rates were measured, coinciding with significant $P$ events. In the summer of 2013 (30 May to 19 July or DOY 150 to 200), high daily ET was measured at both sites, immediately following multiple daily $P$ events exceeding $40\,\mathrm{mm}$ of rain (Figs. 2a, 3a and b). Additionally, in 2015 (20 June to 10 July or DOY 180 to 200), increased ET was measured at both sites following steady $P$ events. Considering the 6 years as a whole, phenological autumn was the only season in which ET significantly differed between the sites. While the mean autumn ET was greater at TP39, the shorter duration of autumn (Table 2) led to rates of daily ET to be higher at TPD as compared to TP39 ($p < 0.01$). In this case, the phenological autumn at TPD occurred when $T_a$ remained high, while at TP39 autumn stretched later into the year when $T_a$ and daily ET were reduced. Both forests experienced similar variability expressed as standard deviation in ET (33 and $34\,\mathrm{mm}$), and in all years except for 2016, the ET of the conifer forest exceeded that of the deciduous forest.

### 3.4 Forest light and water use efficiencies

The forest light and water use efficiencies (i.e., WUE and LUE) were examined to understand the relationships between forest carbon uptake and site resources (i.e., water and light), illustrated in Fig. 6. At TP39, WUE was the highest in the spring of 2016; the summers of 2014 and 2017; and the autumns of 2015, 2016, and 2017 (Fig. 6a). In 2016, an early SOS (15 March; DOY 74) promoted prompt increases in spring GEP, when $T_a$ and ET remained low. In autumn, the years with extended growing seasons saw GEP increase later in the year as ET decreased, leading to higher WUE. At TPD, WUE was lowest in the warm years (i.e., 2012, 2016, and 2017) due to increased annual ET, while the cool and highly productive year of 2014 experienced the highest summer and autumn WUE (Fig. 6b). In the 6 years of measurements, highly significant ($p < 0.01$) linear relationships of the monthly total ET and GEP (calculating WUE) were measured at both sites, with monthly WUE remaining relatively constant (Fig. 6c; $R^2 = 0.92$). While monthly WUE was similar between forests (Fig. 6c), WUE was higher at TPD ($4.70\,\mathrm{g\,C\,kg}^{-1}\,\mathrm{H_2O}$) when compared to that of TP39 ($3.82\,\mathrm{g\,C\,kg}^{-1}\,\mathrm{H_2O}$).

The general LUE trends and deviations were statistically comparable between the two forests. At both sites, 2014 and 2017 had the highest summer LUE, while reduced GEP at both sites during the summers of 2012 and 2016 yielded the lowest summer LUE (Fig. 6d and e). Across all years, mean monthly linear relationships between GEP and APAR yielded similar results, with larger variation ($R^2 = 0.70$) and lower LUE at TP39 when compared to TPD (Fig. 6f; $R^2 = 0.82$). Similarly, TPD had higher annual (data not shown) and summer LUE ($p < 0.01$), although spring and autumn LUE was similar at both sites.

### 3.5 Meteorological controls on fluxes

Meteorological variables (i.e., $T_a$, PAR, $\theta$, etc.) were analyzed during the study period to better understand their im-

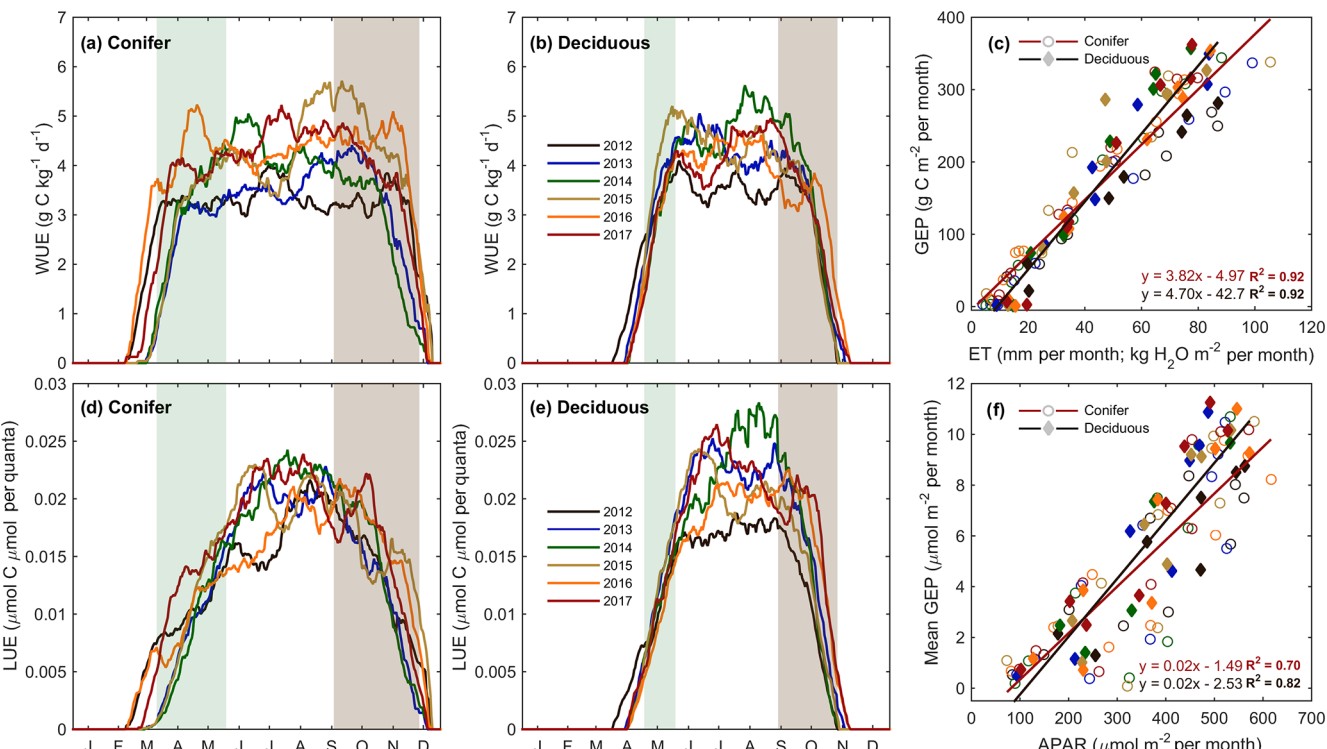

**Figure 6.** Annual smoothed (1-month moving average) time series of the **(a)** conifer (TP39) and **(b)** deciduous forest water use efficiency (WUE; GEP ET$^{-1}$), and **(c)** monthly linear relationships between GEP and ET at both sites from 2012 to 2017. Similarly, light use efficiency (LUE; GEP APAR$^{-1}$) calculations are shown for **(d)** conifer and **(e)** deciduous forests, with linear relationships **(f)** of monthly GEP and APAR also shown. Green and brown shading corresponds to site-specific 6-year mean phenological spring and autumn periods (Table 2), respectively. Linear fit equations and $R^2$ values also shown **(c, f)**.

pact on water and carbon fluxes within each forest. Considering annual values, ET at the deciduous (TPD) forest was found to be highly correlated ($R^2 = 0.84$) to annual mean $T_a$. A smaller secondary effect on ET ($R^2 = 0.83$; Table 4) was found for winter and early spring (1 January to SOS) $\theta_{0-30\,cm}$, which helped to explain the impact of winter soil water storage and seasonal water availability on ET at the start of each year. At TPD, higher winter $\theta_{0-30\,cm}$ was measured in the years with the greatest annual ET. At the conifer (TP39) forest, no strong relationships were found between annual ET values and seasonal or annual meteorological variables. However, monthly linear relationships of $T_a$ and VPD to ET were significant ($p < 0.01$) at both sites (Fig. 7). The evergreen conifer and deciduous broadleaf forests experienced similar increases in monthly ET with increasing monthly mean $T_a$ (Fig. 7a). While the evergreen forest saw higher rates of ET compared to the deciduous forest, the correlation of ET to $T_a$ was greater for the deciduous forest ($R^2 = 0.95$ vs. $R^2 = 0.89$; for TPD and TP39, respectively). The response of monthly ET to monthly VPD was similar between sites, as a mean monthly VPD of 1 kPa corresponded to a monthly total ET of 104 and 97 mm at TP39 and TPD, respectively (Fig. 7b). Overall, the correlation of ET to increas-

ing VPD was greater for the evergreen forest ($R^2 = 0.82$ vs. $R^2 = 0.74$; for TPD and TP39, respectively).

Following similar annual timescales used in the ET comparison, GEP, RE, and NEP were compared to meteorological measurements for each site and season (Table 4). In both forests, no significant relationships were found between meteorological variables and annual GEP. In terms of RE at TP39, the years with the highest annual RE (i.e., 2016 and 2017) resulted from summer drought conditions, as evident through prolonged reductions in mean summer $\theta_{0-30\,cm}$ ($R^2 = 0.89$). The years with the lowest annual RE (i.e., 2013 and 2015) were ultimately the most productive (largest annual carbon sink) and both measured the highest mean summer $\theta_{0-30\,cm}$. The annual NEP was correlated to the length of spring ($R^2 = 0.75$), mean summer $T_a$ ($R^2 = 0.73$), and cumulative summer NEP ($R^2 = 0.99$). For the evergreen conifer site, spring was shorter in years with the highest annual NEP due to rapid photosynthetic development. Higher mean summer $T_a$ decreased annual NEP, highlighting the influence of limitations due to heat stress. Lastly, summer NEP at TP39 was nearly identical to the annual NEP, stressing the importance of this period (roughly June, July, and August) in shaping the annual carbon sink status of the forest.

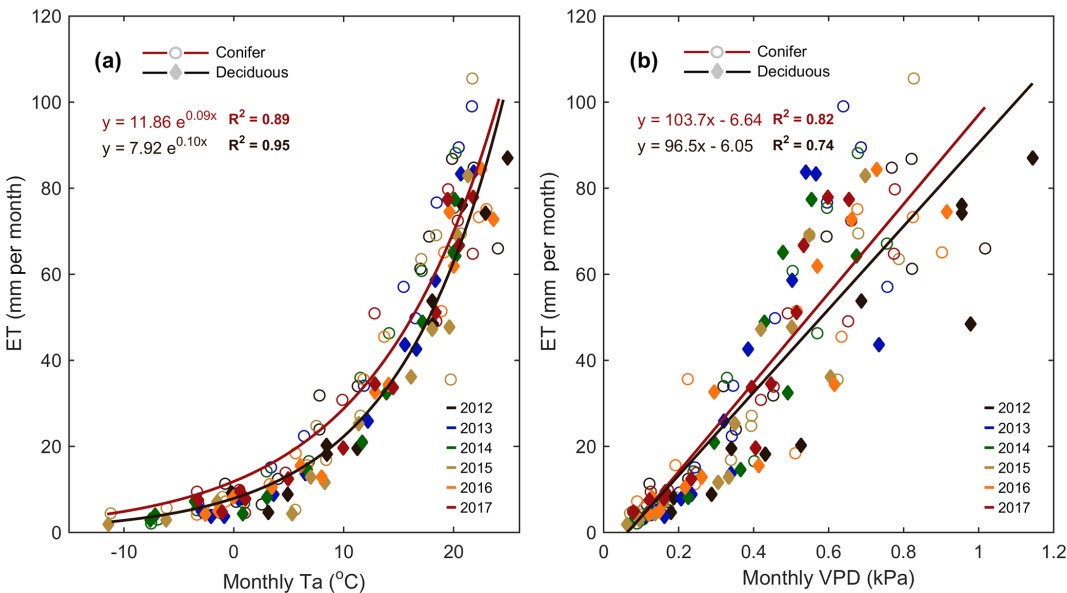

**Figure 7. (a)** Monthly exponential relationships between monthly mean air temperature ($T_a$) and total monthly evapotranspiration (ET) from 2012 to 2017 for the conifer (TP39, open circle) and deciduous (TPD, diamond) forests. Also shown are **(b)** the 6-year linear relationships between monthly mean vapor pressure deficit (VPD) and monthly ET. Fit equations and $R^2$ also shown.

**Table 4.** Select linear relationships between total annual water (ET, mm yr$^{-1}$) and carbon (RE and NEP, g C m$^{-2}$ yr$^{-1}$) flux measurements and both meteorological (i.e., VPD, $T_a$, $\theta_{0-30\,cm}$) and phenological (i.e., spring length, carbon uptake start) variables (annual or seasonal) from 2012 to 2017. In each section, the $R^2$ is for the relationship to the specified annual flux.

| Conifer | 2012 | 2013 | 2014 | 2015 | 2016 | 2017 | $R^2$ |
|---|---|---|---|---|---|---|---|
| Annual RE (g C m$^{-2}$ yr$^{-1}$) | 1386 | 1282 | 1345 | 1328 | 1492 | 1525 | – |
| Summer $\theta_{0-30\,cm}$ (m$^3$ m$^{-3}$) | 0.083 | 0.097 | 0.090 | 0.096 | 0.071 | 0.076 | 0.89 |
| Annual NEP (g C m$^{-2}$ yr$^{-1}$) | 76 | 228 | 263 | 395 | 139 | 208 | – |
| Spring length (days) | 78 | 64 | 58 | 48 | 84 | 80 | 0.75 |
| Summer $T_a$ (°C) | 21.1 | 20.3 | 19.9 | 20.0 | 21.1 | 20.7 | 0.73 |
| Summer NEP (g C m$^{-2}$) | 80 | 223 | 262 | 374 | 127 | 196 | 0.99 |
| Deciduous | 2012 | 2013 | 2014 | 2015 | 2016 | 2017 | $R^2$ |
| Annual ET (mm yr$^{-1}$) | 428 | 381 | 350 | 349 | 417 | 403 | – |
| Annual $T_a$ (°C) | 11.8 | 9.2 | 8.0 | 9.2 | 10.6 | 10.0 | 0.84 |
| Winter $\theta_{0-30\,cm}$ (m$^3$ m$^{-3}$) | 0.131 | 0.118 | 0.116 | 0.101 | 0.133 | 0.127 | 0.83 |
| Annual RE (g C m$^{-2}$ yr$^{-1}$) | 954 | 1250 | 1110 | 1283 | 1260 | 1317 | – |
| Spring $T_a$ (°C) | 16.6 | 15.1 | 16.1 | 15.1 | 15.6 | 14.0 | 0.77 |
| Annual NEP (g C m$^{-2}$ yr$^{-1}$) | 292 | 156 | 305 | 90 | 185 | 169 | – |
| Summer RE (g C m$^{-2}$) | 500 | 672 | 581 | 714 | 684 | 700 | 0.80 |

At the deciduous forest, the relationship between annual RE and spring $T_a$ ($R^2 = 0.77$) suggested that warmer springs generally acted to decrease annual RE. Annual NEP at the deciduous forest was shown to be correlated to summer RE ($R^2 = 0.80$; Table 4). Within the deciduous forest, the years with lower summer RE (i.e., 2012, 2014) were the largest annual carbon sinks. The smallest annual NEP (2015) was observed when summer RE was highest (714 g C m$^{-2}$). On

annual timescales, both sites highlighted the importance of summer meteorological conditions on annual productivity.

Based on the importance of summer outlined above, the flux parameterizations were further examined to understand the dominant meteorological factors during each summer. At the deciduous broadleaf forest, $\theta_{0-30\,cm}$ was shown to have no impact on GEP, while $T_a$, VPD, and PAR contributed to the summer photosynthesis each year (Table 5). Based on me-

**Table 5.** Results of the two-step parameterization process used to explore interannual differences in controlling meteorological variables (i.e., $T_a$, VPD, PAR, $\theta_{0-30\,cm}$) and their impacts on annual RE and GEP during the phenological summer (end of greenup to the start of browndown) for the coniferous and deciduous forests from 2012 to 2017. These normalized values show the cumulative effect of the meteorological variable in reducing GEP and RE from their theoretical maximum values. Higher values (closer to 1) represent more favorable summer conditions for GEP and RE.

| Conifer | 2012 | 2013 | 2014 | 2015 | 2016 | 2017 |
|---|---|---|---|---|---|---|
| GEP: $T_a$ | 0.994 | 0.990 | 0.987 | 0.981 | 1.00 | 0.997 |
| GEP: VPD | 0.939 | 1.00 | 1.00 | 0.981 | 0.914 | 0.975 |
| GEP: PAR | 0.949 | 0.950 | 0.946 | 0.956 | 1.00 | 0.950 |
| GEP: $\theta_{0-30\,cm}$ | 0.956 | 1.00 | 0.998 | 0.993 | 0.976 | 0.973 |
| GEP: All | 0.846 | 0.941 | 0.932 | 0.914 | 0.892 | 0.899 |
| RE: $\theta_{0-30\,cm}$ | 0.958 | 1.00 | 0.996 | 0.991 | 0.968 | 0.965 |
| Deciduous | 2012 | 2013 | 2014 | 2015 | 2016 | 2017 |
| GEP: $T_a$ | 1.00 | 0.971 | 0.974 | 0.967 | 0.989 | 0.974 |
| GEP: VPD | 0.871 | 1.00 | 0.998 | 0.998 | 0.946 | 0.989 |
| GEP: PAR | 0.978 | 0.938 | 0.955 | 0.953 | 1.00 | 0.956 |
| GEP: $\theta_{0-30\,cm}$ | 1.00 | 1.00 | 1.00 | 1.00 | 1.00 | 1.00 |
| GEP: All | 0.852 | 0.911 | 0.929 | 0.920 | 0.936 | 0.920 |
| RE: $\theta_{0-30\,cm}$ | 0.976 | 1.00 | 0.997 | 1.00 | 0.965 | 0.992 |

teorological conditions experienced in each year, 2016 and 2014 were the most favorable for summer GEP, while 2012 was the least favorable. Similar results were found for the evergreen conifer forest, though at that site, low $\theta_{0-30\,cm}$ was shown to influence GEP. Therefore, years with lower $\theta_{0-30\,cm}$ or higher VPD did not experience the same beneficial meteorological inputs necessary for optimal summer GEP. Aside from $T_{s5\,cm}$, $\theta_{0-30\,cm}$ impacted summer RE at both sites. At TPD, the years with the highest summer $\theta_{0-30\,cm}$ (i.e., 2013 and 2015) experienced optimal conditions for enhanced RE, while 2012 and 2016 saw less favorable RE. Similar trends were also found at TP39. Overall, the annual fluxes were a product of the season length and the estimated daily rates of the $CO_2$ fluxes that were in turn influenced by seasonal variability in meteorological variables.

## 4 Discussion

### 4.1 Meteorological and phenological variability

The meteorological conditions at both sites during the study period were characteristic of temperate North American forest ecosystems, characterized by four distinct seasons, with cold winters and warm summers. The close proximity between the two forests ($\sim 20\,km$ apart at the same latitude) led them to experience similar synoptic-scale weather conditions during each year, and therefore nearly identical $T_a$. Even with similar climatic forcings (i.e., $T_a$) seasonal deviations in $T_{s5\,cm}$ were found, likely influenced by the oppos-

ing forest canopy characteristics (Palmroth et al., 2005; Stoy et al., 2006). $T_s$ was linked to the proportion of incoming radiation penetrating the forest canopy, reaching the forest floor. In all years, mean daily $T_{s5\,cm}$ at the conifer forest was higher during each summer, but lower than that of the deciduous forest during the rest of the year. In the conifer forest, branches and needles were closely clumped while the canopy remained comparatively open, leading to minor annual variations in fPAR by the forest canopy and soil, in line with Brümmer et al. (2012). In the deciduous forest, $T_{s5\,cm}$ was higher when leaves were absent and a higher fraction of incoming radiation was directly absorbed by the soil. Following the development and closure of the forest canopy in spring, deciduous $T_{s5\,cm}$ was lower than the conifer forest in our study, which was in line with other similar studies (i.e., Lee et al., 2010; Augusto et al., 2015).

In general, both forests had similar VPD trends in all years, while TP39 had somewhat higher VPD compared to TPD, except in the record warm year of 2012. The higher VPD at the deciduous forest in 2012 could be due to the relative unresponsiveness of stomata to higher VPD typical of broadleaved species, or the suggested larger leaf boundary layers in deciduous trees, where VPD measured above a canopy can be greater than what leaves experience (Baldocchi and Vogel, 1996; Baldocchi et al., 2002; Stokes et al., 2006).

The response of leaf phenology in temperate forests to changes in temperature has been shown throughout much of the Northern Hemisphere (Jeong et al., 2011; Settele et al., 2014). In future climates, rising $T_a$ is predicted to lead to an earlier start, later end, and prolonged duration of the growing season, though ecosystem-level responses are expected to vary as there is a strong genetic control among plant species on the timing of phenological events (Vitasse et al., 2011; Sanz-Perez et al., 2009; Polgar and Primack, 2011; Oishi et al., 2018). In locations such as ours where different tree species face similar climates, the relative advantage of conifer species is seen as the SOS may often precede spring frost events (Givnish, 2002; Augusto et al., 2015). On the other hand, deciduous species (such as *Quercus*) often delay leaf-out to decrease the probability of frost damage (Kramer et al., 2010; Polgar and Primack, 2011), which was seen at our sites. The mean SOS for our conifer (*Pinus Strobus* L.) forest began over a month (38 d) earlier than the deciduous (*Quercus Alba*) forest, with greater variability (between years) seen in the conifer forest, especially in years with warm spring conditions. The timing of the deciduous SOS (2 May; DOY $122 \pm 5$ d) was consistent with similar North American deciduous forests; such as Harvard Forest (4 May; DOY $124 \pm 14$ d; Gonsamo et al., 2015) in Massachusetts and Morgan Monroe State Forest (28 April; DOY $118 \pm 4$ d; Dragoni et al., 2011) in Indiana.

In the autumn, the onset of senescence and EOS has been reported to be advanced by high $\theta$ deficits and delayed with increased warming (Kramer et al., 2010, Warren et al., 2011;

Liu et al., 2016). Both forests experienced later senescence dates with decreased $\theta$ (although likely due to increased $T_a$). For the conifer forest, the two years (i.e., 2012 and 2016) with continued heat and drought stress saw the latest dates of senescence, while at the deciduous forest, greater mean summer $\theta$ led to earlier senescence in all years but decreased $\theta$ extended senescence. Furthermore, we found that the late-summer (August to October) degree of cooling had a significant impact on the EOS as well as overall growing season length. This response has been confirmed by long-term observational data, which has shown strong positive correlations between $T_a$ and EOS, helping to postpone EOS for many forest ecosystems (Dragoni and Rahman, 2012; Gallinat et al., 2015; Liu et al., 2016). More cold days promoted earlier EOS and shorter seasons, while less cooling (greater warming) extended the season and phenologic autumn period at both sites. However, the degree of extension was much different between sites, similar to the response in spring. The mean EOS (10 November; DOY $314 \pm 8$ d) at the deciduous site occurred 1 month (31 d) earlier compared to the evergreen coniferous site (11 December; DOY $345 \pm 17$ d). There was greater variability in EOS at the conifer forest compared to the deciduous broadleaf forest. Based on these findings, in future climates, evergreen conifer forests in the region may expect earlier springs, later autumns, and longer growing seasons, while the deciduous broadleaf forests will likely see greater gains in growing season length from prolonged autumns, limited by their specific leaf strategy.

## 4.2 Meteorological impacts on carbon fluxes

Changes in local meteorology (and climate) have been recognized as a primary factor driving the interannual variability of carbon fluxes within forests (Bonan, 2008; Desai, 2010; Coursolle et al., 2012). Anomalous $T_a$ (extreme heat or cold) and seasonal fluctuations in water availability ($\theta$) over a predictable course of the year were shown to strongly impact the carbon sequestered in many forests (Ciais et al., 2005; Sun et al., 2011; Xie et al., 2014). Conceptually, higher mean $T_a$ will promote longer growing seasons and greater GEP, though increased RE may also be expected (White and Nemani, 2003; Noormets et al., 2015). In this study, the differing forest responses to meteorological conditions led to significant divergences in annual GEP, RE, and NEP. At both sites, the overall growing season length in 2012 was the second shortest (behind 2014), as a result of the anomalously warm $T_a$ experienced throughout much of the year. If this year is excluded, both the conifer and deciduous forests experienced longer growing season lengths with increased annual $T_a$. Annual GEP reductions were experienced in each forest during the heat and drought year of 2012. GEP reductions at our conifer site may also be associated with the reduction in canopy size, due to thinning performed at the site in the early winter of 2012 (see more discussion in the following section). Additionally, higher daily mean $T_a$ and low $\theta$ enhanced

RE in the conifer forest but significantly reduced RE in the deciduous forest. The suppression of RE has been previously reported for other deciduous forests during warm and dry periods (Davidson et al., 1998; Palmroth et al., 2005; Novick et al., 2015; Darenova and Čater, 2018). Overall, these reductions in both the growing season length and the magnitude of carbon fluxes highlighted the forests' sensitivities to heat and drought events, though it ultimately varied between sites. Contrasting studies have shown varying results on the overall drought tolerance of conifer forests. Some studies suggest that conifer species, especially those in resource-poor locations, may be less responsive to seasonal climate anomalies (Aerts, 1995; Way and Oren, 2010; Wolf et al., 2013). Others have found that conifer (i.e., *Pinus*) forests are highly coupled to atmospheric demand and drought sensitivities (Griffis et al., 2003; Stoy et al., 2006). The two years (i.e., 2012 and 2016) with the lowest annual carbon sequestration (NEP) in our conifer forest were found during hot and dry years with high atmospheric demand (i.e., high VPD). These years measured the lowest summer LUE (due to decreased GEP) and the lowest summer NEP, consistent with past studies (Griffis et al., 2003; Vargas et al., 2013). Similar LUE reductions were measured at the deciduous forest during the summers of 2012 and 2016, though annual NEP was drastically different due to comparably large decreases in summer and annual RE, not experienced in the conifer forest. Instead, the two drought years were some of the largest annual carbon sinks (greater positive NEP) during the 6 years of measurements at the deciduous forest. Similar to this study, other research has shown deciduous oak (*Quercus*) forests to be more resilient to drought than their conifer counterparts (Elliot et al., 2015; Wang et al., 2016). Studies have suggested that warm (drought) conditions may lead to reduced carbon uptake or even carbon release (White and Nemani, 2003; Grant et al., 2009; Vargas et al., 2013). Based on our findings, reductions in NEP during expected future intermittent drought conditions in the area could be projected in the evergreen conifer forest, but not in the deciduous broadleaf forest.

Over the measurement period, both forests experienced similar interannual variability in all carbon fluxes ($\sim 100 \, \mathrm{g\,C\,m^{-2}\,yr^{-1}}$) to that expected in midlatitude forests (Yuan et al., 2009; Desai 2010). In all years the magnitudes of GEP and RE were greater in the conifer forest; however, analogous reductions at the deciduous forest led the two forests to have very similar mean annual NEP (despite large annual differences). While evergreen conifer forests have been shown to have lower photosynthetic capacities than deciduous broadleaf forests (Reich et al., 1995; Baldocchi et al., 2010), the longer growing seasons led the conifer forest in this study to have a greater magnitude of annual NEP in half of the years, with drought years being the exceptions. Even in drought years, both the conifer forest and the deciduous forest in our study experienced annual NEP similar to past studies conducted in the temperate region of North America (Barford et al., 2001; Arain and Restrepo-Coupe,

2005; Gough et al., 2013; Xie et al., 2014; Dymond et al., 2016; Oishi et al., 2018). In the coolest year of this study (i.e., 2014), which was closest to the 30-year norm for the area in terms of its mean annual $T_a$, the two forests experienced similar seasonal and annual carbon uptake and some of the highest daily rates over the 6-year study. This suggests that both forests favor meteorologically "normal" years (comparable to the 30-year mean meteorological conditions), equivalent to the conclusion of Griffis et al. (2003) and Gonsamo et al. (2015). Therefore, under future climates, which are predicted to be warmer compared to the current 30-year norm for the area, the carbon sequestration capacity of both forests may be reduced, although to a lesser extent at TPD.

## 4.3 Meteorological impacts on water fluxes

An understanding of WUE is necessary to understand the corresponding release of water vapor (ET) to the atmosphere on seasonal and annual timescales. On average, our conifer forest had greater annual ET and less variability than the deciduous forest. However, we found conflicting results between sites in regard to annual ET during drought years (mainly 2016). At both sites, ET was shown to be strongly driven by $T_a$. ET in 2012 was the highest of all years following amplified $T_a$ for most of the year. Much like RE, ET responds year-round (with summer maxima), so warmer spring or autumn periods often lead to annual increases in ET (Schwartz et al., 2006; Taylor et al., 2008). Similarly, in the deciduous forest, annual ET was heightened during the hot and dry year of 2016. The characteristic amplification of both $T_a$ and VPD during warm drought years led the years with the lowest mean summer $\theta_{0-30\,cm}$ and highest summer $T_a$ (or VPD) to experience increased annual ET at the deciduous forest. A contrasting ET response was measured in the coniferous forest, as 2016 had the lowest annual ET, the only year in which the annual conifer ET was lower than that of the deciduous forest ET.

Typically, transpiration is beneficial to plants, helping to cool leaves and thereby reducing respiration (Rambal et al., 2003; Baldocchi et al., 2010; Brümmer et al., 2012). In our case, high summer $T_a$, the lowest $\theta_{0-30\,cm}$, and very little summer and annual $P$ (input) into the system significantly reduced ET, while RE continued to rise. At the conifer forest, the timing of summer $P$ events appeared to influence ET (i.e., 2013). However, it is likely that the opposing responses of ET to soil water availability between sites was due to the ability of each forest to access deep soil water storages. Studies have shown oak (*Quercus*) forests to be less sensitive and more resilient to drought, due to more efficient access to deeper soil water than conifer forests (Bréda et al., 2006; Bonan, 2008; Wang et al., 2016; Matheny et al., 2017). Evergreen conifer forests may have roots extending just as deep as deciduous broadleaf forests, but they are not as effective at obtaining water as those of broadleaf trees (Oren and Pataki, 2001). With higher atmospheric demand during dry periods often leading to greater ET across many forest types (Meinzer et al., 2013; Wu et al., 2013; Tang et al., 2014), the access and availability of water in deep soil layers allowed the deciduous forest to sustain high ET, even in drought years.

We found the course of annual WUE of both forests to respond similarly across all years, though variations in GEP and ET between the forests led to seasonal WUE differences due to the aforementioned responses of both fluxes. The WUE at the conifer forest was consistent with previously reported values for that location (Brümmer et al., 2012; Skubel et al., 2015), while the deciduous forest WUE was found to be higher than a regionally similar oak-dominated forest in Ohio (Xie et al., 2016). Assuming similar daily rates of carbon assimilation (GEP), higher WUE implies a higher evapotranspiration flux at the conifer forest (Augusto et al., 2015), which we saw.

## 4.4 Forest management and future climate impacts

Forest age, management practices, and historical land use have been shown to impact annual carbon fluxes within forests (Wofsy et al., 1993; Song and Woodcock, 2003). While our forests are of relatively similar age ($\sim$ 80–90 years), they have experienced different management practices over their lifetimes so far, with the coniferous forest being a planted forest that underwent low-density partial thinning in 1983 and 2012, while the deciduous broadleaf forest was naturally regenerated with periodic selective harvesting in the past. The difference in carbon uptake over the forest's life will be influenced by management treatments (Herbst et al., 2015). Some studies (Zha et al., 2009; Dore et al., 2012; Skubel et al., 2017) have suggested that overall forest carbon and water fluxes recover rapidly post-disturbance. Furthermore, some studies have found a positive correlation between species number and productivity in temperate forests (Morin et al., 2011). Similarly, mixed forests are generally assumed to be more resilient to extreme weather events and disturbance events than mono-specific forest stands (Pretzsch, 2014; Herbst et al., 2015). With a greater number of species in our deciduous broadleaf forest (over 500 tree and plant species, as per Elliot et al., 1999), and the resistance to heat and drought induced carbon losses shown in this study, it is likely that the deciduous broadleaf forest will remain a carbon sink well into the future. Even following increased RE losses expected with warmer late-summer and autumn conditions (Dunn et al., 2007; Piao et al., 2008), such as those experienced in 2016 and 2017 at our site, the conclusions remain the same.

For similar forest types, the annual responses of GEP and RE to local meteorology will affect natural and managed forests similarly; however, it has been proposed that many managed forests may already be maximized for a given $T_a$ regime, leaving less room for adaptability or acclimation in the future (Litton and Giardina, 2008; Chen et al., 2014; Noormets et al., 2015). With RE shown to be higher in man-

aged forests compared to natural forests (Arain and Restrepo-Coupe, 2005), it is possible that our conifer forest may see limitations in the annual carbon sequestration capability in the future. With considerable daily RE losses experienced following summer $P$ events (i.e., 2013 and 2014), enough hot periods with intermittent heavy rains in the future could cause forest RE to increase in the conifer forest. As the climate continues to change, the management practices and responses to meteorological conditions will determine the relative carbon sink or source strength in many temperate forests.

## 5  Conclusions

The annual carbon and water dynamics were compared between two forests of different leaf strategy in the Great Lakes region of southern Ontario, Canada, over a 6-year (2012 to 2017) period. The geographic location, forest age, soil characteristics, and climate were similar in both stands, where one was an evergreen needleleaf conifer plantation while the other was a naturally regenerated deciduous broadleaf forest TS1. ~~Management treatments were applied in both forests.~~ On average, the evergreen conifer forest was a greater carbon sink ($218 \pm 109\,\mathrm{g\,C\,m^{-2}\,yr^{-1}}$), with higher annual ET ($442 \pm 33\,\mathrm{mm\,yr^{-1}}$) than the deciduous broadleaf forest ($200 \pm 83\,\mathrm{g\,C\,m^{-2}\,yr^{-1}}$ and $388 \pm 34\,\mathrm{mm\,yr^{-1}}$, respectively). While mean annual fluxes were similar in magnitude and variation, differences were measured between sites, especially during drought years. Summer meteorology was shown to impact fluxes at both sites, though to varying degrees with varying responses. ~~Annual NEP was reduced at the deciduous forest during years with increased summer RE~~ TS2. Similarly, annual ET at the deciduous forest was driven by changes in $T_a$, with the largest annual ET measured in the warmest years. During droughts, the carbon and water fluxes of the deciduous forest were less sensitive to changes in temperature or water availability. The annual NEP at the conifer forest was ultimately shaped by total summer NEP. The significant response of the conifer forest to heat and drought events led the summer months in all years to greatly control the forests annual carbon sink–source status. Additionally, prolonged dry periods with increased $T_a$ were shown to greatly reduce ET (i.e., 2016). Both sites saw average ET but increased NEP (against the 6-year study mean) during climatologically (30-year mean) "normal" years, but only the conifer forest saw annual reductions in carbon sequestration during drought years. We also found that drought-induced RE increases or GEP decreases may impact the overall net carbon uptake in the coniferous stand. Our study suggests that the deciduous forest will continue to be a net carbon sink under increased temperatures and larger variability in precipitation under future climate changes, while the response of the coniferous forest will continue to remain uncertain.

Please note the remarks at the end of the manuscript.

**Appendix A**

**Table A1.** Descriptions of the eddy covariance (EC) instrumentation and meteorological sensors used at both sites during the period of measurements. Note: IRGA = infrared gas analyzer.

|  | Turkey Point 1939 (TP39) | Turkey Point Deciduous (TPD) |
|---|---|---|
| Canopy IRGA | LI-7000 (LI-COR) | LI-7200 (LI-COR) |
| Sonic anemometer | CSAT3 (CSI) | CSAT3 (CSI) |
| Height | 28 m (2003–May 2016) 34 m (May 2016–present) | 36 m (2012–present) |
| Orientation | Oriented west (270°) | Oriented west (270°) |
| Intake tube | 4 m long intake tube | 1 m long intake tube |
| Flow | $15\,\mathrm{L\,min^{-1}}$ | $15\,\mathrm{L\,min^{-1}}$ |
| Mid-canopy IRGA | LI-800 (LI-COR) Measured at 14 m height | LI-820 (LI-COR) Measured at 16 m height |
| Air temperature ($T_a$) Relative humidity (RH) | HMP45C (CSI) | HMP155A (CSI) |
| Wind speed and direction | Model 05103 (R.M. Young) | Model 85000 (2012–2015) Model 05013 (2015–present) |
| Photosynthetically active radiation (PAR) | PAR-Lite (Kipp & Zonen) | PQSI (Kipp & Zonen) |
| Net radiation (Rn) | CNR1 (Kipp & Zonen) | CNR4 (Kipp & Zonen) |
| Soil temperature ($T_s$) | 107B (CSI) | 107B (CSI) |
| Soil water content ($\theta$) | CS615-L/CS616 (CSI) | CS650 (CSI) |
| Precipitation ($P$) | T-200B (GEONOR) | CS700H-L (CSI) |

*Data availability.* The data presented in this study are available at https://doi.org/10.17190/AMF/1246152 (Arain, 2012; deciduous forest) and https://doi.org/10.17190/AMF/1246012 (Arain, 2003; coniferous forest).

*Author contributions.* ERB collected, cleaned, and processed the data with help from JJB and BMB. ERB, MAA, and MK designed the experiment, with grants received by MAA. ERB and JJB performed the statistical analyses. ERB interpreted the data, prepared the figures, and wrote the article with editorial contributions from all authors.

*Competing interests.* The authors declare that they have no conflict of interest.

*Acknowledgements.* This study was funded by the Natural Sciences and Engineering Research Council (NSREC), the Global Water Futures Program (GWF), and the Ontario Ministry of Environment, Conservation and Parks (MOECP). Funding from the Canadian Foundation of Innovation (CFI) through the New Opportunity and Leaders Opportunity Fund and Ontario Research Fund of the Ministry of Research and Innovation is also acknowledged, as is the kind support from the Ontario Ministry of Natural Resources and Forestry (OMNRF). The St. Williams Conservation Reserve Community Council and the Long Point Region Conservation Authority (LPRCA) are also acknowledged. We acknowledge support from Zoran Nesic at the University of British Columbia in assistance with flux measurements at our site, and members of the Hydrometeorology and Climatology lab at McMaster University for their continued support at both sites.

*Financial support.* This research has been supported by the Natural Sciences and Engineering Research Council (NSREC) (grant no. 1506), the Global Water Futures Program (GWF), the Ontario Ministry of Environment, Conservation and Parks (MOECP), the Canadian Foundation of Innovation (CFI), and the Ontario Ministry of Research and Innovation.

*Review statement.* This paper was edited by Ivonne Trebs and reviewed by two anonymous referees.

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

**Remarks from the language copy-editor**

CE1      Please check; "of wood" did not seem to be in your original file.
CE2      Please check.
CE3      Should this be $R_n$?

**Remarks from the typesetter**

TS1      Please note that the requested changes in the conclusion should be approved by the editor. Therefore, I would kindly ask you to provide a short explanation regarding these changes that can be forwarded by us to the editor. Thank you very much in advance for your help.
TS2      See previous remark.