# Peer review of "Response of carbon and water fluxes to meteorological and phenological variability in two eastern North American forests of similar age but contrasting species composition – a multiyear comparison"

_Biogeosciences, 2019_

## Referee Comment (RC1) · Anonymous Referee #1 · 13 Feb 2020

The authors present two long-term eddy covariance carbon dioxide and water vapor flux data sets from two distinct forest ecosystems in Canada. One site is a deciduous broadleaf forest, the other one an evergreen needleleaf forest. The authors report results from modeling and partitioning these fluxes and address impacts of driver variability on different temporal scales on the variability of the fluxes. In particular, the authors found a reduction of net annual carbon dioxide uptake in both forests as a result of above average mean summer air temperatures and a larger reduction of annual net carbon dioxide uptake at the coniferous forest compared to the deciduous forest

during drought years. The data and conclusions are highly relevant and clearly within the scope of the journal.

The manuscript is fairly well written, although editing could have been conducted more thoroughly at times.

General formal points are:

- Latin plant names should consistently be printed in italic font.

- Definitions of abbreviations appear repeatedly throughout the text, they should only be introduced on their first occurrence.

- The verbs "to experience" and "to respond" are used excessively and sometimes not in the appropriate context. It is clearly a matter of taste but I would advise to revise some sentences. Fluency could partly be improved by language simplification.

- Tenses are not always used consistently, please revise (see line comments).

Scientific issues:

- "Carbon" is partly used interchangeably with "carbon dioxide". There are more components to the carbon cycle in forests than vertical CO2 exchange. Therefore, sometimes statements are not entirely correct, please review.

- Measurement and model uncertainty are not addressed. The authors should add some information on this topic. The ranges of the annual flux sums given in the abstract likely describe inter-annual variability (not measurement/model uncertainty), I assume using mean and standard deviation of the six annual flux sums per forest. An explanation should be added.

- The description of the used partitioning models (equations 1 and 2) is very concise, at least units should be added. For equation 1 a citation is provided, the short description is defendable. Equation 2, however, is not clearly referenced and therefore definitely needs more explanation. The optimization process of the temperature, VPD

and soil moisture functions behind the scaling terms need to be described better, to only mention the sigmoidal shape is not enough in my opinion.

- Some of the conclusions about the effects of drought rely on the analysis of the especially warm and dry year 2012. The fact that there was a disturbance (thin cutting) in one of the forests in this year is not discussed comprehensively enough. The authors should for example include the effect of a diminished leaf area on $CO_2$ exchange fluxes in their interpretation of this (and the next?) year's budget and explore if for the interpretation of the data set from 2012 to 2017 post disturbance effects should be considered.

Line comments:

Page 1, Line 3 (Title) "similar-age" should not be hyphenated.

Page 1, Line 19 I would suggest replacing the somewhat complicated sentence ",...the evergreen forest saw greater annual reduction" with e.g. "...,net $CO_2$ uptake was reduced more at the evergreen forest than at the deciduous forest."

Page 1, Line 22 "Annual ET was driven by changes in air temperature" Are you sure? Is T change really the driver? It sounds like the slope of a T change determines ET. If so, which timescale do you refer to? Maybe average temperature actually is the driver?

Page 1, Line 23 "During drought years,..." It is a bit hard to follow the logic. The preceding sentence says that dry periods greatly reduced ET at the deciduous forest. Now it is stated that the sensitivity of ET to temperature changes (?) at the deciduous forest is comparably low. Maybe say: ET is sensitive to dry periods/increased T at both sites. ET reduction at TP39 is comparably larger.

Page 1, Line 25 "If longer periods..." Longer than what? Can you give us an idea about time scales (hours, days, months)?

Page 1, Line 26 "...the carbon sink capacity [...] will continue." is a bit complicated. Maybe "...will continue to act as a sink..." "...while that of..." is not very elegant, consider

reformulating.

Page 1, Line 29 Remove comma before "through". "Absorption of CO2 emissions" can be replaced by "CO2 uptake".

Page 1, Line 30 remove "processes"

Page 2, Line 38 I would add "events" after "extreme weather". Remove "stress". Stress is the consequence of extreme weather not an example for an extreme event.

Page 2, Line 39 "Adversely impacting [...] forest–atmosphere interactions" What does that mean? Sounds like there is no interaction anymore due to extreme weather, you clearly do not mean that. Also: replace hyphen with en-dash in expression "forest–atmosphere".

Page 2, Line 40 The authors state that there are positive and negative feedbacks but give only an example for a process leading to a positive feedback. Example for the opposite case?

Page 2, Line 46 I do not get the reasoning. "The result of a shifting climate..." [which result?] impacts both forest types differently because broad-leaved species are replaced by needle-leaved species? I do not understand the cause-effect concept behind the statement, consider revising.

Page 2, Line 50 I assume you refer to a disturbance of regional cycles and not within forest cycling, can be formulated more clearly.

Page 2, Line 51 "Conversely,..." I do not see an opposition to the previous statement, which is about photosynthetic rate. This sentence talks about season length.

Page 2, Line 58 "...have the ability to conduct research..." is needlessly convoluted. Consider replacing with e. g. "Few studies have reported multi-annual time series." Also: omit "sufficiently long". Otherwise you need to explain which timescale would be sufficient for what and why.

Page 2, Line 59 In my opinion, there is no need to construct ("Such a study would...") the need for the current study. I would omit lines 59 to 63 and go straight to Page 3, Line 73 ("This study...").

Page 2, Line 61 The "benefit" of forests to "terrestrial–atmosphere gas exchange" seems vague. Gas exchange takes place anyway, there only is a benefit if you prescribe a service of forests (e. g. carbon sink function), which is not mentioned here. As stated before, I would omit the whole section.

Page 3, Lines 64 to 69 Should be moved to section 2.1 (Study sites)

Page 3, Lines 70 to 73 As no results of the previous studies are mentioned here, listing them is not very informative. I would move this section to the results or discussion section and mention the results of previous studies there in comparison/relation to the current study.

Page 3, Line 80 "will be used". The choice of tense in confusing to me. Starting in line 73, present tense is used, future here.

Page 3, Line 83 What is "natural terrain"? What would be the difference to unnatural terrain?

Page 3, Line 83 "The forest is classified". By whom? Is there a citation or a classification system this subsumption refers to?

Page 3, Line 91 "Conifer species including make up..." Sentence incomplete.

Page 4 Line 106 Personally, I do not like the frequent use of the verb "experience". For this sentence a more simple way could be: "While edaphic and climatic conditions are similar between both sites, they differ in vegetation cover and canopy structure."

Page 4 Line 107 What do you mean by "historically defined"? That past events (ice age) shaped the landscape or that authors in the past defined the landscape like this?

Page 4 Line 109 It would be easier for international soil scientists to understand if the

name according to the FAO World Reference Base would be given additionally to the name according to the national Canadian system.

Page 4, Line 113 "Help" is not ideal. How does the lake control cold temperatures?

Page 4, Line 114 "...were 8 °C and..." past tense? The mean is still the mean. Next sentence present tense again.

Page 4, Line 116 The citation is incomplete. Based on the information provided, the given data cannot be verified.

Page 4, Line 116 Last sentence of paragraph can be omitted, it is poorly formulated. Information also given in "Data availability" section.

Page 4, Line 121 Omit ", though". Start new sentence with "Measurements".

Page 4, Line 124 Supplementary material would be a separate pdf-file I think. Table A1 seems to be in the appendix.

Page 4, Line 124 "...are calibrated". Present tense? Paragraph starts in present perfect (..."have been measured").

Page 4, Line 125 The expression "Environment Canada Greenhouse gas specified $CO_2$" is not understandable. Which concentration did the span gas have?

Page 5, Line 127 It comes as a surprise that there is more than one IRGA per EC setup. In line 123 singular was used ("...an IRGA"). I would stress this type of setup more as it is typical and necessary for forest EC.

Page 5, first two paragraphs A mixture of tenses is used. "is completed", "were assumed", "have been conducted", "are measured", "will focus". Check for consistency.

Page 5, Line 139 Unclear what "Environment Canada Delhi CDA" is. Why mention if precipitation data is not used after all (as stated in line 141)?

Page 5, Line 145 What is the difference between quality control, filtering and cleaning?

If you do not want to go into detail just mention the citation and say e. g. "processed as described by Brodeur (2014)".

Page 5, Line 147 How was the frequent cross-checking with AmeriFlux done? Statement seems vague.

Page 5, Line 148 How were outliers identified?

Page 5, Line 150 There are other EC towers at Turkey Point Observatory? Where are they? Can you expect them to be representative for your site? Only then using them to gap-fill your data would make sense. More information needed.

Page 5, Line 150 What is "mean flux recovery"? Percentage of half-hourly measurements left after filtering? Including or excluding times of instrument maintenance/malfunction?

Page 5, Line 159 Omit "where daytime and nighttime"; it means all fluxes, correct? No need to specify then.

Page 6, Line 160 It is stated that filtered NEE was gap-filled using soil temperature. Why is "flux recovery" after gap-filling only 49 %. Check if this gap-filling step was actually applied. It seems unlikely. Later more complicated methods for flux partitioning and gap-filling are described. The simple NEE-Ts model seems redundant.

Page 6, Line 164 Symbol for soil moisture appears here first. Explanation too late in line 163.

Page 6, Line 164 Partitioning of NEE into GEP and RE has not been introduced. The reader does not know the RE time series at this point. If you talk about gaps in it you have to introduce it first.

Page 6, Line 166 What is the definition of nighttime? A radiation threshold?

Page 6, Line 166 It is stated that nighttime NEE was modeled as a function of soil temperature and moisture in order to (!) describe the relationship of RE and Ts which

represents diurnal air temperature variability. Check meaning of the sentence. It seems incoherent to use nighttime measurements to describe diurnal variability of something.

Page 6, Line 173 What are the units of the model parameters, especially of a1 and a2? a1 and a2 are not a function of soil moisture (as stated) when looking at equation 1. I assume all four parameters were fit during the same optimization process.

Page 6, Line 173 "...acting to scale the RE relationship" to what?

Page 6, Line 180 Explanation of equation 2 needs more detail. How are these sigmoidal functions set up? Do they have parameters? Are all parameters optimized at the same time?

Page 6, Line 187 Seems inconclusive. Don't you need the modeled GEP time series in order to calculate phenologically-derived summer months? For the GEP model you in turn need the derived summer months. Please explain.

Page 6, Line 189 Sentence starting with "Furthermore..." ending in line 191 with "both sites" can be omitted, unnecessary/circular information. Yes, in the growing season plants grow, therefore it is a key season of CO2 uptake.

Page 7, Line 201 Omit first sentence of paragraph, contains no new information.

Page 7, Line 208 "water or heat stressed periods", check meaning, the periods are not under stress.

Page 7, Line 210 Contents of last paragraph can be moved to results, stays a bit vague here anyway.

Page 8, Line 237 GEP might not be gap-filled, still it is not direct measurement data but modeled as the difference of RE (modeled) and NEE (=EC Fc, measured). Could be stressed here, it took me a while to get my head around this fact.

Page 8, Line 240 The approach does not calculate, the computer calculates according to the approach.

Page 8, Line 241 "logistic curve" instead of "logistics curve"

Page 8, Line 241 "The second derivative estimated the end of greenup..." How? Time when derivative turns zero or similar?

Page 8, Line 242 "while the third derivatives calculated..." see two comments above.

Page 8, Line 251 accumulation

Page 8, Line 257 Ta responds to what?

Page 8, Line 258 "Record warm Ta conditions". Expression unclear to me. Annual mean above 30-year average? Most days/half hours above 30-year average of corresponding DOY/half hour?

Page 8, Line 260 What does extreme mean in terms of values? What does "magnitude of extreme cold days" mean exactly?

Page 9, Line 261 "record Ta outside the normal peak summer period" Unclear, what does record and normal mean? Temperature is outside the period? Check meaning of sentence.

Page 9, Line 262 The sites are not growing, the vegetation is.

Page 9, Line 263 "Meteorological conditions between the sites were [...] examined". Check meaning. Consider replacing with "Differences in meteorological conditions between the sites were examined" or "Meteorological conditions at both sites were examined"

Page 9, Line 263 "..., beginning with the amount..." Sequence of analysis steps not relevant.

Page 9, Line 264 Sentence "However, the shapes..." is circular and can be omitted. It says: The seasonal course of APAR depicts the course of absorbed PAR, meaning APAR is APAR.

Page 9, Line 267 "APAR was similar throughout the year". Not true, see figure 2. Relative quantity FPAR might be about constant during annual course, APAR is not.

Page 9, Line 270 Cloudy conditions along with a reduction in incoming radiation are not a coincidence.

Page 9, Line 278 Could replace "followed closely to Ta" with "follow Ta closely"

Page 9, Line 281 replace "of TPD" with "at TPD".

Page 9, Line 282 replace "similar patterns between sites" with "similar patterns at both sites"

Page 9, Line 283 Soil moisture deficit compared to what? At which value does it start to be deficient?

Page 9, Line 283 "In summer" comma missing

Page 9, Line 285 "while all other times of the year TP39 was higher". Soil moisture was higher not TP39.

Page 9, Line 292 Consider replacing unit "day" with unambiguous "day of year (DOY)" throughout manuscript, first occurrence here.

Page 10, Line 295 Check meaning. "The response [...] to changes in GDD was considered as a trigger for SOS." The response is the trigger? I think GDD change is the trigger and the response of the forest to this trigger manifested in SOS.

Page 10, Line 296 "cumulative GDD" GDD is cumulative by definition, is it not?

Page 10, Line 297 Cumulative heat is not expressed directly in GDD, GDD is a proxy for absorbed heat as correctly stated above. I would omit the half-sentence "However, [...], which we calculated as"

Page 10, Line 299 "represented" not any more? check tense.

Page 10, Line 303 replace "start" with "are reached"

[Figure]

Page 10, Line 314 Omit first sentence of paragraph, it is a bit vague. "influenced by a certain degree of cooling"?

Page 10, Line 316 replace "were found to be highly correlated" with "were highly correlated"

Page 10, Line 325 "At first glance..." Sentence seems vague. What do you mean by similar? Which properties of the forests responded similar to which forcings? What does "seasonal irregularities" mean? Difference between same season of different years or within one year between seasons? How do these irregularities govern annual fluxes (cumulative fluxes?). Highest contribution to sum during periods when forcings deviate from average behavior? Consider restructuring or omitting sentence.

Page 11, Line 327 replace "within" with "at"

Page 11, Line 337 "...did not greatly benefit the forest..." seems unassertive. What do you mean? No increase in CO2 uptake? If the latter is meant, I would question the statement. Sure, when you look at average daily GEP, a longer spring increases n for the conifer forest and adds mostly low values (from earlier in the year) lowering the average. Looking at spring GEP/NEP sums might lead to a different interpretation.

Page 11, Line 339 Details about statistical tests could be inserted here. I am not sure what the p-value refers to, a t-test?

Page 11, Line 341 I would replace "minimums/maximums" with "minima/maxima", might be a matter of taste.

Page 11, Line 341 How is a maximum significant? Consider removing.

Page 11, Line 342 RE was modeled not measured.

Page 11, Line 344 replace "let the year to have" with "led to"

Page 11, Line 346 see previous comment

Page 11, Line 349 response to what?

Page 11, Line 351 check meaning. Ta always high between rain events?

Page 12, Line 363 Should it be "sink" instead of "source"?

Page 12, Line 369 Check meaning. "NEP [...] exceeded TP39"

Page 12, Line 385 "to" missing, should be "let to rates"

Page 12, Line 387 Consider replacing "deviations" with "variability expressed as standard deviation" and omitting the plus-minus sign in brackets.

Page 12, Line 391 "WUE varied [...] due to different [...] overall GEP and ET". Check statement, seems circular to me. Does it say: "The ratio of GEP and ET varies because GEP and ET vary"?

Page 13, Line 394 "...,the SOS began..." Reformulate, now it says "the start began"

Page 13, Line 396 remove "forest"

Page 13, Line 400 Same number for TPD and TP39. Also: What is the uncertainty of these slope estimates?

Page 13, Line 405 monthly GEP and APAR sums or averages?

Page 13, Line 409 Sentence incomplete. "To better understand and the water...."

Page 13, Line 410 remove "first". Sequence of analysis steps irrelevant.

Page 13, Line 412 "the impact of winter soil water storage..." on what?

Page 13, Line 419 Consider reformulating "responses between". I would expect "the response of something to something else"

Page 13, Line 425 Maybe there is no linear relationship between GEP and meteorological variables. There should, however, definitely be relations with PAR. As far as I understand GEP was modeled using PAR, you should see the saturation curve you

prescribed in the model (eqn. 2) in a PAR-GEP plot.

Page 13, Line 426 There is an extra space after the closing bracket and "resulted"

Page 14, Line 429 Why "most importantly"? Mean or cumulative summer NEP?

Page 14, Line 431 "was seen" is not very elegant. Consider simplifying the sentence, e.g. "...spring was shorter due to..."

Page 14, Line 431 "Higher summer Ta". Season average or half-hourly or daily peaks?

Page 14, Line 431 "relationship between RE and spring Ta". timescales? annual RE, spring RE, sums or averages?

Page 14, Line 437 "Lastly,...", "Ultimately,..." can be omitted. Sequence of analysis irrelevant.

Page 14, Line 428 They sites do not emphasize, you do.

Page 14, paragraph starting in line 439 This paragraph requires more explanation. How were the model parameters examined? The methods section is not detailed enough about this type of analysis, Table 5 is also ambiguous ("GPP:Ta" sounds like correlation analysis. Should it be f(Ta) as in eqn. 1 to denote that the scaling factor is meant?). The scaling method is very interesting, it deserves a proper explanation for others to be able to reproduce it.

Page 14, Line 445 "Outside of Ts". Sounds strange to me. Do you mean "apart from"?

Page 14, Line 447 Similar response of what to what?

Page 14, Line 448 What do you mean with "predicted daily rate"? The observed fluxes were the result of a prediction? I do not understand, consider clarifying.

Page 14, Line 451 replace "experienced by" with "at"

Page 14, Line 451 Typical meteorological conditions? Introduction says air temperature was consistently above the 30-year average.

Page 14, Line 455 "certain differences were primarily influenced" is a bit vague, which differences, why primarily. What about relief position, water content or soil type?

Page 14, Line 456 "In this case" Soil temperature is always linked to incoming radiation.

Page 14, Line 457 Mean Ts or each half-hourly value?

Page 15, Line 459 What does "highly clumped" mean? High compared to what?

Page 15, Line 459 Minor variations in APAR? Maybe true for fPAR, looking at Figure 2 APAR seems highly variable throughout an annual course.

Page 15, Line 461 "Incoming radiation was directly absorbed by the soil" All of it? What about LE etc.? Not all energy goes into ground heat flux.

Page 15, Line 464 Incomplete sentence. "...similar trends VPD..."

Page 15, Line 469 "species specific responses shaped the timing of phenological events" Responses to what? Isn't it obvious that species type determines phenology?

Page 15, Line 480 There is only one SOS per year. How can SOS have high variability in a warm year when there is only one value per year?

Page 15, Line 486 Seems contradictory. Either timing of senescence and soil moisture are not related ("insignificant") or the forests experienced "later senescence dates with decreased soil moisture". If the finding opposes previous studies it would be interesting to read about possible reasons (water stress?).

Page 16, Line 496 replace "in the deciduous site occurred a month (31 days) before that of the evergreen..." with "at the deciduous site occurred one month (31 days) earlier compared to the evergreen..."

Page 16, Line 497 omit "experienced"

Page 16, Line 500 "only limited by their specific leaf strategy". This seems to be a major argument (Title!). Can you expand more, why "only" limited by this strategy?

Page 16, Line 503 "Ta anomalies [...] strongly determine the carbon sequestered". Check meaning. Ta determines the carbon? Maybe the amount of carbon? Are you sure the anomalies determine C uptake as opposed to the average temperature?

Page 16, Line 505 ",... higher Ta..." Anomalies, average, min/max?

Page 16, Line 506 "drawback" only if maximum sink strength is the goal. why judge?

Page 16, Line 507 typo: "differing forest[s] responses"

Page 16, Line 508 "season length in 20123 was the second shortest despite..." Maybe there is another factor co-controlling season length then?

Page 16, Line 509 Maybe not "despite" but "because" high air temperatures. There could be an temperature optimum (parabolic function) for GEP. What does "record Ta" mean? Daily/Half-hourly maximum, mean, average above long-term average?

Page 16, Line 510 Why "also"? Section already talks about the outlier year 2012.

Page 16, Line 512 "due to thinning performed..." Definitely! This fact is introduced too late. Such a disturbance could single-handedly be responsible for budget deviations in 2012 and override all possible reasons stated before. The disturbance must be stressed and discussed more and earlier.

Page 16, Line 513 "higher Ta and low theta" Annual/seasonal mean or each/most half-hours/days? Replace "acted to enhance" with "enhanced"

Page 16, Line 525 replace "due to comparable decreases" with "due to comparably large decreases"

Page 17, Line 535 "very similar NEP" at both sites vs. Page 17, Line 538 "led the conifer forest [...] to have a greater magnitude of annual NEP". Is NEP similar or different?

Page 17, Line 543 "...some of the highest rates..." Highest single half-hourly fluxes?

Page 17, Line 543 "especially the deciduous forest)." remove extra full stop.

Page 17, Line 543 What is the definition of a "normal" year? Is this really the conclusion of Griffis et al and Gonsamo et al.? Do they use the term "normal"? Are you surprised that the forests adapted to average site conditions? Before, I read the conclusion that the deciduous forest NEP could profit from comparably dry conditions.

Page 17, Line 548 Statement in first sentence of paragraph is trivial, omit sentence.

Page 17, Line 549 "With insufficient water availability annual tree growth and productivity may be limited". Seems circular to me: When you say insufficient, I suspect you implicitly have in mind that water availability is not sufficient for optimal productivity? To me the sentence says then: When productivity is limited it is limited.

Page 17, Line 555 "ET responds year-round" What do you mean? There is no particularly rainy season?

Page 17, Line 555 "...so warmer spring or autumn periods often lead to annual increases in ET" Warm summer did not impact ET?

Page 18, Line 559 "An opposing ET response..." To what? "...was measured in the coniferous forest" Any idea why?

Page 18, Line 564 "...little summer and annual P removed most of the water from the system, significantly reducing ET" There is no negative precipitation, removal is the wrong term here. The process that (vertically) removes water from the soil is evapotranspiration, why is ET reduced then? Please clarify.

Page 18, Line 565 "timing of summer P" I do not understand, what is meant by timing? Is there only one rain event during summer? Do you mean a peak precipitation event?

Page 18, Line 565 "...the availability of rainfall [...] led to the greatest demand for water" Sorry, I do not get it, consider revising.

Page 18, Line 566 "...differing response" to what?

Page 18, Line 574 "...to respond similarly" to what?

Page 18, Line 577 Is there a reason you picked the forest in Ohio for comparison?

Page 18, Line 578 "..., this implies..." What does "this" refer to. I cannot follow.

Page 19, Line 610 "significant abnormalities were measured between sites" Strange wording, do you mean "differences between sites"?

Page 19, Line 610 "...meteorology was shown to greatly impact fluxes at both sites, though to varying degrees" Either the impact is great or it is sometimes great and sometimes minor (= varying degrees).

Page 19, Line 614 Why "Conversely"? No contradiction to sentence before (which talks about drought years), this sentence about all years. Secondly, NEP is also the result of respiration and photosynthesis at the broad-leaved forest.

Page 19, Line 618 "Both sites saw average ET, but increased NEP during 'normal' years,..." What is the definition of a normal year, 30-year average? What is your base-line for a "normal" NEP? Should be average NEP during average years, shouldn't it? How can NEP deviate (be increased) from the average during an average year then? Please clarify.

Page 19, Line 621 "...while the response of the conifer forest remains uncertain." Sure, there is uncertainty, which is true for the projections about the deciduous forest's sink strength as well. Why not report some of the ideas about conifer forest in a future climate developed before in the discussion?
* * *

---

## Referee Comment (RC2) · Anonymous Referee #2 · 22 Feb 2020

Reviewer #2 recommendation: Return to author for major revisions

General comments

Carbon and water fluxes and their relationships with several environmental variables were presented in this study. The data were collected during the period from 2012-2017 at a conifer and a deciduous forest, which I believe represents an impressive dataset for both forest types. The presented and discussed results showed interesting findings in terms of responses of different forest types to drought and high temperature. If there is a weakness, it is my opinion that it relates to the quantification and identification of the key environmental controls on the fluxes. A significant amount of the manuscript content is dedicated to describing the linear relationships between the fluxes and a series of meteorological variables. However, little regarding the physiological and physical processes is described. In addition, the results and discussions are a bit overwhelming and hard for me to find focuses, which makes the paper seem to lack a truly novel finding. In my opinion, this manuscript should be revised with focuses on: 1) outlining the key/novel science questions; and 2) emphasizing the findings that have clear/great implications in forest management and in ecosystem's responses to future climate. I believe this concern can be addressed through modest revision.

Specific comments

1.-Titile. The leaf-retention and shape strategies are only implied not studied in the manuscript. I suggest changing to a more relevant and accurate title.

2.-Line 16-24. The influences of drought and temperature on NEP and ET are entangled together here, which is a bit unclear. Also, some sentences seem to be repetitive. I suggest rewriting this part of the abstract to make it clearer.

3.-Abstract. Clarify and quantify (if possible) "greatly controlled", "greatly reduced", and "greatly impact".

4.-Line 55-57. Can you add a sentence or two summarizing the previous studies contrasting fluexes coniferous and deciduous forests?

5.-Methods. I noticed the distances of EC relative to the canopy top are different for the two sites. Would the heights of the EC affect the fluxes due to flux divergence or convergence?

6.-Line 57. Is friction velocity a good metric for filtering intermittent turbulence? Previous studies show intermittent turbulence is frequently observed during evening hours at forested sites.

7.-Section 2.3. Have the data been filtered for stationarity?

8.-Section 2.3. The threshold u* seem to be large (0.2 or 0.3 m/s are pretty standard)? Any explanations associated with the sites?

9.-Section 2.3. Add one or two sentences explaining how you processed/averaged the meteorological data.

10.-Section 2.4. Can you describe the uncertainties associated with the approach estimating phenological seasons?

11.-Line 257 and Line 349. Clarify "responded similarly".

12.-Line 255-262. Can you also show the standard deviations of the annual mean Ta in Fig.1?

13.-Line 265. Better explanation for the discrepancies is needed here. The discrepancies are over 300 umol m-2day-1 in spring. Is it in the range of the measurement uncertainty? I'd suggest check the downward PAR to tease out the influences from the canopies and to evaluate the meteorological differences.

14.-Line 267. Clarify "APAR was similar throughout the year". What are the values (mean and standard deviations) of the FPAR mentioned?

15.-Line 281. "Ts(5cm) at TP39 exceeded that of TPD" seems to suggest that the PARgroud at TPD is less, which implies that the APAR at TPD should be higher in summer and autumn. Please explain.

16.-Line 296. Can you explain why 6-year mean day of season growth was used instead of the days of individual years?

17.-Line 327. Could you also add a sentence or two at the beginning of this paragraph to explain the physical meaning of the cumulative (seasonal and annual) fluxes, especially its differences from daily fluxes?

18.-Line 336. "spring was the only season when daily GEP was similar between the forests". As shown in Table 3, the seasonal GEP in spring show larger differences between sites, which I think to some extent contradicts with your statement in Line 336. Please reconcile. Also, when you compare the daily GEP for phenological seasons, how did you address the different lengths of the seasons (i.e. the different number of data points)?

19.-There are a few places where I have similar comments as the previous one. I suggest adding some explanations for the statistical techniques (ANOVA and MANOVA) you used, which would shed some light on the discrepancies. -Line 338. The cumulative GEP in autumn (and 2012, 2014, 2015 summer) is higher at TP39 (except for 2012). Does it contradict the argument in Line 338? -Line 352. "RE was higher at TPD". But the cumulative RE were lower at TPD in spring and autumn. -Line 384. Seasonal ET is more different in spring not autumn. Also, "daily ET" or "seasonal ET"?

20.-Line 339. "the 2016 summer was the only period . . .". Clarify "sufficiently". Also, it seems a false statement to me because summer GEP in 2013 and 2017 are also greater at TPD.

21.-Line 353. Any figure or data to support this statement?

22.-Line 399. How the low WUE in winter is reflected in Figure 6c? Did you only use data from spring to autumn? If so, clarify in the manuscript.

23.-Line 405. Can you clarify "similar results"? The LUE at TPD is 30% higher than that at TP39.

24.-Line 406. Is the annual and seasonal LUE shown in the manuscript? If not, clarify it in the manuscript by adding "(data not shown)". Also, as shown in Table 3, TPD has lower annual GEP, which contradicts with the "greater GEP" referred here. Reconcile.

25.-Line 435. Do you mean "deciduous forest" instead of "conifer"? If not, add the correlation of annual NEP and summer RE for the conifer forest to Table 4. If the

answer is yes, I'd suggest delete this sentence because it conveys the same meaning as the following two sentences.

26.-Line 434-435. Can you add a brief explanation for the relationship of RE and spring Ta.

27.-Line 439-448. The annual GEP has no significant relationships with meteorological variables as stated in Line 425. But this paragraph talks about GEP and meteo controls. Is it only summer GEP discussed in this paragraph?

28.-Line 439. What does "flux parameterizations" mean here? Is it explained in the methodology section? If not, I suggest adding it to the methods section.

29.-Line 578. Is the assumption of similar carbon assimilation valid here given the different NEP?

30.-Table 3. Why the GEP sum for Jan 1 to SOS is missing? They seem to be available in Fig. 3.

31.-Table 4. Can you change this table to a figure similar to Fig. 4? The reasons are (i) you'd be able to show the standard deviations; (ii) the positive/negative correlation would be easier to tell.

32.-Table 5. What model did you use for this calculation?

33.-I notice the uncertainty analysis for measurements and calculations is missing. Can you add a brief subsection to Methods section (or wherever you find appropriate) dedicated to uncertainties?

Minor comments

1.-. I suggest changing all "warm temperatures/Ta" to "high temperatures/Ta" in the manuscript.

2.-Line 78. Clarify "controls". Environmental/meteorological controls?

3.-Line 88-91. Are percentages available for the tree species?

4.-Line 119. Did you use the momentum and heat fluxes in this study? If not, there's no need to mention them.

5.-Line 258. What is the value of "record Ta". Also, "record high Ta".

6.-Line 315. Are the "days 230 to 290" 6-year mean? Explain.

7.-Line 325-326. This statement is not clear. Clarify or delete.

8.-Line 347. Define "outlier".

9.-Line 398. Clarify "the ratio of monthly ET". Then modify the figure caption accordingly.

10.-Line 354-355. Confusing sentence. How do "comparable" results shape the "differences"? Rephrase.

11.-Line 363. "for either site"? It's hard to tell that the monthly NEP is negative at TPD in Figure 5b. Rephrase.

12.-Line 416. P value for being "significant"? "linear relationships of monthly Ta and monthly VPD"?

13.-Be concise. See examples below. -Line 325. "at first glance" is not necessary. -Line 341-342. "significant daily minimums and maximums" seems to be repetitive as "highly variable". -Line 417. Delete ",". -Line 409-410. Delete "and". Also, make the sentence more clear. -Line 372. "the highest" ——-> "highest".

14. Given the different time scales used here, I suggest be more mindful about the uses of "daily, season, annual" when talking about fluxes. -Line 261. In "Ta at both sites", do you mean "daily mean Ta"? -Line 360. Change "The NEP in the conifer…" to "The annual NEP" or "The cumulative NEP". -Line 352. "spring and autumn RE was higher . . .". Do you mean "daily RE in spring and summer"? -Line 410. Delete "When

first considering . . .". Change "ET"—-> "Annual ET". -Line 325. Should "daily patterns" be "seasonal patterns"? Also, subsitute "expanded upon in Table3" with "the cumulative fluxes in Table 3", just to be clear and accurate.

15. I noticed quite a few miscitation or misspelling or inaccurate statements. See some examples below. -Line 270. "daily reductions in PAR (shouldn't it be APAR?)". -Line 401. 4.7 —-> 3.82 gC kg-1 H2O. -Line 406. R2 = 0.96 —-> R2 = 0.86. -Line 535. "increases" ——-> "decreases"? -Line 538. "most years" ——-> "half of the years"? -Line 553. "during drought years" is not accurate. It's really just 2016.

16. I have a few minor comments regarding the tables and figures. See below. -Table 3. Can you highlight the highest and lowest annual fluxes with colored boxes? -Be more clear with figure captions, especially for words like "daily, monthly, seasonal, and annual". For example, "A daily time series" in Fig. 2 is a bit confusing. -Figure 3. Green-red combination is not color-blind friendly. Also, can you annotate SOS, EOG, SOB, and EDS on the top panels? -Figure 4 caption. Two "and".

---

## Author Response (AR1)

**Referee # 1**

**General formal points are:**

- Latin plant names should consistently be printed in italic font. REVISED

1. - Definitions of abbreviations appear repeatedly throughout the text, they should only be introduced on their first occurrence. REMOVED all repetitive abbreviations

2. - The verbs "to experience" and "to respond" are used excessively and sometimes not in the appropriate context. It is clearly a matter of taste but I would advise to revise some sentences. Fluency could partly be improved by language simplification. Removed/modified a number of recurring instances, may need to further edit

3. - Tenses are not always used consistently, please revise (see line comments). REVISED

**Scientific issues:**

4. - "Carbon" is partly used interchangeably with "carbon dioxide". There are more components to the carbon cycle in forests than vertical $CO_2$ exchange. Therefore, sometimes statements are not entirely correct, please review. REVIEWED

5. - Measurement and model uncertainty are not addressed. The authors should add some information on this topic. The ranges of the annual flux sums given in the abstract likely describe inter-annual variability (not measurement/model uncertainty), I assume using mean and standard deviation of the six annual flux sums per forest. An explanation should be added. Added a section in the methods detailing the model uncertainty and confidence intervals for the measurements presented. However, I still kept the standard deviations with the mean values to highlight interannual variability as mentioned.

6. - The description of the used partitioning models (equations 1 and 2) is very concise, at least units should be added. For equation 1 a citation is provided, the short description is defendable. Equation 2, however, is not clearly referenced and therefore definitely needs more explanation. The optimization process of the temperature, VPD and soil moisture functions behind the scaling terms need to be described better, to only mention the sigmoidal shape is not enough in my opinion. Added the necessary citation (Richardson et al., 2007), completely rearranged the entire section, and added an additional equation to better explain the sigmoidal functions used within the partitioning models.

7. - Some of the conclusions about the effects of drought rely on the analysis of the especially warm and dry year 2012. The fact that there was a disturbance (thin cutting) in one of the forests in this year is not discussed comprehensively enough. The authors should for example include the effect of a diminished leaf area on $CO_2$ exchange fluxes in their interpretation of this (and the

next?) year's budget and explore if for the interpretation of the data set from 2012 to 2017 post disturbance effects should be considered. Included additional information in the site information and methods which highlighted the past findings at the site in regards to the thinning/disturbance

***Line comments:***

8. Page 1, Line 3 (Title) "similar-age" should not be hyphenated. REVISED

9. Page 1, Line 19 I would suggest replacing the somewhat complicated sentence ", ... the evergreen forest saw greater annual reduction" with e.g. "..., net $CO_2$ uptake was reduced more at the evergreen forest than at the deciduous forest." However, during warm and dry years, the evergreen forest had largely reduced annual NEP values compared to the deciduous forest.

10. Page 1, Line 22 "Annual ET was driven by changes in air temperature" Are you sure? Is T change really the driver? It sounds like the slope of a T change determines ET. If so, which timescale do you refer to? Maybe average temperature actually is the driver? Variability in annual ET at both forests was related most to the variability in annual air temperature (Ta), with the largest annual ET observed in the warmest years in the deciduous forest.

11. Page 1, Line 23 "During drought years, …" It is a bit hard to follow the logic. The preceding sentence says that dry periods greatly reduced ET at the deciduous forest. Now it is stated that the sensitivity of ET to temperature changes (?) at the deciduous forest is comparably low. Maybe say: ET is sensitive to dry periods/increased T at both sites. ET reduction at TP39 is comparably larger. Additionally, ET was sensitive to prolonged dry periods that reduced ET at both stands, although the reduction at the conifer forest was relatively larger than that of the deciduous forest.

12. Page 1, Line 25 "If longer periods..." Longer than what? Can you give us an idea about time scales? If prolonged periods (weeks to months) of increased Ta and reduced precipitation are to be expected under future climates during summer months…

13. Page 1, Line 26 "...the carbon sink capacity [...] will continue." is a bit complicated. Maybe "...will continue to act as a sink..." "...while that of..." is not very elegant, consider reformulating. … the deciduous broadleaf forest will likely remain an annual carbon sink, while the carbon sink-source status of the coniferous forest remains uncertain.

14. Page 1, Line 29 Remove comma before "through". "Absorption of CO2 emissions" can be replaced by "CO2 uptake". REVISED

15. Page 1, Line 30 remove "processes". REVISED

16. Page 2, Line 38 I would add "events" after "extreme weather". Remove "stress". Stress is the consequence of extreme weather not an example for an extreme event. REVISED

17. Page 2, Line 39 "Adversely impacting [...] forest–atmosphere interactions" What does that mean? Sounds like there is no interaction anymore due to extreme weather, you clearly do not mean that. Also: replace hyphen with en-dash in expression "forest–atmosphere". REVISED … forests to sequester carbon, and thus regional forest–atmosphere interactions

18. Page 2, Line 40 The authors state that there are positive and negative feedbacks but give only an example for a process leading to a positive feedback. Example for opposite case? Had thought to mention enhanced $CO_2$ leading to partial stomatal closures and reduced water loss leading to possible cooling, but REMOVED feedback sentence instead

19. Page 2, Line 46 I do not get the reasoning. "The result of a shifting climate..." [which result?] impacts both forest types differently because broad-leaved species are replaced by needle-leaved species? I do not understand the cause-effect concept behind the statement, consider revising. REVISED beginning of the paragraph. However, climate change will impact deciduous and coniferous forest ecosystems differently due to their physiological differences.

20. Page 2, Line 50 I assume you refer to a disturbance of regional cycles and not within forest cycling, can be formulated more clearly. REMOVED sentence

21. Page 2, Line 51 "Conversely," I do not see an opposition to the previous statement, which is about photosynthetic rate. This sentence talks about season length. REVISED sentence but ultimately removed conversely

22. Page 2, Line 58 "...have the ability to conduct research..." is needlessly convoluted. Consider replacing with e. g. "Few studies have reported multi-annual time series." Also: omit "sufficiently long". Otherwise you need to explain which timescale would be sufficient. Even fewer studies have reported multi-annual time series

23. Page 2, Line 59 In my opinion, there is no need to construct ("Such a study would...") the need for the current study. I would omit lines 59 to 63 and go straight to Page 3, Line 73 ("This study..."). REMOVED suggested section

24. Page 2, Line 61 The "benefit" of forests to "terrestrial–atmosphere gas exchange" seems vague. Gas exchange takes place anyway, there only is a benefit if you prescribe a service of forests (e. g. carbon sink function), which is not mentioned here. As stated before, I would omit the whole section. REMOVED suggested section

25. Page 3, Lines 64-69 Should be moved to section 2.1 (Study sites) REMOVED

26. Page 3, Lines 70 to 73 As no results of the previous studies are mentioned here, listing them is not very informative. I would move this section to the results or discussion section and mention the results of previous studies there in comparison/relation to the current study. REMOVED

27. Page 3, Line 80 "will be used". The choice of tense in confusing to me. Starting in line 73, present tense is used, future here. Sentence was REMOVED

28. Page 3, Line 83 What is "natural terrain"? REMOVED the word natural

29. Page 3, Line 83 "The forest is classified". By whom? Is there a citation or a classification system this assumption refers to? REMOVED. The forest is unevenly aged

30. Page 3, Line 91 "Conifer species including make-up..." Sentence incomplete.
REVISED Conifer species only account for a minor component…

31. Page 4 Line 106 Personally, I do not like the frequent use of the verb "experience". For this sentence a simpler way could be: "While edaphic and climatic conditions are similar between both sites, they differ in vegetation cover and canopy structure." REVISED
While edaphic and climatic conditions are similar between both sites, they differ in vegetation cover and canopy structure and physiology.

32. Page 4 Line 107 What do you mean by "historically defined"? That past events (ice age) shaped the landscape or that authors in the past defined the landscape like this? These sandy soils are part of the Southern Norfolk Sand Plains, an area shaped by past ice age glacial melt processes.

33. Page 4 Line 109 It would be easier for international soil scientists to understand if the name according to the FAO World Reference Base would be given additionally to the name according to the national Canadian system. … Canadian Soil Classification Scheme and FAO World Reference Base as Brunisolic grey-brown luvisol and Albic Luvisol/Haplic Luvisol, respectively

34. Page 4, Line 113 "Help" is not ideal. How does the lake control cold temperatures?
REMOVED sentence – moderating effect of water body

35. Page 4, Line 114 "...were 8 ◦C and..." past tense? The mean is still the mean. Next sentence present tense again. REVISED is 8.0 ± 1.6°C and 997 mm

36. Page 4, Line 116 The citation is incomplete. Based on the information provided, the given data cannot be verified. Updated the citation and included a link to the website

37. Page 4, Line 116 Last sentence of paragraph can be omitted, it is poorly formulated. Information also given in "Data availability" section. REMOVED

38. Page 4, Line 121 Omit ", though". Start new sentence with "Measurements". REVISED Measurements at both sites are still ongoing

39. Page 4, Line 124 Supplementary material would be a separate pdf-file, I think. Table A1 seems to be in the appendix. REVISED …are outlined in the appendix (Table A1).

40. Page 4, Line 124 "...are calibrated". Present tense? Paragraph starts in present perfect (..."have been measured"). At both sites, IRGAs are calibrated monthly using high purity $N_2$ gas for the zero offset. Measurements at both sites are still ongoing/being calibrated.

41. Page 4, Line 125 The expression "Environment Canada Greenhouse gas specified CO2" is not understandable. Which concentration did the span gas have? At both sites, IRGAs were calibrated monthly using high purity N2 gas for the zero offset and $CO_2$ gas (360 $\mu$mol mol$^{-1}$ $CO_2$; following WMO standards) for the $CO_2$ check.

42. Page 5, Line 127 It comes as a surprise that there is more than one IRGA per EC setup. In line 123 singular was used ("...an IRGA"). I would stress this type of setup more as it is typical and necessary for forest EC. Half-hourly net ecosystem exchange (NEE, $\mu$mol m$^{-2}$ s$^{-1}$) is calculated as the sum of the vertical $CO_2$ flux ($F_c$), and the rate of $CO_2$ storage ($S_{CO2}$) change in the air column below the IRGA ($NEE = F_c + S_{CO2}$).

43. Page 5, first two paragraphs A mixture of tenses is used. "is completed", "were assumed", "have been conducted", "are measured", "will focus". Check for consistency. REVISED

44. Page 5, Line 139 Unclear what "Environment Canada Delhi CDA" is. Why mention if precipitation data is not used after all (as stated in line 141)? REMOVED … P data from an accumulation rain gauge (T-200B, GEONOR) installed 1 km south of TP39

45. Page 5, Line 145 What is the difference between quality control, filtering and cleaning? If you do not want to go into detail just mention the citation and say e. g. "processed as described by Brodeur (2014)". REVISED entire section. All meteorological and flux data were processed on lab-developed software following the FluxNet Canada Research Network (FCRN) guidelines as described by Brodeur (2014).

46. Page 5, Line 147 How was the frequent cross-checking with AmeriFlux done? Statement seems vague. Sentence REMOVED – not very frequently

47. Page 5, Line 148 How were outliers identified? A two-step cleaning process was used to remove outliers in half-hourly meteorological data: coarse upper and lower thresholds were applied to half-hourly values to remove obvious outliers, and additional erroneous half-hourly data were removed from time series when instruments were known to be malfunctioning or visual

inspection by multiple reviewers resulted in certain agreement that an outlier was present. Added citation of Papale et al. (2006).

48. Page 5, Line 150 There are other EC towers at Turkey Point Observatory? Where are they? Can you expect them to be representative for your site? Only then using them to gap-fill your data would make sense. More information needed. Missing meteorological data of all lengths were gapfilled using extant data for the same half hours from either (in order of preference) a second sensor at the site, or an equivalent sensor from a nearby (1-3 km away) station in the network (sites described in Peichl et al., 2010).

49. Page 5, Line 150 What is "mean flux recovery"? Percentage of half-hourly measurements left after filtering? Including or excluding times of instrument maintenance/malfunction? Yes, the mean flux recovery was the data remaining after all the filtering processes, including data lost from the start (i.e. maintenance and malfunctions). The resulting final mean flux data recovery following both threshold filtering methods

50. Page 5, Line 159 Omit "where daytime and nighttime"; it means all fluxes, correct? No need to specify then. REMOVED

51. Page 6, Line 160 It is stated that filtered NEE was gap-filled using soil temperature. Why is "flux recovery" after gap-filling only 49 %. Check if this gap-filling step was actually applied. It seems unlikely. Later more complicated methods for flux partitioning and gap-filling are described. The simple NEE-Ts model seems redundant. REMOVED the sentence. Flux recovery would be before any gap-filling just filtering.

52. Page 6, Line 164 Symbol for soil moisture appears here first. Explanation too late in line 163. Introduced the soil water content in Line 119 when discussing met measurements.

53. Page 6, Line 164 Partitioning of NEE into GEP and RE has not been introduced. The reader does not know the RE time series at this point. If you talk about gaps in it you have to introduce it first. NEE gap-filling and its partitioning into components of ecosystem respiration (RE) and gross ecosystem productivity (GEP) were achieved using the methods described in Peichl et al. (2010a), which are summarized below.

54. Page 6, Line 166 What is the definition of nighttime? A radiation threshold? Yes, radiation threshold… nighttime (PAR < 100 $\mu$mol m$^{-2}$ s$^{-1}$) fluxes

55. Page 6, Line 166 It is stated that nighttime NEE was modeled as a function of soil temperature and moisture in order to (!) describe the relationship of RE and Ts which represents diurnal air temperature variability. Check meaning of the sentence. It seems incoherent to use nighttime measurements to describe diurnal variability of something. Sentence was removed and preceding paragraph modified (edit shown above)

56. Page 6, Line 173 What are the units of the model parameters, especially of a1 and a2? a1 and a2 are not a function of soil moisture (as stated) when looking at equation 1. I assume all four parameters were fit during the same optimization process. Added an additional equation to better explain everything (Equation 2). where $a_1$ and $a_2$ are fitted parameters that describe a sigmoidal curve that ranges from 0 to 1 (Richardson et al., 2007). In this approach, the $Ts_{5cm}$ component of the function defines a theoretical maximum half-hourly respiration rate based on soil temperature (i.e. driving variable), while the $\theta_{0\text{-}30cm}$ component modulates the resultant predicted value as a function of the volumetric water content (i.e. scaling variable).

57. Page 6, Line 173 "...acting to scale the RE relationship" to what? REMOVED (above)

58. Page 6, Line 180 Explanation of equation 2 needs more detail. How are these sigmoidal functions set up? Do they have parameters? Parameters optimized at the same time? The remaining terms use the functional form introduced in Equation 2 to described the responses of GEP to Ts, vapor pressure deficit (VPD), and $\theta_{0\text{-}30cm}$, respectively.

59. Page 6, Line 187 Seems inconclusive. Don't you need the modeled GEP time series in order to calculate phenologically-derived summer months? For the GEP model you in turn need the derived summer months. Please explain. No change made. Phenologically-derived summer months and all phenologically-modelled periods were found using 'non-gapfilled GEP' (only periods where non-gapfilled NEE matched gap-filled NEE).

60. Page 6, Line 189 Sentence starting with "Furthermore..." ending in line 191 with "both sites" can be omitted, unnecessary/circular information. Yes, in the growing season plants grow, therefore it is a key season of CO2 uptake. REMOVED

61. Page 7, Line 201 Omit first sentence of paragraph, contains no new information. REMOVED sentence

62. Page 7, Line 208 "water or heat stressed periods", check meaning, the periods are not under stress. … during low water or high heat periods

63. Page 7, Line 210 Contents of last paragraph can be moved to results, stays a bit vague here anyway. Section mostly REMOVED. Added an ANOVA/t-test sentence in results

64. Page 8, Line 237 GEP might not be gap-filled, still it is not direct measurement data but modeled as the difference of RE (modeled) and NEE (=EC Fc, measured). Could be stressed here, it took me a while to get my head around this fact. From half-hourly non-gapfilled data (calculated as the difference between modeled RE and measured non-gapfilled NEE), the maximum daily photosynthetic uptake (GEPMax) was calculated.

65. Page 8, Line 240 The approach does not calculate, the computer calculates according to the approach. This approach identified photosynthetic transition dates

66. Page 8, Line 241 "logistic curve" instead of "logistics curve" REVISED

67. Page 8, Line 241 "The second derivative estimated the end of greenup..." How? Time when derivative turns zero or similar? The local minima of the second derivatives estimated the end of greenup (EOG), the length of canopy closure (LOCC), and the start of browndown (SOB), while the local maxima of the third derivatives estimated the start of the growing season (SOS), and the end of the growing season (EOS).

68. Page 8, Line 242 "while the third derivatives calculated..." see two comments above. REVISED

69. Page 8, Line 251 accumulation REVISED

70. Page 8, Line 257 Ta responds to what? behaved similarly

71. Page 8, Line 258 "Record warm Ta conditions". Expression unclear to me. Annual mean above 30-year average? Most days/half hours above 30-year average of corresponding DOY/half hour? Added (exceeding 30-year mean daily maximum values)

72. Page 8, Line 260 What does extreme mean? What does "magnitude of extreme cold days" mean exactly? Added (exceeding 30-year mean daily minimum values)

73. Page 9, Line 261 "record Ta outside the normal peak summer period" Unclear, what does record and normal mean? Temperature is outside the period? Check meaning of sentence. with record Ta outside of the typical summer (June – August) period

74. Page 9, Line 262 The sites are not growing, the vegetation is. REVISED

75. Page 9, Line 263 "Meteorological conditions between the sites were [...] examined". Check meaning. Consider replacing with "Differences in meteorological conditions between the sites were examined" or "Meteorological conditions at both sites were examined" REVISED

76. Page 9, Line 263 "..., beginning with..." Sequence of analysis steps not relevant. REMOVED

77. Page 9, Line 264 Sentence "However, the shapes..." is circular and can be omitted. It says: The seasonal course of APAR depicts the course of absorbed PAR, meaning APAR is APAR. REMOVED

78. Page 9, Line 267 "APAR was similar throughout the year". Not true, see figure 2. Relative quantity FPAR might be about constant during annual course, APAR is not. Updated the methods/figure to describe why the sites show similar APAR measurements

79. Page 9, Line 270 Cloudy conditions along with a reduction in incoming radiation are not a coincidence. Daily reductions in PARdn and APAR often resulted from cloudy conditions and precipitation (P) events (Fig. 2a).

80. Page 9, Line 278 Could replace "followed closely to Ta" with "follow Ta closely" REVISED

81. Page 9, Line 281 replace "of TPD" with "at TPD". REVISED

82. Page 9, Line 282 replace "similar patterns between sites" with "similar patterns at both sites" REPLACED

83. Page 9, Line 283 Soil moisture deficit compared to what? At which value does it start to be deficient? …with prolonged summer θ declines in 2012, 2016, and 2017 (Fig. 2f). Changed to declines instead of deficits so there's no specific threshold

84. Page 9, Line 283 "In summer" comma missing REVISED

85. Page 9, Line 285 "while all other times of the year TP39 was higher". Soil moisture was higher not TP39. Yes. during all other times of the year θ at TP39 was higher (Fig. 2g).

86. Page 9, Line 292 Consider replacing unit "day" with unambiguous "day of year (DOY)" throughout manuscript, first occurrence here. Replaced all cases of 'day' with DOY

87. Page 10, Line 295 Check meaning. "The response [...] to changes in GDD was considered as a trigger for SOS." The response is the trigger? I think GDD change is the trigger and the response of the forest to this trigger manifested in SOS. The response of the forest to increasing GDD was shown to be a trigger for the SOS.

88. Page 10, Line 296 "cumulative GDD" GDD is cumulative by definition, is it not? Total

89. Page 10, Line 297 Cumulative heat is not expressed directly in GDD, GDD is a proxy for absorbed heat as correctly stated above. I would omit the half-sentence "However, [...], which we calculated as" REMOVED

90. Page 10, Line 299 "represented" not anymore? check tense. Represents - REVISED

91. Page 10, Line 303 replace "start" with "are reached" REVISED

92. Page 10, Line 314 Omit first sentence of paragraph, it is a bit vague. "influenced by a certain degree of cooling"? REMOVED

93. Page 10, Line 316 replace "were found to be highly correlated" with "were highly correlated" REVISED

94. Page 10, Line 325 "At first glance..." Sentence seems vague. What do you mean by similar? Which properties of the forests responded similar to which forcings? What does "seasonal irregularities" mean? Difference between same season of different years or within one year between seasons? How do these irregularities govern annual fluxes (cumulative fluxes?). Highest contribution to sum during periods when forcings deviate from average behavior? Consider restructuring or omitting sentence. REMOVED The water (evapotranspiration) and carbon (photosynthesis and respiration) fluxes were analysed in both forests from 2012 to 2017, with the seasonal patterns of these fluxes illustrated in Fig. 3 and cumulative fluxes in Table 3.

95. Page 11, Line 327 replace "within" with "at" REVISED

96. Page 11, Line 337 "...did not greatly benefit the forest..." seems unassertive. What do you mean? No increase in $CO_2$ uptake? If the latter is meant, I would question the statement. Sure, when you look at average daily GEP, a longer spring increases n for the conifer forest and adds mostly low values (from earlier in the year) lowering the average. Looking at spring GEP/NEP sums might lead to a different interpretation. In all 6-years, spring was the only season when daily GEP was similar between the forests, as the advancement of SOS at TP39 did not statistically benefit carbon uptake due to seasonal meteorological conditions (i.e. low PAR, Ta, etc.) acting to limit photosynthesis.

97. Page 11, Line 339 Details about statistical tests could be inserted here. I am not sure what the p-value refers to, a t-test? Added a sentence describing tests. Using the analysis of variance (ANOVA) technique, t-tests were completed to evaluate statistical differences between the two groups (i.e. deciduous broadleaf vs. evergreen needleleaf) of data.

98. Page 11, Line 341 I would replace "minimums/maximums" with "minima/maxima", might be a matter of taste. Sentence removed at advice of reviewer/comment below

99. Page 11, Line 341 How is a maximum significant? Consider removing. REMOVED

100. Page 11, Line 342 RE was modeled not measured. greatest annual RE was found…

101. Page 11, Line 344 replace "let the year to have" with "led to" REVISED

102. Page 11, Line 346 see previous comment REVISED

103. Page 11, Line 349 response to what? Updated to behaved similarly

104. Page 11, Line 351 check meaning. Ta always high between rain events? In both cases, maximum rates of RE and ET occurred following precipitation events, as the soil was sufficiently wet, helping to promote ET and enhance RE through respiration pulses (Misson et al., 2006).

105. Page 12, Line 363 Should it be "sink" instead of "source"? REVISED

106. Page 12, Line 369 Check meaning. "NEP [...] exceeded TP39" Following SOS, daily NEP at TPD exceeded that at TP39 in all years except 2015 ($p < 0.01$).

107. Page 12, Line 385 "to" missing, should be "let to rates" REVISED

108. Page 12, Line 387 Consider replacing "deviations" with "variability expressed as standard deviation" and omitting the plus-minus sign in brackets. REVISED

109. Page 12, Line 391 "WUE varied [...] due to different [...] overall GEP and ET". Check statement, seems circular to me. Does it say: "The ratio of GEP and ET varies because GEP and ET vary"? REMOVED sentence

110. Page 13, Line 394 "..., the SOS began..." Reformulate, now it says "the start began" In 2016, an early SOS (March 15; DOY 74) promoted prompt increases in spring GEP, when Ta and ET remained low.

111. Page 13, Line 396 remove "forest" REMOVED

112. Page 13, Line 400 Same number for TPD and TP39. REVISED

113. Page 13, Line 405 monthly GEP and APAR sums or averages? Mean monthly

114. Page 13, Line 409 Sentence incomplete. "To better understand and the water...." Meteorological variables (i.e. Ta, PAR, $\theta$, etc.) were analysed during the study period to better understand their impact on water and carbon fluxes within each forest.

115. Page 13, Line 410 remove "first". Sequence of analysis steps irrelevant. REMOVED

116. Page 13, Line 412 "the impact of winter soil water storage..." on what? A smaller secondary effect on ET ($R^2 = 0.83$; Table 4) was found for winter and early spring (January 1st to SOS) $\theta_{0\text{-}30cm}$, which helped to explain the impact of winter soil water storage and seasonal water availability on ET at the start of each year.

117. Page 13, Line 419 Consider reformulating "responses between". I would expect "the response of something to something else" REVISED to The response of monthly ET to monthly VPD was similar between sites

118. Page 13, Line 425 Maybe there is no linear relationship between GEP and meteorological variables. There should, however, definitely be relations with PAR. As far as I understand GEP was modeled using PAR, you should see the saturation curve you prescribed in the model (eqn. 2) in a PAR-GEP plot. You are correct. I believe here it's only considering the annual values, so no annual relationship between PAR and GEP.

119. Page 13, Line 426 There is an extra space after the closing bracket and "resulted" REVISED

120. Page 14, Line 429 Why "most importantly"? Mean or cumulative summer NEP? REMOVED most importantly …cumulative summer NEP ($R^2 = 0.99$).

121. Page 14, Line 431 "was seen" is not very elegant. Consider simplifying the sentence, e.g. "...spring was shorter due to..." For the evergreen conifer site, spring was shorter in years with the highest annual NEP due to rapid photosynthetic development.

122. Page 14, Line 431 "Higher summer Ta". Season average or half-hourly or daily? Higher mean summer Ta decreased annual NEP, highlighting the influence of limitations due to heat stress

123. Page 14, Line 434 "relationship between RE and spring Ta". timescales? annual RE, spring RE, sums or averages? At the deciduous forest, the relationship between annual RE and spring Ta ($R^2 = 0.77$) suggested that warmer springs generally acted to decrease annual RE.

124. Page 14, Line 437 "Lastly,...", "Ultimately,..." can be omitted. Sequence of analysis irrelevant. REMOVED

125. Page 14, Line 438 They sites do not emphasize, you do. REVISED – Highlighted

126. Page 14, paragraph starting in line 439 This paragraph requires more explanation. How were the model parameters examined? The methods section is not detailed enough about this type of analysis, Table 5 is also ambiguous ("GPP:Ta" sounds like correlation analysis. Should it be f(Ta) as in eqn. 1 to denote that the scaling factor is meant?). The scaling method is very interesting, it deserves a proper explanation for others to be able to reproduce it. Added a section in the methods: 2.4 Estimating effects of meteorological variables on carbon component fluxes. This helps to better explain the modeling and parameterization of the data outlined in the paragraph and Table 5.

127. Page 14, Line 445 "Outside of Ts". Sounds strange to me. Do you mean "apart from"? Yes. \Aside from $Ts_{5cm}$, $\theta_{0\text{-}30cm}$ impacted summer RE at both sites.

128. Page 14, Line 447 Similar response of what to what? REVISED – similar trends

129. Page 14, Line 448 What do you mean with "predicted daily rate"? The observed fluxes were the result of a prediction? I do not understand, consider clarifying. Overall, the annual fluxes were a product of the season length and the estimated daily rates of the $CO_2$ fluxes that were in turn influenced by seasonal variability in meteorological variables.

130. Page 14, Line 451 replace "experienced by" with "at" REVISED

131. Page 14, Line 451 Typical meteorological conditions? Introduction says air temperature was consistently above the 30-year average. The meteorological conditions at both sites during the study period were characteristic of temperate North American forest ecosystems, characterized by four distinct seasons, with cold winters and warm summers.

132. Page 14, Line 455 "certain differences were primarily influenced" is a bit vague, which difference, why primarily. What about relief position, water content or soil type? Even with similar climatic forcings (i.e. Ta) seasonal deviations in $Ts_{5cm}$ were found, likely influenced by the opposing forest canopy characteristics

133. Page 14, Line 456 "In this case" Soil temperature is always linked to incoming radiation. REMOVED 'in this case'

134. Page 14, Line 457 Mean Ts or each half-hourly value? In all years, mean daily $Ts_{5cm}$ at the conifer forest was higher during each summer

135. Page 15, Line 459 What does "highly clumped" mean? High compared to what? In the conifer forest, branches and needles were closely clumped... highlighting that conifer canopies show less ability to fill canopy gaps, instead driven by shape.

136. Page 15, Line 459 Minor variations in APAR? Maybe true for fPAR, looking at Figure 2 APAR seems highly variable throughout an annual course. REVISED to fPAR

137. Page 15, Line 461 "Incoming radiation was directly absorbed by the soil" All of it? What about LE etc.? Not all energy goes into ground heat flux. In the deciduous forest, $Ts_{5cm}$ was higher when leaves were absent and a higher fraction of incoming radiation was directly absorbed by the soil.

138. Page 15, Line 464 Incomplete sentence. "...similar trends VPD..." similar VPD trends

139. Page 15, Line 469 "species specific responses shaped the timing of phenological events" Responses to what? Isn't it obvious that species type determines phenology? REMOVED

140. Page 15, Line 480 There is only one SOS per year. How can SOS have high variability in a warm year when there is only one value per year? …variability (between years)

141. Page 15, Line 486 Seems contradictory. Either timing of senescence and soil moisture are not related ("insignificant") or the forests experienced "later senescence dates with decreased soil moisture". If the finding opposes previous studies it would be interesting to read about possible reasons (water stress?). Both forests experienced later senescence dates with decreased θ (although likely due to increased Ta). For the conifer forest, the two years (i.e. 2012 & 2016) with continued heat and drought stress saw the latest dates of senescence, while at the deciduous forest, greater mean summer θ led to earlier senescence in all years but decreased θ extended senescence.

142. Page 16, Line 496 replace "in the deciduous site occurred a month (31 days) before that of the evergreen..." with "at the deciduous site occurred one month (31 days) earlier compared to the evergreen..." REVISED

143. Page 16, Line 497 omit "experienced" REVISED

144. Page 16, Line 500 "only limited by their specific leaf strategy". **This seems to be a major argument (Title!)**. Can you expand more, why "only" limited by this strategy? … season length from prolonged autumns, limited by their specific leaf-strategy. But ultimately decided to change the title to not include 'leaf strategy'

145. Page 16, Line 503 "Ta anomalies [...] strongly determine the carbon sequestered". Check meaning. Ta determines the carbon? Maybe the amount of carbon? Are you sure the anomalies determine C uptake as opposed to the average temperature? REVISED Anomalous Ta (extreme heat or cold) and seasonal fluctuations in water availability (θ) over a predictable course of the year were shown to strongly impact the carbon sequestered in many forests.

146. Page 16, Line 505 ",... higher Ta..." Anomalies, average, min/max? higher mean Ta

147. Page 16, Line 506 "drawback" only if maximum sink strength is the goal. why judge? Conceptually, higher mean Ta will promote longer growing seasons and greater GEP, though increased RE may also be expected

148. Page 16, Line 507 typo: "differing forest[s] responses" REVISED

149. Page 16, Line 508 "season length in 2012 was the second shortest despite..." Maybe there is another factor co-controlling season length then? It's definitely possible. The determination of the phenological dates and the growing season length were modeled from GEP data, which was reduced in 2012 at both sites as a result of drought.

150. Page 16, Line 509 Maybe not "despite" but "because" high air temperatures. There could be a temperature optimum (parabolic function) for GEP. What does "record Ta" mean? Daily/Half-hourly maximum, mean, average above long-term average? At both sites, the overall growing season length in 2012 was the second shortest (behind 2014), as a result of the anomalously warm Ta experienced throughout much of the year.

151. Page 16, Line 510 Why "also"? Section already talks about outlier year 2012. REMOVED

152. Page 16, Line 512 "due to thinning performed..." Definitely! This fact is introduced too late. Such a disturbance could single-handedly be responsible for budget deviations in 2012 and override all possible reasons stated before. The disturbance must be stressed and discussed more and earlier. Introduced the thinning and management in the methods

153. Page 16, Line 513 "higher Ta and low theta" Annual/seasonal mean or each/most half hours/days? Replace "acted to enhance" with "enhanced" Additionally, higher daily mean Ta and low θ enhanced RE in the conifer forest, but significantly reduced RE in the deciduous forest.

154. Page 16, Line 525 replace "due to comparable decreases" with "due to comparably large decreases" REVISED

155. Page 17, Line 535 "very similar NEP" at both sites vs. Page 17, Line 538 "led the conifer forest [...] to have a greater magnitude of annual NEP". Is NEP similar or different? In all years the magnitude of GEP and RE were greater in the conifer forest, however, analogous reductions at the deciduous forest led the two forests to have very similar mean annual NEP (despite large annual differences).

156. Page 17, Line 543 "...some of the highest rates..." Highest single half-hourly fluxes? REVISED – highest daily rates

157. Page 17, Line 543 "especially the deciduous forest)." remove extra full stop. REMOVED

158. Page 17, Line 543 What is the definition of a "normal" year? Is this really the conclusion of Griffis et al and Gonsamo et al.? Do they use the term "normal"? Are you surprised that the forests adapted to average site conditions? Before, I read the conclusion that the deciduous forest NEP could profit from comparably dry conditions. Yes, they use the term normal, which was edited here to better explain the thought process. This suggests that both forests favor meteorologically “normal” years (comparable to the 30-year mean meteorological conditions), equivalent to the conclusion of Griffis et al. (2003) and Gonsamo et al. (2015). Therefore, under future climates, which are predicted to be warmer compared to the current 30-year norm for the

area, the carbon sequestration capacity of both forests may be reduced, although to a lesser effect at TPD.

159. Page 17, Line 548 Statement in first sentence of paragraph is trivial, omit sentence. REVISED

160. Page 17, Line 549 "With insufficient water availability annual tree growth and productivity may be limited". Seems circular to me: When you say insufficient, I suspect you implicitly have in mind that water availability is not sufficient for optima productivity? To me the sentence says then: When productivity is limited it is limited. REMOVED

161. Page 17, Line 555 "ET responds year-round" What do you mean? There is no particularly rainy season? Much like RE, ET responds year-round (with summer maxima), so warmer spring or autumn periods often lead to annual increases in ET

162. Page 17, Line 555 "...so warmer spring or autumn periods often lead to annual increases in ET" Warm summer did not impact ET? Yes it did, outside of the summer maxima

163. Page 18, Line 559 "An opposing ET response..." To what? "...was measured in the coniferous forest" Any idea why? A contrasting ET response was measured in the coniferous forest. The deciduous forest measured increased ET during the hot/dry year of 2016, but it was too dry at the conifer forest, leading to an opposite response

164. Page 18, Line 564 "...little summer and annual P removed most of the water from the system, significantly reducing ET" There is no negative precipitation, removal is the wrong term here. The process that (vertically) removes water from the soil is ET, why is ET reduced then? Please clarify. In our case, high summer Ta, the lowest $\theta_{0\text{-}30cm}$, and very little summer and annual P (input) into the system, significantly reduced ET, while RE continued to rise.

165. Page 18, Line 565 "timing of summer P" I do not understand, what is meant by timing? Is there only one rain event during summer? Do you mean a peak precipitation event? At the conifer forest, the timing of summer P events appeared to influence ET (i.e. 2013)

166. Page 18, Line 565 "...the availability of rainfall [...] led to the greatest demand for water" Sorry, I do not get it, consider revising. REMOVED

167. Page 18, Line 566 "...differing response" to what? opposing responses of ET to $\theta$

168. Page 18, Line 574 "...to respond similarly" to what? We found the course of annual WUE of both forests to respond similarly across all years

169. Page 18, Line 577 Is there a reason you picked the forest in Ohio for comparison? The Ohio forest was used as WUE was researched in a regionally local oak-dominated forest

170. Page 18, Line 578 "..., this implies..." What does "this" refer to. I cannot follow. Assuming similar daily rates of carbon assimilation (GEP), higher WUE implies a higher evapotranspiration flux at the conifer forest (Augusto et al., 2015), which we saw.

171. Page 19, Line 610 "significant abnormalities were measured between sites" Strange wording, do you mean "differences between sites"? Yes, REVISED

172. Page 19, Line 610 "...meteorology was shown to greatly impact fluxes at both sites, though to varying degrees" Either the impact is great or it is sometimes great and sometimes minor (= varying degrees). REMOVED greatly. Summer meteorology was shown to impact fluxes at both sites

173. Page 19, Line 614 Why "Conversely"? No contradiction to sentence before (which talks about drought years), this sentence about all years. Secondly, NEP is also the result of respiration and photosynthesis at the broad-leaved forest. The annual NEP at the conifer forest was ultimately shaped by total summer NEP.

174. Page 19, Line 618 "Both sites saw average ET, but increased NEP during 'normal' years..." What is the definition of a normal year, 30-year average? What is your baseline for a "normal" NEP? Should be average NEP during average years, shouldn't it? How can NEP deviate (be increased) from the average during an average year then? Clarify.
Both sites saw average ET, but increased NEP (against the 6-year study mean) during climatologically (30-year mean) 'normal' years, but only the conifer forest saw annual reductions in carbon sequestration during drought years.

175. Page 19, Line 621 "...while the response of the conifer forest remains uncertain." Sure, there is uncertainty, which is true for the projections about the deciduous forest's sink strength as well. Why not report some of the ideas about conifer forest in a future climate developed before in the discussion? We also found that drought-induced RE increases or GEP decreases may impact the overall net carbon uptake in the coniferous stand. Our study suggests that the deciduous forest will continue to be a net carbon sink under increased temperatures and larger variability in precipitation under future climate changes, while the response of the coniferous forest will continue to remain uncertain.

**Referee # 2**

***Specific comments:***

1. -Title. The leaf-retention and shape strategies are only implied not studied in the manuscript. I suggest changing to a more relevant and accurate title. Title changed to: Response of carbon and water fluxes to meteorological and phenological variability in two Eastern North American forests of similar age but contrasting species composition – a multiyear comparison

1. -Line 16-24. The influences of drought and temperature on NEP and ET are entangled together here, which is a bit unclear. Also, some sentences seem to be repetitive. I suggest rewriting this part of the abstract to make it clearer. REVISED
Summer meteorology greatly impacted the carbon and water fluxes in both stands, however the degree of response varied among the two stands. In general, warm temperatures caused higher ecosystem respiration (RE), resulting in reduced annual NEP values – an impact that was more pronounced at the deciduous broadleaf forest compared to the evergreen needleleaf forest. However, during warm and dry years, the evergreen forest had largely reduced annual NEP values compared to the deciduous forest.

3. - Abstract. Clarify and quantify (if possible) "greatly controlled", "greatly reduced", and "greatly impact". Updated the abstract so most uses of greatly were removed

4. -Line 55-57. Can you add a sentence or two summarizing the previous studies contrasting fluxes coniferous and deciduous forests? Ultimately reduced the focus on the previous studies in the revised introduction. Mentioned a few differences in past sentences.

5. -Methods. I noticed the distances of EC relative to the canopy top are different for the two sites. Would the heights of the EC affect the fluxes due to flux divergence or convergence? Following the assumptions that we are above the canopy roughness layer in each forest, and we're footprint-filtering appropriately, we don't think there is an effect.

6. -Line 157. Is friction velocity a good metric for filtering intermittent turbulence? Previous studies show intermittent turbulence is frequently observed during evening hours at forested sites. No, it's not. It should be paired with stationarity tests, to make it more appropriate. We also calculate the storage change as a means of capturing significant changes in carbon storage in the volume.

7. -Section 2.3. Have the data been filtered for stationarity? Yes. Stationarity test is done.

8. -Section 2.3. The threshold u* seem to be large (0.2 or 0.3 m/s are pretty standard)? Any explanations associated with the sites? I don't think our sites have particularly denser canopies than other sites. May hint at advection processes playing a role?

9. -Section 2.3. Add one or two sentences explaining how you processed/averaged the meteorological data. Meteorological variables were sampled at 5 second intervals and averaged at a half-hourly scale. A two-step cleaning process was used to remove outliers in half-hourly meteorological data: coarse upper and lower thresholds were applied to half-hourly values to remove obvious outliers, and additional erroneous half-hourly data were removed from time series when instruments were known to be malfunctioning or visual inspection by multiple reviewers resulted in certain agreement that an outlier was present. Missing meteorological data of all lengths were gapfilled using extant data for the same half hours from either (in order of preference) a second sensor at the site, or an equivalent sensor from a nearby (1-3 km away) station in the network (sites described in Peichl et al., 2010).

10. -Section 2.4. Can you describe the uncertainties associated with the approach estimating phenological seasons? Uncertainties would be similar to gap-filling processes. While the estimation of the phenological seasons used 'non-gapfilled' GEP, this still includes the modeled RE and non-gapfilled NEE. A closing comment in Gonsamo et al. (2013) was that studies should also look into detailed uncertainty analysis with representative study sites from global distributions of plant functional types, as it was not previously done.

11. -Line 257 and Line 349. Clarify "responded similarly". REVISED – behaved similarly

12. -Line 255-262. Can you show the standard deviations of the annual mean Ta in Fig.1? Not entirely sure what was being asked, if it is a standard deviation of daily/annual temperature data or a comparison with the climate normals (deviations from mean). Added 30-year mean standard deviation in methods (8.0 ± 1.6°C).

13. -Line 265. Better explanation for the discrepancies is needed here. The discrepancies areover 300 umol m-2 day-1 in spring. Is it in the range of the measurement uncertainty? I'd suggest check the downward PAR to tease out the influences from the canopies and to evaluate the meteorological differences. This section was heavily edited. A paragraph was added in the methods section to highlight the reason for the discrepancies and how they were fixed. Once fixed, this sentence was edited accordingly.

14. -Line 267. Clarify "APAR was similar throughout the year". What are the values (mean and standard deviations) of the FPAR mentioned? At TP39, APAR exhibited a similar parabolic curve each year due to the seasonal amplitude in PARdn and the continuous presence of an apparently dense coniferous canopy promoting a nearly constant fraction (fPAR) of PARdn being absorbed (Fig 2a). Mean fPAR at TP39 was 0.9375 ± 0.05.

15. -Line 281. "Ts(5cm) at TP39 exceeded that of TPD" seems to suggest that thePARgroud at TPD is less, which implies that the APAR at TPD should be higher in summer and autumn. Please explain. However, during the summer and autumn of each year, $Ts_{5cm}$ at TP39 exceeded that of at

TPD due to differences in canopy cover. Also, a higher seasonal fPAR at TPD due to the presence of a dense deciduous canopy.

16. -Line 296. Can you explain why 6-year mean day of season growth was used instead of the days of individual years? The 6-year mean was used as it produced a better fit, but also helped explain a more long-term trend of growing season start dates.

17. -Line 327. Could you also add a sentence or two at the beginning of this paragraph to explain the physical meaning of the cumulative (seasonal and annual) fluxes, especially its differences from daily fluxes? Seasonal and total fluxes provide insight on each stands ability to sequester carbon and release water over interannually comparable timescales.

18. -Line 336. "spring was the only season when daily GEP was similar between the forests". As shown in Table 3, the seasonal GEP in spring show larger differences between sites, which I think to some extent contradicts with your statement in Line 336. Please reconcile. Also, when you compare the daily GEP for phenological seasons, how did you address the different lengths of the seasons (i.e. different number of data points)? The second part of this question answers the first part. They were similar in terms of daily rates of GEP not the total seasonal sum, which was impacted by the total length.

19. -There are a few places where I have similar comments as the previous one. I suggest adding some explanations for the statistical techniques (ANOVA and MANOVA) you used, which would shed some light on the discrepancies. -Line 338. The cumulative GEP in autumn (and 2012, 2014, 2015 summer) is higher at TP39 (except for 2012). Does it contradict the argument in Line 338? -Line 352. "RE was higher at TPD". But the cumulative RE were lower at TPD in spring and autumn. -Line 384. Seasonal ET is more different in spring not autumn. Also, "daily ET" or "seasonal ET"? The other reviewer suggested to remove the statistical techniques from the previous section. A sentence was added at Line 325 to briefly highlight the t-tests used. I revised the majority (if not all) the instances where I mentioned comparisons. I added time scales and key words to highlight the comparison of rates or averages in different periods.

20. -Line 339. "the 2016 summer was the only period . . .". Clarify "sufficiently". Also, it seems a false statement to me because summer GEP in 2013 and 2017 are also greater at TPD. REMOVED

21. -Line 353. Any figure or data to support this statement? Daily rates but REMOVED

22. -Line 399. How the low WUE in winter is reflected in Figure 6c? Did you only use data from spring to autumn? If so, clarify in the manuscript. All months were plotted

23. -Line 405. Can you clarify "similar results"? The LUE at TPD is 30% higher than that at TP39. Fixed the figure to implement corrected APAR data

24. -Line 406. Is the annual and seasonal LUE shown in the manuscript? If not, clarify it in the manuscript by adding "(data not shown)". Also, as shown in Table 3, TPD has lower annual GEP, which contradicts with the "greater GEP" referred here. Reconcile.
Similarly, TPD had higher annual (data not shown) and summer LUE ($p < 0.01$), although spring and autumn LUE was similar at both sites.

25. -Line 435. Do you mean "deciduous forest" instead of "conifer"? If not, add the correlation of annual NEP and summer RE for the conifer forest to Table 4. If the answer is yes, I'd suggest delete this sentence because it conveys the same meaning as the following two sentences. Meant conifer, but only included the key linear relationships

26. -Line 434-435. Can you add a brief explanation for the relationship of RE and spring Ta. Could be because of the fact that there's only 6 data points, but the warmest spring/year (2012) had the lowest annual RE, which the highest annual RE (2017) saw the coldest spring. Similarly, the coolest year in our record (2014) had a very warm spring.

27. -Line 439-448. The annual GEP has no significant relationships with meteorological variables as stated in Line 425. But this paragraph talks about GEP and meteo controls. Is it only summer GEP discussed in this paragraph? Yes, only looking at summer fluxes

28. -Line 439. What does "flux parameterizations" mean here? Is it explained in the methodology section? If not, I suggest adding it to the methods section. Yes. Added a new section to the methods: 2.4 Estimating effects of meteorological variables on carbon component fluxes

29. -Line 578. Is the assumption of similar carbon assimilation valid here given the different NEP? Changed to: Assuming similar daily rates of carbon assimilation (GEP)

30. -Table 3. Why the GEP sum for Jan 1 to SOS is missing? They seem to be available in Fig. 3. The assumption is that leaves aren't present so GEP remains zero until the SOS

31.-Table 4. Can you change this table to a figure similar to Fig. 4? The reasons are (i) you'd be able to show the standard deviations; (ii) the positive/negative correlation would be easier to tell. Ultimately chose not to, but it could be added to an appendix if needed

32.-Table 5. What model did you use for this calculation? Highlighted in methods (2.4)

33.-I notice the uncertainty analysis for measurements and calculations is missing. Can you add a brief subsection to Methods section (or wherever you find appropriate) dedicated to uncertainties? Added a paragraph on the uncertainty analysis in Section 2.3

***Minor comments:***

34. -I suggest changing all “warm temperatures/Ta” to “high temperatures/Ta” in the manuscript. REVISED

35. -Line 78. Clarify “controls”. Environmental/meteorological controls? REVISED determine the impact of meteorological controls on overall forest productivities

36. -Line 88-91. Are percentages available for the tree species? Not that we know of for the specific study area. Could probably be done by students in the future.

37. -Line 119. Did you use the momentum and heat fluxes in this study? If not, there’s no need to mention them. We do not. REMOVED

38. -Line 258. What is the value of “record Ta”? Also, “record high Ta”. Record high Ta conditions (exceeding 30-year mean daily maximum values)

39. -Line 315. Are the “days 230 to 290” 6-year mean? Explain. At both sites, the cumulative CDD from DOY 230 to 290 (mid-August to mid-October; loosely based on the range of dates in Oishi et al. [2018]). They used DOY 210 to 290.

40. -Line 325-326. This statement is not clear. Clarify or delete. DELETED

41. -Line 347. Define “outlier”. RE within the deciduous forest was greatly reduced, leading to an apparent outlier (exceeding mean and standard deviation) in annual RE

42. -Line 398. Clarify “the ratio of monthly ET”. Then modify the figure caption accordingly. … linear relationships of the monthly total ET and GEP (calculating WUE)

43. -Line 354-355. Confusing sentence. How do “comparable” results shape the “differences”? Rephrase. REMOVED

44. -Line 363. “for either site”? It’s hard to tell that the monthly NEP is negative at TPD in Figure 5b. Rephrase. Figure inset highlights the negative TP39 NEP during the summer

45. -Line 416. P value for being “significant”? “linear relationships of monthly Ta and monthly VPD”? Linear relationships of the 6-year monthly mean Ta and VPD ($p<0.01$).

46. -Be concise. See examples below. -Line 325. “at first glance” is not necessary. -Line 341-342. “significant daily minimums and maximums” seems to be repetitive as “highly variable”. -Line 417. Delete “,”. -Line 409-410. Delete “and”. Also, make the sentence clearer. -Line 372. “the highest” ——-> “highest”. REVISED ALL

47. -Given the different time scales used here, I suggest be more mindful about the uses of “daily, season, annual” when talking about fluxes. -Line 261. In “Ta at both sites”, do you mean “daily mean Ta”? daily mean Ta -Line 360. Change “The NEP in the conifer...” to “The annual NEP” or “The cumulative NEP”. annual NEP -Line 352. “spring and autumn RE was higher . . .”. Do you mean “daily RE in spring and summer”? Sentence removed -Line 410. Delete “When first considering . . .”. DELETED Change “ET”—-> “Annual ET”. -Line 325. Should “daily patterns” be “seasonal patterns”? Seasonal Also, substitute “expanded upon in Table3” with “the cumulative fluxes in Table 3”, just to be clear and accurate. REVISED

48. -I noticed quite a few miscitation or misspelling or inaccurate statements. See some examples below. -Line 270. “daily reductions in PAR (shouldn’t it be APAR?)”. -Line 401. 4.7 —-> 3.82 gC kg-1 H2O. -Line 406. R2 = 0.96 —-> R2 = 0.86. -Line 535. “increases” ——-> “decreases”? -Line 538. “most years” ——-> “half of the years”? - Line 553. “during drought years” is not accurate. It’s really just 2016. REVISED ALL

49. -I have a few minor comments regarding the tables and figures. See below. -Table 3. Can you highlight the highest and lowest annual fluxes with colored boxes? -Be more clear with figure captions, especially for words like “daily, monthly, seasonal, and annual”. For example, “A daily time series” in Fig. 2 is a bit confusing. -Figure 3. Green-red combination is not color-blind friendly. Also, can you annotate SOS, EOG, SOB, and EDS on the top panels? -Figure 4 caption. Two “and”. REVISED

````
````

**Response of carbon and water fluxes to meteorological and phenological variability in two Eastern North American forests of similar age but contrasting species composition – a multiyear comparison**

Eric R. Beamesderfer[1], M. Altaf Arain[1], Myroslava Khomik[1], Jason J. Brodeur[1], Brandon M. Burns[1]

[1]School of Geography and Earth Sciences and McMaster Centre for Climate Change, McMaster University, Hamilton, Ontario, L8S 4L8, Canada

*Correspondence to*: M. Altaf Arain (arainm@mcmaster.ca)

**Abstract.** The annual carbon and water dynamics of two Eastern North American temperate forests were compared over a six-year period from 2012 to 2017. The geographic location, forest age, soil, and climate were similar between the two stands, however, stand composition varied in terms of tree leaf-retention and shape strategy: one stand was a deciduous broadleaf forest, while the other was an evergreen needleleaf forest. The 6-year mean annual net ecosystem productivity (NEP) of the coniferous forest was slightly higher and more variable (218 ± 109 g C m$^{-2}$ yr$^{-1}$) compared to that of the deciduous forest NEP (200 ± 83 g C m$^{-2}$ yr$^{-1}$). Similarly, the 6-year mean annual evapotranspiration (ET) of the coniferous forest was higher (442 ± 33 mm yr$^{-1}$) than that of the deciduous forest (388 ± 34 mm yr$^{-1}$), but with similar interannual variability. Summer meteorology greatly impacted the carbon and water fluxes in both stands, however the degree of response varied among the two stands. In general, warm temperatures caused higher ecosystem respiration (RE), resulting in reduced annual NEP values – an impact that was more pronounced at the deciduous broadleaf forest compared to the evergreen needleleaf forest. However, during warm and dry years, the evergreen forest had largely reduced annual NEP values compared to the deciduous forest. Variability in annual ET at both forests was related most to the variability in annual air temperature (Ta), with the largest annual ET observed in the warmest years in the deciduous forest. Additionally, ET was sensitive to prolonged dry periods that reduced ET at both stands, although the reduction at the coniferous forest was relatively larger than that of the deciduous forest. If prolonged periods (weeks to months) of increased Ta and reduced precipitation are to be expected under future climates during summer months in the study region, our findings suggest that the deciduous broadleaf forest will likely remain an annual carbon sink, while the carbon sink-source status of the coniferous forest remains uncertain.

**1 Introduction**

Temperate forests play a significant role in the global carbon and water cycles through their photosynthetic $CO_2$ uptake and through their evapotranspiration (ET) (Huntington, 2006; Houghton et al., 2007). In Eastern North America, temperate forests are a significant sink of carbon and are an important element of future climate mitigation strategies; however, these forests have been going through transformations due to both natural and anthropogenic impacts for quite some time (Bonan, 2008; Cubasch et al., 2013; Weed et al., 2013). At the start of the 20th century, many of these forests were cleared for agricultural

**Deleted:** …year period from 2012 to 2017. The geographic location, forest age, soil, and climate were similar between the sites…wo stands, however, the species…tand composition varied in terms of tree leaf-retention and shape strategy: one stand was a deciduous broadleaf forest, while the other was an evergreen needleleaf forest. During the…he 6-year study period, the…mean annual net ecosystem productivity (NEP) of the coniferous forest was slightly higher and more variable (218 ± 109 g C m$^{-2}$ yr$^{-1}$) compared to that of the deciduous broadleaf …orest NEP of …200 ± 83 g C m$^{-2}$ yr$^{-1}$….. Similarly, the 6-year mean annual evapotranspiration (ET) of the conifer…oniferous forest over the 6-year study period …as higher (442 ± 33 mm yr$^{-1}$) compared to…han that of the broadleaf…eciduous forest (388 ± 34 mm yr$^{-1}$), but with similar interannual variability. Significant abnormalities in fluxes were measured between sites during drought years. …ummer meteorology greatly impacted the carbon and water fluxes at…n both sites, but to varying degrees and with varying however the degree of response varied among the two stands. In general, warm temperatures caused higher ecosystem respiration (RE), resulting in reduced mean…annual NEP values – an impact that was more pronounced at the deciduous broadleaf forest compared to the evergreen needle-leaf…eedleleaf forest. However, during drought…arm and dry years, the evergreen forest saw greater…ad largely reduced annual reduction in carbon sequestration…EP values compared to the deciduous forest. In the evergreen conifer forest, variability of summer meteorology greatly controlled the forest's annual carbon sink-source strength. Annual Variability in annual ET at both forests was driven by changes related most to the variability in annual air temperature (Ta), with the largest annual ET measured…bserved in the warmest years in the deciduous forest. Additionally, ET was sensitive to prolonged dry periods with increased Ta, greatly…hat reduced ET. During drought years,…at both stands, although the reduction at the carbon and water fluxes…oniferous forest was relatively larger than that of the deciduous forest were less sensitive to changes in temperature or water availability compared to the evergreen forest.… If longer…rolonged periods (weeks to months) of increased temperatures…a and larger…educed precipitation variability during summer months …re to be expected under future climates during summer months in the study region, our findings suggest the carbon sink capacity of…hat the deciduous broadleaf forest will continue…ikely remain an annual carbon sink, while that …he carbon sink-source status of the conifer…oniferous forest remains uncertain in the study region. ... [2]

purposes, effectively releasing carbon to the atmosphere (Bonan, 2008; Richart and Hewitt, 2008). With the rise of industrial development and movement of agricultural practices to other regions, many of these agricultural lands were abandoned and subsequently reforested through natural regrowth and afforestation practices (Canadell and Raupach, 2008). Currently, much of the forested area within the mixed-wood plains ecozone in the Great Lakes region of Canada and the USA is comprised of reforested or plantation stands which are in different stages of growth (Wiken et al., 2011).

Climate change and the associated changes in extreme weather events and the hydrologic cycle such as warmer spring temperatures, intense heat and drought events in the summer, early snowmelt, reduced snowfall, or increased freeze and thaw cycles in winter, may impact the ability of these local forests to sequester carbon, and thus impact regional forest-atmosphere interactions (Bonan, 2008; Allen et al., 2010; Teskey et al., 2015). However, climate change will impact deciduous and coniferous forest ecosystems differently due to their physiological differences. Even in deciduous and coniferous forests of similar age, geographic location, climatic conditions, and soil properties, differences in the timing and rate of photosynthesis, ecosystem respiration, and evapotranspiration may lead to asymmetries in the overall forest productivity, water use, and hence longevity and survival. Consequently, regions once dominated by coniferous forests may yield way to more deciduous species (Givnish, 2002; Bonan, 2008). Such a shift could disturb regional carbon and water budgets, as deciduous forests typically have shorter growing seasons, and higher photosynthetic rates and water use efficiencies when compared to coniferous forests (Givnish, 2002; Ciais et al., 2005). While many studies have examined the annual carbon and water fluxes within specific land use and forest types, to date, only a handful of studies have compared these fluxes among similar-age deciduous and coniferous forests growing in close proximity, in similar climatic and edaphic conditions (Gaumont-Guay et al., 2009; Baldocchi et al., 2010; Novick et al., 2015; Wagle et al., 2016). Even fewer studies have reported multi-annual time series.

This study examined the carbon and water fluxes in two Eastern North American forest ecosystems of different tree species but similar age, climate, and edaphic conditions during a 6-year period from 2012 to 2017. One stand was an 80-year old (as of 2019) evergreen needleleaf forest, while the other was a roughly 90-year old, broadleaf deciduous forest. The specific objectives of the study were: (1) examine seasonal and interannual dynamics of carbon and water exchanges in the two forests, (2) determine the impact of meteorological controls on overall forest productivities, and (3) analyse and contrast the varying responses of the two different species forests to extreme meteorological events such as heat and drought.

**2 Methods**

**2.1 Study Sites**

The two forests are located within 20 km of each other, situated on the north side of Lake Erie, in Norfolk County, Ontario, Canada (Table 1). These forests are a part of the Turkey Point Observatory in association with the global FluxNet program. The landscape in the region is dominated by agricultural lands, while plantation and regenerated forests cover a small fraction (~25%) of the land cover; accounting for the highest forest cover in southeastern Ontario. The broadleaf deciduous forest (from here on abbreviated and referred to as, Turkey Point Deciduous, TPD) was naturally regenerated in the early 1900s from

The result of a shifting climate…nd rate of photosynthesis, ecosystem respiration, and evapotranspiration may lead to different impacts on deciduous broadleaf and evergreen needleleaf ... [5]

Previous carbon and water studies conducted within the conifer forests of the Turkey Point Observatory have been reported in literature (i.e. Arain and Restrepo-Coupe, 2005; Peichl and Arain... [6]

[revised manuscript text omitted]

$$GEP = \frac{\alpha PAR\, A_{max}}{\alpha PAn + A_{max}} \times f(Ts_{5cm}) \times f(VPD) \times f(\theta_{0\text{-}30cm})\,, \qquad (3)$$

where the first term is a rectangular hyperbolic functional relationship between PAR and GEP, defined by the values of the photosynthetic flux per quanta (α, quantum yield) and the light-saturated rate of $CO_2$ fixation ($A_{max}$). The remaining terms use the functional form introduced in Equation 2 to described the responses of GEP to Ts, vapor pressure deficit (VPD), and $\theta_{0\text{-}30cm}$, respectively. Parameters were optimized using the same approach described above. Finally, gaps in the NEP timeseries were filled using the differences between the filled GEP and modeled RE timeseries.

Following the aforementioned threshold and point cleaning, gaps in the latent heat flux (LE), and therefore the mass equivalent evapotranspiration, were filled using an artificial neural network which utilized Rn, wind speed, $Ts_{5cm}$, VPD, and $\theta_{0\text{-}30cm}$ (Brodeur, 2014). Following the approach outlined by Amiro et al. (2006), any remaining gaps in LE data were filled using a moving window linear regression method. Past studies examining the relationships between ET and meteorological variables for the forests of the Turkey Point Observatory have found Ta to largely drive ET, with smaller secondary effects driven by VPD and $\theta_{0\text{-}30cm}$ during low water or high heat periods (McLaren et al., 2008; MacKay et al., 2012; Skubel et al., 2015; Burns, 2017). All data processing and analyses were completed using MatLab R2014b software (The MathWorks, Inc.).

- **Deleted:** function of
- **Deleted:** &
- **Deleted:** fitted ordinary least square non-linear regression models applied to half-hourly nighttime NEE. This was done in order to describe the relationship between RE and $Ts_{5cm}$, representing
- **Deleted:** diurnal variation in Ta, modified by a soil moisture (θ) function as shown (Brodeur, 2014):
- **Deleted:** ¶
- **Deleted:** are fitted temperature response parameters describing the RE and
- **Deleted:** relationship, while
- **Deleted:** ranging
- **Deleted:** , as a
- **Deleted:** of the independent variable, $\theta_{0\text{-}30cm}$, acting to scale the RE relationship. Modeled daytime RE was then added to measured NEP to estimate the gross ecosystem productivity (GEP). When
- **Deleted:** periods existed with gaps in NEP, GEP was
- **Deleted:** using a rectangular hyperbolic function
- **Deleted:** 2
- **Deleted:** defines the
- **Deleted:** through
- **Deleted:** calculation
- **Deleted:** describe sigmoidal scales (ranging from 0
- **Deleted:** 1)
- **Deleted:** During half-hourly periods where meteorological data
- **Deleted:** missing
- **Deleted:** RE and GEP
- **Deleted:** a non-linear regression approach and a marginal distribution sampling approach (Reichstein et al., 2005; Brodeur, 2014). Missing NEP data due to instrumentation errors, maintenance, calibrations, and power outages, were filled as the difference…
- **Deleted:** and modeled GEP.
- **Deleted:** These gapfilling methods were further used to explore relationships between each sites component fluxes (RE and GEP) and controlling meteorological and edaphic variables using residual analyses. RE and GEP gapfilling models (see above) were parameterized using pooled data from the phenologically-derived summer months (end of greenup to start of browndown, defined in the next section) for all years (2012 to 2017). Furthermore, much like other temperate forests, due to the influence of canopy cover and optimal growing conditions, summer was identified as a key period for carbon uptake at both sites. Based on the functional ... [8]
- **Deleted:** (ET),
- **Deleted:** net radiation (
- **Deleted:** ),
- **Deleted:** stressed

[revised manuscript text omitted]
 θ declines in 2012, 2016, and 2017 (Fig. 2f). The magnitudes again were different, but each forest experienced similar declining θ and the subsequent recharging θ analogous to local P events. In the summer, θ was typically lower at TPD than TP39, while during all other times of the year θ at TP39 was higher (Fig. 2g).

**3.2 Phenological Variability**

The meteorological conditions had a significant impact on the timing and duration of key phenological events, although ultimately the response was governed by different leaf-strategies of the various dominant tree species in each forest. The phenological transition dates and seasons calculated from EC-flux data are shown in Table 2 and Fig. 3. The SOS varied considerably between the two forests, with the SOS at the evergreen forest, TP39, beginning on average 38 ± 14 days earlier than at the deciduous forest, TPD. TP39 experienced a larger variation in SOS dates, spanning a period of 26 days between the earliest (10 March 2012; day of year [DOY] 70) and latest (6 April 2014; DOY 96) years, while TPD varied by 11 days between years.

Growing degree days (GDD) are a proxy used to assess the amount of heat the ecosystem has absorbed, as a result of increasing air temperatures. The response of the forest to increasing GDD was shown to be a trigger for the SOS. The total GDD from the start of the year (January 1st, DOY 1) to 6-year mean day of season growth (25 March; DOY 84), was found to be highly correlated to SOS at TP39 ($R^2 = 0.81$), but not at TPD (Fig. 4a & 4b). GDD for DOY 117-127 (27 April to 7 May; which represents the range of 6-year mean SOS data ± one standard deviation) was found to significantly influence the SOS

Deleted: , beginning with the amount of photosynthetically active radiation absorbed by the forest canopy (APAR, Fig. 2a). The use of different sensors (Table A1) and corresponding coefficients needed for the calculation of incoming PARdn, likely led to some of the discrepancies in the total magnitudes of APAR. However, the shapes ... [9]

[revised manuscript text omitted]
 Ta. Even with similar climatic forcings (i.e. Ta) seasonal deviations in $Ts_{5cm}$ were found, likely influenced by the opposing forest canopy characteristics (Palmroth et al., 2005; Stoy et al., 2006). Ts was linked to the proportion of incoming radiation penetrating the forest canopy, reaching the forest floor. In all years, mean daily $Ts_{5cm}$ at the conifer forest was higher during each summer, but lower than that of the deciduous forest during the rest of the year. In the conifer forest, branches and needles were closely clumped while the canopy remained comparatively open, leading to minor annual variations in fPAR by the forest canopy and soil, in line with Brummer et al. (2012). In the deciduous forest, $Ts_{5cm}$ was higher when leaves were absent and a higher fraction of incoming radiation was directly absorbed by the soil. Following the development and closure of the forest canopy in spring, deciduous $Ts_{5cm}$ was lower than the conifer forest in our study, which was in line with other similar studies (i.e. Lee et al., 2010; Augusto et al., 2015).

In general, both forests had similar VPD trends in all years while TP39 had somewhat higher VPD compared to TPD, except in the record warm year of 2012. The higher VPD at the deciduous forest in 2012 could be due to the relative unresponsiveness of stomata to higher VPD typical of broadleaved species, or the suggested larger leaf boundary layers in deciduous trees, where VPD measured above a canopy can be greater than what leaves experience (Baldocchi and Vogel, 1996; Baldocchi et al., 2002; Stokes et al., 2006).

[revised manuscript text omitted]

*Author contributions*. ERB collected, cleaned, and processed the data with help from JJB and BMB. ERB, MAA, and MK designed the experiment, with grants received by MAA. ERB and JJB performed the statistical analyses. ERB interpreted the data, prepared the figures, and wrote the manuscript with editorial contributions from all authors.

*Competing interests*. The authors declare no competing interests.

*Acknowledgements*. This study was funded by the Natural Sciences and Engineering Research Council (NSREC), the Global Water Futures Program (GWF), and the Ontario Ministry of Environment, Conservation and Parks (MOECP). Funding from the Canadian Foundation of Innovation (CFI) through New Opportunity and Leaders Opportunity Fund and Ontario Research Fund of the Ministry of Research and Innovation is also acknowledged. In kind support from the Ontario Ministry of Natural Resources and Forestry (OMNRF). The St. Williams Conservation Reserve Community Council and the Long Point Region Conservation Authority (LPRCA) are also acknowledged. We acknowledge support from Zoran Nesic at the University of British Columbia in assistance with flux measurements at our site, and members of the Hydrometeorology and Climatology lab at McMaster University for their continued support at both sites.

[revised manuscript text omitted]
| | *D* | *145* | *155* | *160* | *146* | *154* | *159* | *153 ± 6* |
| Peak of Season (Midpoint between EOG & SOB) | **C** | **214** | **205** | **202** | **193** | **212** | **201** | **204 ± 8** |
| | *D* | *198* | *199* | *205* | *193* | *203* | *207* | *201 ± 5* |
| Start of Browndown (SOB, start of senescence) | **C** | **271** | **258** | **258** | **257** | **270** | **248** | **260 ± 9** |
| | *D* | *257* | *249* | *255* | *249* | *262* | *261* | *255 ± 6* |
| Mid of Browndown (MOB, fastest senescence) | **C** | **287** | **292** | **287** | **289** | **305** | **287** | **291 ± 7** |
| | *D* | *275* | *273* | *274* | *271* | *286* | *282* | *277 ± 6* |
| End of Season (EOS) | **C** | **314** | **351** | **338** | **345** | **366** | **354** | **345 ± 17** |
| | *D* | *306* | *314* | *307* | *309* | *328* | *318* | *314 ± 8* |
| **Phenologically-Defined Seasons** | | **2012** | **2013** | **2014** | **2015** | **2016** | **2017** | **Mean** |
| Spring (EOG – SOS) | **C** | **78** | **64** | **58** | **48** | **84** | **80** | **69 ± 14** |
| | *D* | *25* | *39* | *34* | *28* | *28* | *34* | *31 ± 5* |
| Summer (SOB – EOG) (LOCC, Length of Canopy Closure) | **C** | **124** | **97** | **105** | **117** | **112** | **89** | **107 ± 13** |
| | *D* | *112* | *94* | *95* | *103* | *107* | *102* | *102 ± 7* |
| Autumn (EOS – SOB) | **C** | **43** | **94** | **80** | **89** | **96** | **106** | **85 ± 22** |
| | *D* | *49* | *65* | *52* | *61* | *67* | *57* | *58 ± 7* |
| Length of Growing Season (LOS) | **C** | **245** | **255** | **242** | **254** | **292** | **275** | **260 ± 19** |
| | *D* | *186* | *198* | *180* | *191* | *202* | *193* | *192 ± 8* |

**Table 3.** Seasonal and annual sums of eddy covariance (EC) measured carbon (GEP, RE, and NEP, g C $m^{-2}$ $yr^{-1}$) and water fluxes (ET, mm $yr^{-1}$) from 2012 to 2017 for both the conifer (TP39, **bolded C**) and deciduous (TPD, *italicized D*) forests. The phenologically-defined seasonal dates were calculated using the timing of transitions in phenological dates, outlined in Table 2. The six-year mean and standard deviations are also included for each row.

| | Season | | 2012 | 2013 | 2014 | 2015 | 2016 | 2017 | Mean |
|---|---|---|---|---|---|---|---|---|---|
| **GEP Sum** | Jan 1 to SOS | -- | -- | -- | -- | -- | -- | -- | -- |
| | Spring (SOS to EOG) | **C** | **308** | **306** | **279** | **213** | **359** | **418** | **314 ± 70** |
| | | *D* | *104* | *197* | *165* | *117* | *129* | *174* | *148 ± 36* |
| | Summer (EOG to SOB) | **C** | **990** | **942** | **1070** | **1160** | **1014** | **930** | **1018 ± 86** |
| | | *D* | *942* | *949* | *1023* | *1006* | *1084* | *1070* | *1012 ± 59* |
| | Autumn (SOB to EOS) | **C** | **132** | **264** | **265** | **340** | **249** | **377** | **271 ± 85** |
| | | *D* | *147* | *239* | *200* | *240* | *219* | *213* | *210 ± 34* |
| | EOS to Dec 31 | -- | -- | -- | -- | -- | -- | -- | -- |
| | Annual | **C** | **1452** | **1501** | **1601** | **1701** | **1617** | **1709** | **1597 ± 104** |
| | | *D* | *1198* | *1369* | *1382* | *1347* | *1420* | *1447* | *1360 ± 87* |
| **RE Sum** | Jan 1 to SOS | **C** | **66** | **83** | **78** | **79** | **82** | **81** | **78 ± 6** |
| | | *D* | *167* | *107* | *129* | *109* | *163* | *170* | *141 ± 30* |
| | Spring (SOS to EOG) | **C** | **205** | **205** | **169** | **122** | **233** | **276** | **202 ± 53** |
| | | *D* | *78* | *151* | *133* | *109* | *109* | *144* | *121 ± 27* |
| | Summer (EOG to SOB) | **C** | **908** | **718** | **809** | **790** | **888** | **735** | **808 ± 78** |
| | | *D* | *500* | *672* | *581* | *714* | *684* | *700* | *642 ± 84* |
| | Autumn (SOB to EOS) | **C** | **142** | **272** | **269** | **310** | **302** | **434** | **288 ± 94** |
| | | *D* | *138* | *269* | *196* | *259* | *266* | *252* | *230 ± 52* |
| | EOS to Dec 31 | **C** | **77** | **14** | **33** | **39** | **--** | **13** | **35 ± 26** |
| | | *D* | *82* | *64* | *84* | *110* | *55* | *65* | *77 ± 20* |
| | Annual | **C** | **1386** | **1282** | **1345** | **1328** | **1492** | **1525** | **1393 ± 96** |
| | | *D* | *954* | *1250* | *1110* | *1283* | *1260* | *1317* | *1196 ± 138* |
| **NEP Sum** | Jan 1 to SOS | **C** | **-58** | **-74** | **-68** | **-73** | **-66** | **-66** | **-67 ± 6** |
| | | *D* | *-117* | *-79* | *-88* | *-82* | *-129* | *-130* | *-104 ± 24* |
| | Spring (SOS to EOG) | **C** | **103** | **101** | **110** | **92** | **128** | **144** | **113 ± 19** |
| | | *D* | *25* | *45* | *30* | *4* | *18* | *29* | *25 ± 14* |
| | Summer (EOG to SOB) | **C** | **80** | **223** | **262** | **374** | **127** | **196** | **210 ± 104** |
| | | *D* | *442* | *276* | *441* | *288* | *398* | *371* | *369 ± 73* |
| | Autumn (SOB to EOS) | **C** | **-12** | **-5** | **-6** | **33** | **-48** | **-51** | **-15 ± 31** |
| | | *D* | *16* | *-26* | *4* | *-18* | *-46* | *-37* | *-18 ± 24* |
| | EOS to Dec 31 | **C** | **-35** | **-12** | **-30** | **-24** | **--** | **-12** | **-23 ± 10** |
| | | *D* | *-68* | *-56* | *-79* | *-103* | *-51* | *-58* | *-69 ± 19* |
| | Annual | **C** | **76** | **228** | **263** | **395** | **139** | **208** | **218 ± 109** |
| | | *D* | *292* | *156* | *305* | *90* | *185* | *169* | *200 ± 83* |
| **ET Sum** | Jan 1 to SOS | **C** | **22** | **23** | **11** | **19** | **13** | **15** | **17 ± 5** |
| | | *D* | *56* | *28* | *33* | *24* | *39* | *44* | *37 ± 12* |
| | Spring (SOS to EOG) | **C** | **105** | **97** | **67** | **65** | **85** | **106** | **87 ± 18** |
| | | *D* | *36* | *55* | *45* | *31* | *39* | *48* | *42 ± 9* |
| | Summer (EOG to SOB) | **C** | **315** | **277** | **260** | **286** | **249** | **210** | **266 ± 36** |
| | | *D* | *283* | *231* | *213* | *219* | *266* | *237* | *242 ± 27* |
| | Autumn (SOB to EOS) | **C** | **43** | **73** | **82** | **67** | **64** | **97** | **71 ± 18** |
| | | *D* | *45* | *63* | *50* | *66* | *69* | *64* | *60 ± 9* |
| | EOS to Dec 31 | **C** | **15** | **2** | **6** | **4** | **--** | **3** | **6 ± 5** |
| | | *D* | *14* | *9* | *12* | *14* | *11* | *15* | *12 ± 2* |
| | Annual | **C** | **495** | **468** | **421** | **436** | **408** | **424** | **442 ± 33** |
| | | *D* | *428* | *381* | *350* | *349* | *417* | *403* | *388 ± 34* |

**Table 4.** Select linear relationships between total annual water (ET, mm $yr^{-1}$) and carbon (RE and NEP, g C $m^{-2}$ $yr^{-1}$) flux measurements and both meteorological (i.e. VPD, Ta, $\theta_{0\text{-}30cm}$) and phenological (i.e. spring length, carbon uptake start) variables (annual or seasonal) from 2012 to 2017. In each section, the $R^2$ is for the relationship to the specified annual flux.

| **Conifer** | **2012** | **2013** | **2014** | **2015** | **2016** | **2017** | **$R^2$** |
|---|---|---|---|---|---|---|---|
| Annual RE (g C $m^{-2}$ $yr^{-1}$) | 1386 | 1282 | 1345 | 1328 | 1492 | 1525 | -- |
| Summer $\theta_{0\text{-}30cm}$ ($m^3$ $m^{-3}$) | 0.083 | 0.097 | 0.090 | 0.096 | 0.071 | 0.076 | 0.89 |
| | | | | | | | |
| Annual NEP (g C $m^{-2}$ $yr^{-1}$) | 76 | 228 | 263 | 395 | 139 | 208 | -- |
| Spring Length (Days) | 78 | 64 | 58 | 48 | 84 | 80 | 0.75 |
| Summer Ta (°C) | 21.1 | 20.3 | 19.9 | 20.0 | 21.1 | 20.7 | 0.73 |
| Summer NEP (g C $m^{-2}$) | 80 | 223 | 262 | 374 | 127 | 196 | 0.99 |

| **Deciduous** | **2012** | **2013** | **2014** | **2015** | **2016** | **2017** | **$R^2$** |
|---|---|---|---|---|---|---|---|
| Annual ET (mm $yr^{-1}$) | 428 | 381 | 350 | 349 | 417 | 403 | -- |
| Annual Ta (°C) | 11.8 | 9.2 | 8.0 | 9.2 | 10.6 | 10.0 | 0.84 |
| Winter $\theta_{0\text{-}30cm}$ ($m^3$ $m^{-3}$) | 0.131 | 0.118 | 0.116 | 0.101 | 0.133 | 0.127 | 0.83 |
| | | | | | | | |
| Annual RE (g C $m^{-2}$ $yr^{-1}$) | 954 | 1250 | 1110 | 1283 | 1260 | 1317 | -- |
| Spring Ta (°C) | 16.6 | 15.1 | 16.1 | 15.1 | 15.6 | 14.0 | 0.77 |
| | | | | | | | |
| Annual NEP (g C $m^{-2}$ $yr^{-1}$) | 292 | 156 | 305 | 90 | 185 | 169 | -- |
| Summer RE (g C $m^{-2}$) | 500 | 672 | 581 | 714 | 684 | 700 | 0.80 |

**Table 5.** Results of the two-step parameterization process used to explore interannual differences in controlling meteorological variables (i.e. Ta, VPD, PAR, $\theta_{0\text{-}30cm}$) and their impacts on annual RE and GEP during the phenological summer (end of greenup to the start of browndown) for the coniferous and deciduous forests from 2012 to 2017. These normalized values show the cumulative effect of the meteorological variable in reducing GEP and RE from their theoretical maximum values. Higher values (closer to 1) represent more favorable summer conditions for GEP and RE.

| **Conifer** | **2012** | **2013** | **2014** | **2015** | **2016** | **2017** |
|---|---|---|---|---|---|---|
| GEP: Ta | 0.994 | 0.990 | 0.987 | 0.981 | 1.00 | 0.997 |
| GEP: VPD | 0.939 | 1.00 | 1.00 | 0.981 | 0.914 | 0.975 |
| GEP: PAR | 0.949 | 0.950 | 0.946 | 0.956 | 1.00 | 0.950 |
| GEP: $\theta_{0\text{-}30cm}$ | 0.956 | 1.00 | 0.998 | 0.993 | 0.976 | 0.973 |
| GEP: All | 0.846 | 0.941 | 0.932 | 0.914 | 0.892 | 0.899 |
| RE: $\theta_{0\text{-}30cm}$ | 0.958 | 1.00 | 0.996 | 0.991 | 0.968 | 0.965 |

| **Deciduous** | **2012** | **2013** | **2014** | **2015** | **2016** | **2017** |
|---|---|---|---|---|---|---|
| GEP: Ta | 1.00 | 0.971 | 0.974 | 0.967 | 0.989 | 0.974 |
| GEP: VPD | 0.871 | 1.00 | 0.998 | 0.998 | 0.946 | 0.989 |
| GEP: PAR | 0.978 | 0.938 | 0.955 | 0.953 | 1.00 | 0.956 |
| GEP: $\theta_{0\text{-}30cm}$ | 1.00 | 1.00 | 1.00 | 1.00 | 1.00 | 1.00 |
| GEP: All | 0.852 | 0.911 | 0.929 | 0.920 | 0.936 | 0.920 |
| RE: $\theta_{0\text{-}30cm}$ | 0.976 | 1.00 | 0.997 | 1.00 | 0.965 | 0.992 |

[Figure]

**Figure 1**. Daily above canopy air temperature (Ta, red dots) measured from 2012 to 2017 at the (a) conifer forest (TP39) and (b) deciduous forest (TPD), with the grey shading and black line corresponding to the 30-year Environment Canada (Delhi station) minimum and maximum range of daily Ta and mean daily Ta, respectively. Values shown represent the annual mean Ta for each year of measurements. Also included is the (c) comparison of daily Ta at TP39 and TPD.

[revised manuscript text omitted]